# Active Test-Time Adaptation: Theoretical Analyses and An Algorithm

**Shurui Gui**[*]
Texas A&M University
College Station, TX 77843
`shurui.gui@tamu.edu`

**Xiner Li**[*]
Texas A&M University
College Station, TX 77843
`lxe@tamu.edu`

**Shuiwang Ji**
Texas A&M University
College Station, TX 77843
`sji@tamu.edu`

## Abstract

Test-time adaptation (TTA) addresses distribution shifts for streaming test data in unsupervised settings. Currently, most TTA methods can only deal with minor shifts and rely heavily on heuristic and empirical studies. To advance TTA under domain shifts, we propose the novel problem setting of active test-time adaptation (ATTA) that integrates active learning within the fully TTA setting. We provide a learning theory analysis, demonstrating that incorporating limited labeled test instances enhances overall performances across test domains with a theoretical guarantee. We also present a sample entropy balancing for implementing ATTA while avoiding catastrophic forgetting (CF). We introduce a simple yet effective ATTA algorithm, known as SimATTA, using real-time sample selection techniques. Extensive experimental results confirm consistency with our theoretical analyses and show that the proposed ATTA method yields substantial performance improvements over TTA methods while maintaining efficiency and shares similar effectiveness to the more demanding active domain adaptation (ADA) methods. Our code is available at https://github.com/divelab/ATTA.

## 1 Introduction

Deep learning has achieved remarkable success across various fields, attaining high accuracy in numerous applications (Krizhevsky et al., 2017; Simonyan and Zisserman, 2014). Nonetheless, When training and test data follow distinct distributions, models often experience significant performance degradation during test. This phenomenon, known as the distribution shift or out-of-distribution (OOD) problem, is extensively studied within the context of both domain generalization (DG) (Gulrajani and Lopez-Paz, 2020; Koh et al., 2021; Gui et al., 2022) and domain adaptation (DA) (Ganin et al., 2016; Sun and Saenko, 2016). While these studies involve intensive training of models with considerable generalization abilities towards target domains, they overlook an important application property; namely, continuous adaptivity to real-time streaming data under privacy, resource, and efficiency constraints. This gap leads to the emergence of test-time adaptation (TTA) tasks, targeting on-the-fly adaptation to continuous new domains during the test phase or application deployment. The study of TTA encompasses two main categories; namely test-time training (TTT) methods (Sun et al., 2020; Liu et al., 2021c) and fully test-time adaptation (FTTA) (Niu et al., 2023; Wang et al., 2021). The TTT pipeline incorporates retraining on the source data, whereas FTTA methods adapt arbitrary pre-trained models to the given test mini-batch by conducting entropy minimization, without access to the source data. Nevertheless, most TTA methods can only handle corrupted distribution shifts (Hendrycks and Dietterich, 2019b) (e.g., Gaussian noise,) and rely heavily on human intuition or empirical studies. To bridge this gap, our paper focuses on tackling significant domain distribution shifts in real time with theoretical insights.

We investigate FTTA, which is more general and adaptable than TTT, particularly under data accessibility, privacy, and efficiency constraints. Traditional FTTA aims at adapting a pre-trained model to streaming test-time data from diverse domains under unsupervised settings. However, recent works (Lin et al., 2022; Pearl, 2009) prove that *it is theoretically infeasible to achieve OOD generalization without extra information* such as environment partitions. Since utilizing environment partitions requires heavy pretraining, contradicting the nature of TTA, we are motivated to incorporate extra information in a different way, *i.e.*, integrating a limited number of labeled test-time samples to alleviate distribution shifts, following the active learning (AL) paradigm (Settles, 2009). To this end, we propose the novel problem setting of active test-time adaptation (ATTA) by incorporating

---

[*]Equal contributions

AL within FTTA. ATTA faces two major challenges; namely, catastrophic forgetting (CF) (Kemker et al., 2018; Li and Hoiem, 2017) and real-time active sample selection. CF problem arises when a model continually trained on a sequence of domains experiences a significant performance drop on previously learned domains, due to the inaccessibility of the source data and previous test data. Real-time active sample selection requires AL algorithms to select informative samples from a small buffer of streaming test data for annotation, without a complete view of the test distribution.

In this paper, we first formally define the ATTA setting. We then provide its foundational analysis under the learning theory's paradigm to guarantee the mitigation of distribution shifts and avoid CF. Aligned with our empirical validations, while the widely used entropy minimization (Wang et al., 2021; Grandvalet and Bengio, 2004) can cause CF, it can conversely become the key to preventing CF problems with our sample selection and balancing techniques. Building on the analyses, we then introduce a simple yet effective ATTA algorithm, SimATTA, incorporating balanced sample selections and incremental clustering. Finally, we conducted a comprehensive experimental study to evaluate the proposed ATTA settings with three different settings in the order of low to high requirement restrictiveness, *i.e.*, TTA, Enhanced TTA, and Active Domain Adaptation (ADA). Intensive experiments indicate that ATTA jointly equips with the efficiency of TTA and the effectiveness of ADA, rendering an uncompromising real-time distribution adaptation direction.

**Comparison to related studies.** Compared to TTA methods, ATTA requires extra active labels, but the failure of TTA methods (Sec. 5.1) and the theoretical proof of Lin et al. (2022); Pearl (2009) justify its necessity and rationality. Compared to active online learning, ATTA focuses on lightweight real-time fine-tuning without round-wise re-trainings as Saran et al. (2023) and emphasizes the importance of CF avoidance instead of resetting models and losing learned distributions. In fact, active online learning is partially similar to our enhanced TTA setting (Sec. 5.2. Compared to ADA methods (Prabhu et al., 2021; Ning et al., 2021), ATTA does not presuppose access to source data, model parameters, or pre-collected target samples. Furthermore, without this information, ATTA can still perform on par with ADA methods (Sec. 5.3). The recent source-free active domain adaptation (SFADA) method SALAD (Kothandaraman et al., 2023) still requires access to model parameter gradients, pre-collected target data, and training of additional networks. Our ATTA, in contrast, with non-regrettable active sample selection on streaming data, is a much lighter and more realistic approach distinct from ADA and SFADA. More related-work discussions are provided in Appx. C.

## 2 THE ACTIVE TEST-TIME ADAPTATION FORMULATION

TTA methods aim to solve distribution shifts by dynamically optimizing a pre-trained model based on streaming test data. We introduce the novel problem setting of Active Test-Time Adaptation (ATTA), which incorporates active learning during the test phase. In ATTA, the model continuously selects the most informative instances from the test batch to be labeled by an explicit or implicit oracle (*e.g.*, human annotations, self-supervised signals) and subsequently learned by the model, aiming to improve future adaptations. Considering the labeling costs in real-world applications, a "budget" is established for labeled test instances. The model must effectively manage this budget distribution and ensure that the total number of label requests throughout the test phase does not surpass the budget.

We now present a formal definition of the ATTA problem. Consider a pre-trained model $f(x; \phi)$ with parameters $\phi$ trained on the source dataset $D_S = (x, y)_{|D_S|}$, with each data sample $x \in \mathcal{X}$ and a label $y \in \mathcal{Y}$. We aim to adapt model parameters $\theta$, initialized as $\phi$, to an unlabeled test-time data stream. The streaming test data exhibit distribution shifts from the source data and varies continuously with time, forming multiple domains to which we must continuously adapt. The test phase commences at time step $t = 1$ and the streaming test data is formulated in batches. The samples are then actively selected, labeled (by the oracle) and collected as $D_{te}(t) = ActAlg(U_{te}(t))$, where $ActAlg(\cdot)$ denotes an active selection/labeling algorithm. The labeled samples $D_{te}(t)$ are subsequently incorporated into the ATTA training set $D_{tr}(t)$. Finally, we conclude time step $t$ by performing ATTA training, updating model parameters $\theta(t)$ using $D_{tr}(t)$, with $\theta(t)$ initialized as the previous final state $\theta(t-1)$.

**Definition 1** (The ATTA problem). Given a model $f(x; \theta)$, with parameters $\theta$, initialized with parameters $\theta(0) = \phi$ obtained by pre-training on source domain data, and streaming test data batches $U_{te}(t)$ continually changing over time, the ATTA task aims to optimize the model at any time step $t$ (with test phase commencing at $t = 1$) as

$$\theta(t)^* := \underset{\theta(t)}{\operatorname{argmin}}(\mathbb{E}_{(x,y,t) \in D_{tr}(t)}[\ell_{CE}(f(x; \theta(t)), y)] + \mathbb{E}_{(x,t) \in U_{te}(t)}[\ell_U(f(x; \theta(t)))]), \tag{1}$$

$$\text{where} \quad D_{tr}(t) = \begin{cases} \emptyset, & t = 0 \\ D_{tr}(t-1) \cup D_{te}(t), & t \geq 1, \end{cases} \quad s.t. \quad |D_{tr}(t)| \leq \mathcal{B}, \quad (2)$$

$D_{te}(t) = ActAlg(U_{te}(t))$ is actively selected and labeled, $\ell_{CE}$ is the cross entropy loss, $\ell_U$ is an unsupervised learning loss, and $\mathcal{B}$ is the budget.

## 3 THEORETICAL STUDIES

In this section, we conduct an in-depth theoretical analysis of TTA based on learning theories. We mainly explore two questions: How can significant distribution shifts be effectively addressed under the TTA setting? How can we simultaneously combat the issue of CF? Sec. 3.1 provides a solution with theoretical guarantees to the first question, namely, active TTA (ATTA), along with the conditions under which distribution shifts can be well addressed. Sec. 3.2 answers the second question with an underexplored technique, *i.e.*, selective entropy minimization, building upon the learning bounds established in Sec. 3.1. We further validate these theoretical findings through experimental analysis. Collectively, we present a theoretically supported ATTA solution that effectively tackles both distribution shift and CF.

### 3.1 ALLEVIATING DISTRIBUTION SHIFTS THROUGH ACTIVE TEST-TIME ADAPTATION

Traditional TTA is performed in unsupervised or self-supervised context. In contrast, ATTA introduces supervision into the adaptation setting. In this subsection, we delve into learning bounds and establish generalization bounds to gauge the efficacy of ATTA in solving distribution shifts. We scrutinize the influence of active learning and evidence that the inclusion of labeled test instances markedly enhances overall performances across incremental test domains.

Following Kifer et al. (2004), we examine statistical guarantees for binary classification. A hypothesis is a function $h : \mathcal{X} \to \{0, 1\}$, which can serve as the prediction function within this context. In the ATTA setting, the mapping of $h$ varies with time as $h(x, t)$. We use $\mathcal{H}\Delta\mathcal{H}$-distance following Ben-David et al. (2010), which essentially provides a measure to quantify the distribution shift between two distributions $\mathcal{D}_1$ and $\mathcal{D}_2$, and can also be applied between datasets. The probability that an estimated hypothesis $h$ disagrees with the true labeling function $g : \mathcal{X} \to \{0, 1\}$ according to distribution $\mathcal{D}$ is defined as $\epsilon(h(t), g) = \mathbb{E}_{(x) \sim \mathcal{D}}[|h(x, t) - g(x)|]$, which we also refer to as the error or risk $\epsilon(h(t))$. While the source data is inaccessible under ATTA settings, we consider the existence of source dataset $D_S$ for accurate theoretical analysis. Thus, we initialize $D_{tr}$ as $D_{tr}(0) = D_S$. For every time step $t$, the test and training data can be expressed as $U_{te}(t)$ and $D_{tr}(t) = D_S \cup D_{te}(1) \cup D_{te}(2) \cup \cdots \cup D_{te}(t)$.

Building upon two lemmas (provided in Appx. D), we establish bounds on domain errors under the ATTA setting when minimizing the empirical weighted error using the hypothesis $h$ at time $t$.

**Theorem 1.** *Let $H$ be a hypothesis class of VC-dimension $d$. At time step $t$, for ATTA data domains $D_S, U_{te}(1), \cdots, U_{te}(t), \cdots, S_i$ are unlabeled samples of size $m$ sampled from each of the $t+1$ domains respectively. The total number of samples in $D_{tr}(t)$ is $N$ and the ratio of sample numbers in each component is $\boldsymbol{\lambda} = (\lambda_0, \cdots, \lambda_t)$. If $\hat{h}(t) \in \mathcal{H}$ minimizes the empirical weighted error $\hat{\epsilon}_{\boldsymbol{w}}(h(t))$ with the weight vector $\boldsymbol{w} = (w_0, \cdots, w_t)$ on $D_{tr}(t)$, and $h_j^*(t) = \arg\min_{h \in \mathcal{H}} \epsilon_j(h(t))$ is the optimal hypothesis on the $j$th domain, then for any $\delta \in (0, 1)$, with probability of at least $1 - \delta$, we have*

$$\epsilon_j(\hat{h}(t)) \leq \epsilon_j(h_j^*(t)) + 2 \sum_{i=0, i \neq j}^{t} w_i \left( \frac{1}{2} \hat{d}_{\mathcal{H}\Delta\mathcal{H}}(S_i, S_j) + 2\sqrt{\frac{2d \log(2m) + \log \frac{2}{\delta}}{m}} + \gamma_i \right) + 2C,$$

*where $C = \sqrt{\left( \sum_{i=0}^{t} \frac{w_i^2}{\lambda_i} \right) \left( \frac{d \log(2N) - \log(\delta)}{2N} \right)}$ and $\gamma_i = \min_{h \in \mathcal{H}} \{\epsilon_i(h(t)) + \epsilon_j(h(t))\}$. For future test domains $j = t + k$ ($k > 0$), assuming $k' = \arg\min_{k' \in \{0, 1, \dots t\}} d_{\mathcal{H}\Delta\mathcal{H}}(D(k'), U_{te}(t + k))$ and $\min d_{\mathcal{H}\Delta\mathcal{H}}(D(k'), U_{te}(t + k)) \leq \delta_D$, where $0 \leq \delta_D \ll +\infty$, then $\forall \delta$, with probability of at least $1 - \delta$, we have*

$$\epsilon_{t+k}(\hat{h}(t)) \leq \epsilon_{t+k}(h_{t+k}^*(t)) + \sum_{i=0}^{t} w_i \left( \hat{d}_{\mathcal{H}\Delta\mathcal{H}}(S_i, S_{k'}) + 4\sqrt{\frac{2d \log(2m) + \log \frac{2}{\delta}}{m}} + \delta_D + 2\gamma_i \right) + 2C.$$

The adaptation performance on a test domain is majorly bounded by the composition of (labeled) training data, estimated distribution shift, and ideal joint hypothesis performance, which correspond to $C$, $\hat{d}_{\mathcal{H}\Delta\mathcal{H}}(S_i, S_j)$, and $\gamma_i$, respectively. The ideal joint hypothesis error $\gamma_i$ gauges the inherent adaptability between domains. Further theoretical analysis are in Appx. D.

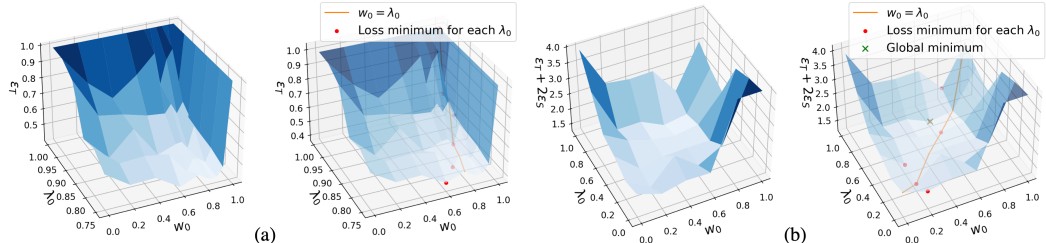

Figure 1: **(a) Empirical validation of Thm. 1.** We train a series of models on $N = 2000$ samples from the PACS (Li et al., 2017) dataset given different $\lambda_0$ and $w_0$ and display the test domain loss of each model. Red points are the test loss minimums given a fixed $\lambda_0$. The orange line is the reference where $w_0 = \lambda_0$. We observe that $w_0$ with loss minimums are located closed to the orange line but slightly smaller than $\lambda_0$, which validates our findings in Eq. (4). **(b) Empirical analysis with an uncertainty balancing.** Given source pre-trained models, we fine-tune the models on 500 samples with different $\lambda_0$ and $w_0$, and display the combined error surface of test and source error. Although a small $\lambda_0$ is good for test domain error, it can lead to non-trivial source error exacerbation. Therefore, we can observe that the global loss minimum (green X) locates in a relatively high-$\lambda_0$ region.

If we consider the multiple test data distributions as a single test domain, *i.e.*, $\bigcup_{i=1}^{t} U_{te}(i)$, Thm. 1 can be reduced into bounds for the source domain error $\epsilon_S$ and test domain error $\epsilon_T$. Given the optimal test/source hypothesis $h_T^*(t) = \arg\min_{h \in \mathcal{H}} \epsilon_T(h(t))$ and $h_S^*(t) = \arg\min_{h \in \mathcal{H}} \epsilon_S(h(t))$, we have

$$|\epsilon_T(\hat{h}(t)) - \epsilon_T(h_T^*(t))| \leq w_0 A + \sqrt{\frac{w_0^2}{\lambda_0} + \frac{(1-w_0)^2}{1-\lambda_0}} B, \tag{3a}$$

$$|\epsilon_S(\hat{h}(t)) - \epsilon_S(h_S^*(t))| \leq (1-w_0) A + \sqrt{\frac{w_0^2}{\lambda_0} + \frac{(1-w_0)^2}{1-\lambda_0}} B, \tag{3b}$$

where the distribution divergence term $A = \hat{d}_{\mathcal{H}\Delta\mathcal{H}}(S_0, S_T) + 4\sqrt{\frac{2d\log(2m) + \log\frac{2}{\delta}}{m}} + 2\gamma$, the empirical gap term $B = 2\sqrt{\frac{d\log(2N) - \log(\delta)}{2N}}$, $S_T$ is sampled from $\bigcup_{i=1}^{t} U_{te}(i)$, and $\gamma = \min_{h \in \mathcal{H}}\{\epsilon_0(h(t)) + \epsilon_T(h(t))\}$. Our learning bounds demonstrates the trade-off between the small amount of budgeted test-time data and the large amount of less relevant source data. Next, we provide an approximation of the condition necessary to achieve optimal adaptation performance, which is calculable from finite samples and can be readily applied in practical ATTA scenarios. Following Eq. (3.a), with approximately $B = c_1\sqrt{d/N}$, the optimal value $w_0^*$ to tighten the test error bound is a function of $\lambda_0$ and $A$:

$$w_0^* = \lambda_0 - \sqrt{\frac{A^2 N}{c_1^2 d - A^2 N \lambda_0 (1-\lambda_0)}}, \quad for \quad \lambda_0 \geq 1 - \frac{d}{A^2 N}, \tag{4}$$

where $c_1$ is a constant. Note that $\lambda_0 \geq 1 - \frac{d}{A^2 N}$ should be the satisfied condition in practical ATTA settings, where the budget is not sufficiently big while the source data amount is relatively large. The following theorem offers a direct theoretical guarantee that ATTA reduces the error bound on test domains in comparison to TTA without the integration of active learning.

**Theorem 2.** *Let $H$ be a hypothesis class of VC-dimension $d$. For ATTA data domains $D_S, U_{te}(1),$ $U_{te}(2), \cdots, U_{te}(t)$, considering the test-time data as a single test domain $\bigcup_{i=1}^{t} U_{te}(i)$, if $\hat{h}(t) \in \mathcal{H}$ minimizes the empirical weighted error $\hat{\epsilon}_{\mathbf{w}}(h(t))$ with the weight vector $\mathbf{w}$ on $D_{tr}(t)$, let the test error be upper-bounded with $|\epsilon_T(\hat{h}(t)) - \epsilon_T(h_T^*(t))| \leq EB_T(\mathbf{w}, \boldsymbol{\lambda}, N, t)$. Let $\mathbf{w}'$ and $\boldsymbol{\lambda}'$ be the weight and sample ratio vectors when no active learning is included, i.e., $\mathbf{w}'$ and $\boldsymbol{\lambda}'$ s.t. $w_0' = \lambda_0' = 1$ and $w_i' = \lambda_i' = 0$ for $i \geq 1$, then for any $\boldsymbol{\lambda} \neq \boldsymbol{\lambda}'$, there exists $\mathbf{w}$ s.t.*

$$EB_T(\mathbf{w}, \boldsymbol{\lambda}, N, t) < EB_T(\mathbf{w}', \boldsymbol{\lambda}', N, t). \tag{5}$$

Therefore, the incorporation of labeled test instances in ATTA theoretically enhances the overall performance across test domains, substantiating the significance of the ATTA setting in addressing distribution shifts. All proofs are provided in Appx. E. Finally, we support the theoretical findings with experimental analysis and show the numerical results of applying the principles on real-world datasets, as shown in Fig. 1. For rigorous analysis, note that our theoretical results rest on the underlying condition that $N$ should at least be of the same scale as $d$, according to the principles of VC-dimension theory. The empirical alignment of our experiments with the theoretical framework can be attributed to the assumption that fine-tuning a model is roughly equivalent to learning a model with a relatively small $d$. Experiment details and other validations can be found in Appx. H.

## 3.2 MITIGATING CATASTROPHIC FORGETTING WITH BALANCED ENTROPY MINIMIZATION

Catastrophic forgetting (CF), within the realm of Test-Time Adaptation (TTA), principally manifests as significant declines in overall performance, most notably in the source domain. Despite the lack of well-developed learning theories for analyzing training with series data, empirical studies have convincingly illustrated the crucial role of data sequential arrangement in model learning, thereby accounting for the phenomenon of CF. Traditionally, the mitigation of CF in adaptation tasks involves intricate utilization of source domain data. However, under FTTA settings, access to the source dataset is unavailable, leaving the problem of CF largely unexplored in the data-centric view.

To overcome this challenge of source dataset absence, we explore the acquisition of "source-like" data. In TTA scenarios, it is generally assumed that the amount of source data is considerably large. We also maintain this assumption in ATTA, practically assuming the volume of source data greatly surpasses the test-time budget. As a result, we can safely assume that the pre-trained model is well-trained on abundant source domain data $D_S$. Given this adequately trained source model, we can treat it as a "true" source data labeling function $f(x; \phi)$. The model essentially describes a distribution, $\mathcal{D}_{\phi,S}(\mathcal{X}, \mathcal{Y}) =$

Table 1: **Correlation analysis of high/low entropy samples and domains.** We use a source pre-trained model to select samples with lowest/highest entropy, and 1.retrain the model on 2000 samples; 2.fine-tune the model on 300 samples. We report losses on source/test domains for each setting, showing that low-entropy samples form distributions close to the source domain.

| Sample type | Retrain | | Fine-tune | |
|---|---|---|---|---|
| | $\epsilon_S$ | $\epsilon_T$ | $\epsilon_S$ | $\epsilon_T$ |
| Low entropy | 0.5641 | 0.8022 | 0.0619 | 1.8838 |
| High entropy | 2.5117 | 0.3414 | 0.8539 | 0.7725 |

$\{(x, \hat{y}) \in (\mathcal{X}, \mathcal{Y}) \mid \hat{y} = f(x; \phi), x \in D_S\}$. The entropy of the model prediction is defined as $H(\hat{y}) = -\sum_c p(\hat{y}_c) \log p(\hat{y}_c)$, $\hat{y} = f(x; \phi)$, where $c$ denotes the class. Lower entropy indicates that the model assigns high probability to one of the classes, suggesting a high level of certainty or confidence in its prediction, which can be interpreted as the sample being well-aligned or fitting closely with the model's learned distribution. In other words, the model recognizes the sample as being similar to those it was trained on. Thus entropy can be used as an indicator of how closely a sample $x$ aligns with the model distribution $\mathcal{D}_{\phi,S}$. Since the model distribution is approximately the source distribution, selecting (and labeling) low-entropy samples using $f(x; \phi)$ essentially provides an estimate of sampling from the source dataset. Therefore, in place of the inaccessible $D_S$, we can feasibly include the source-like dataset into the ATTA training data at each time step $t$:

$$D_{\phi,S}(t) = \{(x, f(x; \phi)) | x \in U_{te}(t), H(f(x; \phi)) < e_l\}, \tag{6}$$

where $e_l$ is the entropy threshold. The assumption that $D_{\phi,S}(t)$ is an approximation of $D_S$ can be empirically validated, as shown by the numerical results on PACS in Tab. 1. In contrast, high-entropy test samples typically deviate more from the source data, from which we select $D_{te}(t)$ for active labeling. Following the notations in Thm. 1, we are practically minimizing the empirical weighted error of hypothesis $h(t)$ as

$$\hat{\epsilon}'_{\boldsymbol{w}}(h(t)) = \sum_{j=0}^{t} w_j \hat{\epsilon}_j(h(t)) = \frac{w_0}{\lambda_0 N} \sum_{x \in D_{\phi,S}(t)} |h(x,t) - f(x; \phi)| + \sum_{j=1}^{t} \frac{w_j}{\lambda_j N} \sum_{x,y \in D_{te}(j)} |h(x,t) - y|. \tag{7}$$

By substituting $D_S$ with $D_{\phi,S}(t)$ in Thm. 1, the bounds of Thm. 1 continue to hold for the test domains. In the corollary below, we bound the source error for practical ATTA at each time step $t$.

**Corollary 3.** *At time step t, for ATTA data domains $D_{\phi,S}(t), U_{te}(1), U_{te}(2), \cdots, U_{te}(t)$, $S_i$ are unlabeled samples of size $m$ sampled from each of the $t + 1$ domains respectively, and $S_S$ is unlabeled samples of size $m$ sampled from $D_S$. If $\hat{h}(t) \in \mathcal{H}$ minimizes $\hat{\epsilon}'_{\boldsymbol{w}}(h(t))$ while other conditions remain identical to Thm. 1, then*

$$\epsilon_S(\hat{h}(t)) \leq \epsilon_S(h_S^*(t)) + \sum_{i=0}^{t} w_i \left( \hat{d}_{\mathcal{H}\Delta\mathcal{H}}(S_i, S_S) + 4\sqrt{\frac{2d \log(2m) + \log \frac{2}{\delta}}{m}} + 2\gamma_i \right) + 2C,$$

*with probability at least $1 - \delta$, where $C$ follows Thm. 1 and $\gamma_i = \min_{h \in \mathcal{H}}\{\epsilon_i(h(t)) + \epsilon_S(h(t))\}$.*

Further analysis and proofs are in Appx. D and E. The following corollary provides direct theoretical support that our strategy conditionally reduces the error bound on the source domain.

**Corollary 4.** *At time step t, for ATTA data domains $D_{\phi,S}(t), U_{te}(1), U_{te}(2), \cdots, U_{te}(t)$, suppose that $\hat{h}(t) \in \mathcal{H}$ minimizes $\hat{\epsilon}_{\boldsymbol{w}'}(h(t))$ under identical conditions to Thm. 2. Let's denote the source error upper bound with $|\epsilon_S(\hat{h}(t)) - \epsilon_S(h_S^*(t))| \leq EB_S(\boldsymbol{w}, \boldsymbol{\lambda}, N, t)$. Let $\boldsymbol{w}'$ and $\boldsymbol{\lambda}'$ be the weight*

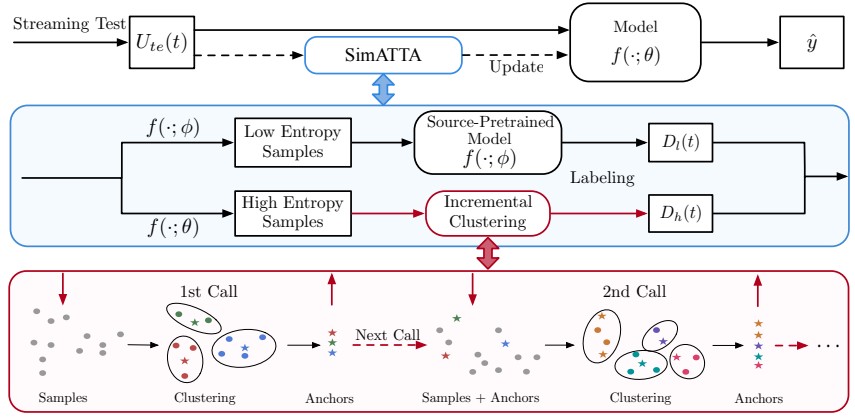

Figure 2: Overview of the SimATTA framework.

*and sample ratio vectors when $D_{\phi,S}(t)$ is not included,* i.e., $w'$ and $\lambda'$ *s.t.* $w'_0 = \lambda'_0 = 0$. *If* $\hat{d}_{\mathcal{H}\Delta\mathcal{H}}(D_S, D_{\phi,S}(t)) < \hat{d}_{\mathcal{H}\Delta\mathcal{H}}(D_S, \bigcup_{i=1}^{t} U_{te}(i))$, *then for any* $\lambda \neq \lambda'$, *there exists* $w$ *s.t.*

$$EB_S(w, \lambda, N, t) < EB_S(w', \lambda', N, t). \tag{8}$$

Corollary 4 validates that the selected low-entropy samples can mitigate the CF problem under the assumption that these samples are source-like, which is also empirically validated in Fig. 1. Note that our strategy employs entropy minimization in a selective manner, aiming to solve CF rather than the main adaptation issue. While many FTTA works use entropy minimization to adapt across domains without guarantees, our use is more theoretically-sound.

## 4    AN ATTA ALGORITHM

Building on our theoretical findings, we introduce a simple yet effective ATTA method, known as SimATTA, that innovatively integrates incremental clustering and selective entropy minimization techniques, as illustrated in Fig. 2. We start with an overview of our methodology, including the learning framework and the comprehensive sample selection strategies. We then proceed to discuss the details of the incremental clustering technique designed for real-time sample selections.

### 4.1    ALGORITHM OVERVIEW

Let $(x, y)$ be a labeled sample and $f(\cdot; \theta)$ be our neural network, where $\hat{y} = f(x; \theta)$ and $\theta$ represents the parameters. We have a model pre-trained on source domains with the pre-trained parameters $\phi$. We initialize model parameters as $\theta(0) = \phi$ and aim to adapt the model $f(\cdot; \theta)$ in real-time. During the test phase, the model continuously predicts labels for streaming-in test data and concurrently gets fine-tuned. We perform sample selection to enable active learning. As discussed in Sec. 3.2, we empirically consider informative high-entropy samples for addressing distribution shifts and source-like low-entropy samples to mitigate CF. As shown in Alg. 1, at each time step $t$, we first partition unlabeled test samples $U_{te}(t)$ into high entropy and low entropy datasets, $U_h(t)$ and $U_l(t)$, using an entropy threshold. The source-pretrained model $f(\cdot; \phi)$ is frozen to predict pseudo labels for low entropy data. We obtain labeled low-entropy data $D_l(t)$ by labeling $U_l(t)$ with $f(\cdot; \phi)$ and combining it with $D_l(t-1)$. In contrast, the selection of high-entropy samples for active labeling is less straightforward. Since the complete test dataset is inaccessible for analyzing the target domain distribution, real-time sample selection is required. We design an incremental clustering sample selection technique to reduce sample redundancy and increase distribution coverage, detailed in Sec. 4.2. The incremental clustering algorithm outputs the labeled test samples $D_h(t)$, also referred to as anchors, given $D_h(t-1)$ and $U_h(t)$. After sample selection, the model undergoes test-time training using the labeled test anchors $D_h(t)$ and pseudo-labeled source-like anchors $D_l(t)$. Following the analyses in Sec. 3.1, the training weights and sample numbers should satisfy $w(t) \approx \lambda(t)$ for $D_h(t)$ and $D_l(t)$ for optimal results. The analyses and results in Sec. 3.2 further indicate that *balancing the source and target ratio is the key to mitigating CF.* However, when source-like samples significantly outnumber test samples, the optimal $w(t)$ for test domains can deviate from $\lambda(t)$ according to Eq. (4).

### 4.2    INCREMENTAL CLUSTERING

We propose incremental clustering, a novel continual clustering technique designed to select informative samples in unsupervised settings under the ATTA framework. The primary goal of this strategy is to store representative samples for distributions seen so far. Intuitively, we apply clusters to cover all seen distributions while adding new clusters to cover newly seen distributions. During this process with new clusters added, old clusters may be merged due to the limit of the cluster budget. Since

---

**Algorithm 1** SIMATTA: A SIMPLE ATTA ALGORITHM

---

**Require:** A fixed source pre-trained model $f(\cdot; \phi)$ and a real-time adapting model $f(\cdot; \theta(t))$ with $\theta(0) = \phi$. Streaming test data $U_{te}(t)$ at time step $t$. Entropy of predictions $H(\hat{y}) = -\sum_c p(\hat{y}_c) \log p(\hat{y}_c)$. Low entropy and high entropy thresholds $e_l$ and $e_h$. The number of cluster centroid budget $NC(t)$ at time step $t$. Centroid increase number $k$. Learning step size $\eta$.

1: **for** $t = 1, \ldots, T$ **do**
2:      Model inference on $U_{te}(t)$ using $f(\cdot; \theta(t-1))$.
3:      $D_l(t) \leftarrow D_l(t-1) \cup \{(x, f(x; \phi)) | x \in U_{te}(t), H(f(x; \phi)) < e_l\}$
4:      $U_h(t) \leftarrow \{x | x \in U_{te}(t), H(f(x; \theta)) > e_h\}$
5:      $D_h(t) \leftarrow D_h(t-1) \cup \{(x, y) | \forall x \in \text{IC}(D_h(t-1), U_h(t), NC(t)), y = \text{Oracle}(x)\}$
6:      $\boldsymbol{\lambda}(t) \leftarrow |D_l(t)| / (|D_l(t)| + |D_h(t)|), |D_h(t)| / (|D_l(t)| + |D_h(t)|)$
7:      $\boldsymbol{w}(t) \leftarrow \text{GetW}(\boldsymbol{\lambda}(t))$              ▷ Generally, $\text{GetW}(\boldsymbol{\lambda}(t)) = \boldsymbol{\lambda}(t)$ is a fair choice.
8:      $\theta(t) \leftarrow \theta(t-1)$
9:      **for** $(x_l, y_l)$ in $D_l$ and $(x_h, y_h)$ in $D_h$ **do**
10:        $\theta(t) \leftarrow \theta(t) - \eta w_0 \nabla \ell_{CE}(f(x_l; \theta(t)), y_l) - \eta(1 - w_0) \nabla \ell_{CE}(f(x_h; \theta(t)), y_h)$
11:      **end for**
12:      $NC(t+1) \leftarrow \text{UpdateCentroidNum}(NC(t))$      ▷ Naive choice: $NC(t+1) \leftarrow NC(t) + k$.
13: **end for**

---

clusters cannot be stored efficiently, we store the representative samples of clusters, named anchors, instead. In this work, we adopt weighted $K$-means (Krishna and Murty, 1999) as our base clustering method due to its popularity and suitability for new setting explorations.

When we apply clustering with new samples, a previously selected anchor should not weigh the same as new samples since the anchor is a representation of a cluster, *i.e.*, a representation of many samples. Instead, the anchor should be considered as a barycenter with a weight of the sum of its cluster's sample weights. For a newly added cluster, its new anchor has the weight of the whole cluster. For clusters containing multiple old anchors, *i.e.*, old clusters, the increased weights are distributed equally among these anchors. These increased weights are contributed by new samples that are close to these old anchors. Intuitively, this process of clustering is analogous to the process of planet formation. Where there are no planets, new planets (anchors) will be formed by the aggregation of the surrounding material (samples). Where there are planets, the matter is absorbed by the surrounding planets. This example is only for better understanding without specific technical meanings.

Specifically, we provide the detailed Alg. 2 for incremental clustering. In each iteration, we apply weighted K-Means for previously selected anchors $D_{\text{anc}}$ and the new streaming-in unlabeled data $U_{\text{new}}$. We first extract all sample features using the model from the previous step $f(\cdot; \theta(t-1))$, and then cluster these weighted features. The initial weights of the new unlabeled samples are 1, while anchors inherit weights from previous iterations. After clustering, clusters including old anchors are old clusters, while clusters only containing new samples are newly formed ones. For each new cluster, we select the centroid-closest sample as the new anchor to store. As shown in line 10 of Alg. 2, for both old and new clusters, we distribute the sample weights in this cluster as its anchors' weights. With incremental clustering, although we can control the number of clusters in each iteration, we cannot control the number of new clusters/new anchors. This indirect control makes the increase of new anchors adaptive to the change of distributions, but it also leads to indirect budget control. Therefore, in experimental studies, we set the budget limit, but the actual anchor budget will not reach this limit. The overall extra storage requirement is $O(\mathcal{B})$ since the number of saved unlabeled samples is proportional to the number of saved labeled samples (anchors).

## 5 EXPERIMENTAL STUDIES

In this study, we aim to validate the effectiveness of our proposed method, as well as explore the various facets of the ATTA setting. Specifically, we design experiments around the following research questions: **RQ1:** Can TTA methods address domain distribution shifts? **RQ2:** Is ATTA as efficient as TTA? **RQ3:** How do the components of SimATTA perform? **RQ4:** Can ATTA perform on par with stronger Active Domain Adaptation (ADA) methods? We compare ATTA with three settings, *TTA* (Tab. 2), *enhanced TTA* (Tab. 3 and 5), and *ADA* (Tab. 4).

**Datasets.** To assess the OOD performance of the TTA methods, we benchmark them using datasets from DomainBed (Gulrajani and Lopez-Paz, 2020) and Hendrycks and Dietterich (2019a). We employ PACS (Li et al., 2017), VLCS (Fang et al., 2013), Office-Home (Venkateswara et al., 2017), and Tiny-ImageNet-C datasets for our evaluations. For each dataset, we designate one domain as

Table 2: **TTA comparisons on PACS and VLCS.** This table includes the two data stream mentioned in the dataset setup and reports performances in accuracy. Results that outperform all TTA baselines are highlighted in **bold** font. N/A denotes the adaptations are not applied on the source domain.

| PACS | Domain-wise data stream | | | | Post-adaptation | | | | Random data stream | | | | Post-adaptation | | | |
|---|---|---|---|---|---|---|---|---|---|---|---|---|---|---|---|---|
| | P | →A→ | →C→ | →S | P | A | C | S | →1→ | →2→ | →3→ | →4→ | P | A | C | S |
| BN w/o adapt | 99.70 | 59.38 | 28.03 | 42.91 | 99.70 | 59.38 | 28.03 | 42.91 | 43.44 | 43.44 | 43.44 | 43.44 | 99.70 | 59.38 | 28.03 | 42.91 |
| BN w/ adapt | 98.74 | 68.07 | 64.85 | 54.57 | 98.74 | 68.07 | 64.85 | 54.57 | 62.50 | 62.50 | 62.50 | 62.50 | 98.74 | 68.07 | 64.85 | 54.57 |
| Tent (steps=1) | N/A | 67.29 | 64.59 | 44.67 | 97.60 | 66.85 | 64.08 | 42.58 | 56.35 | 54.09 | 51.83 | 48.58 | 97.19 | 63.53 | 60.75 | 41.56 |
| Tent (steps=10) | N/A | 67.38 | 57.85 | 20.23 | 62.63 | 34.52 | 40.57 | 13.59 | 47.36 | 31.01 | 22.84 | 20.33 | 50.78 | 23.68 | 20.95 | 19.62 |
| EATA | N/A | 67.04 | 64.72 | 50.27 | 98.62 | 66.50 | 62.46 | 48.18 | 57.31 | 56.06 | 58.17 | 59.78 | 98.62 | 69.63 | 65.70 | 54.26 |
| CoTTA | N/A | 65.48 | 62.12 | 53.17 | 98.62 | 65.48 | 63.10 | 53.78 | 56.06 | 54.33 | 57.16 | 57.42 | 98.62 | 65.97 | 62.97 | 54.62 |
| SAR (steps=1) | N/A | 66.75 | 63.82 | 49.58 | 98.32 | 66.94 | 62.93 | 45.74 | 56.78 | 56.35 | 56.68 | 56.70 | 98.44 | 68.16 | 64.38 | 52.53 |
| SAR (steps=10) | N/A | 69.38 | 68.26 | 49.02 | 96.47 | 62.16 | 56.15 | 54.62 | 53.51 | 51.15 | 51.78 | 45.60 | 94.13 | 56.64 | 56.02 | 36.37 |
| SimATTA ($\mathcal{B} \leq 300$) | N/A | 76.86 | 70.90 | 75.39 | 98.80 | 84.47 | 82.25 | 81.52 | 69.47 | 76.49 | 82.45 | 82.22 | 98.98 | 84.91 | 83.92 | 86.00 |
| SimATTA ($\mathcal{B} \leq 500$) | N/A | 77.93 | 76.02 | 76.30 | 98.62 | 88.33 | 83.49 | 83.74 | 68.46 | 78.22 | 80.91 | 85.49 | 99.16 | 86.67 | 84.77 | 87.71 |

| VLCS | Domain-wise data stream | | | | Post-adaptation | | | | Random data stream | | | | Post-adaptation | | | |
|---|---|---|---|---|---|---|---|---|---|---|---|---|---|---|---|---|
| | C | →L→ | →S→ | →V | C | L | S | V | →1→ | →2→ | →3→ | →4→ | C | L | S | V |
| BN w/o adapt | 100.00 | 33.55 | 41.10 | 49.05 | 100.00 | 33.55 | 41.10 | 49.05 | 41.23 | 41.23 | 41.23 | 41.23 | 100.00 | 33.55 | 41.10 | 49.05 |
| BN w/ adapt | 85.16 | 37.31 | 33.27 | 52.16 | 85.16 | 37.31 | 33.27 | 52.16 | 40.91 | 40.91 | 40.91 | 40.91 | 85.16 | 37.31 | 33.27 | 52.16 |
| Tent (steps=1) | N/A | 38.55 | 34.40 | 53.88 | 84.73 | 43.86 | 33.61 | 53.11 | 44.85 | 44.29 | 47.38 | 44.98 | 85.30 | 43.49 | 37.81 | 53.35 |
| Tent (steps=10) | N/A | 45.41 | 31.44 | 32.32 | 42.54 | 37.65 | 27.79 | 33.12 | 46.13 | 42.31 | 43.51 | 39.48 | 52.01 | 40.32 | 33.64 | 40.37 |
| EATA | N/A | 37.24 | 33.15 | 52.58 | 84.10 | 37.69 | 32.39 | 52.49 | 43.77 | 42.48 | 43.34 | 41.55 | 83.32 | 36.67 | 31.47 | 52.55 |
| CoTTA | N/A | 37.39 | 32.54 | 52.25 | 82.12 | 37.65 | 33.12 | 52.90 | 43.69 | 42.14 | 43.21 | 42.32 | 81.98 | 37.99 | 33.52 | 53.23 |
| SAR (steps=1) | N/A | 36.18 | 34.43 | 52.46 | 83.96 | 39.72 | 36.53 | 52.37 | 43.64 | 43.04 | 44.20 | 41.93 | 85.09 | 40.70 | 36.44 | 53.02 |
| SAR (steps=10) | N/A | 35.32 | 34.10 | 51.66 | 82.12 | 41.49 | 33.94 | 53.08 | 43.56 | 42.05 | 42.53 | 41.16 | 85.09 | 37.58 | 33.12 | 52.01 |
| SimATTA ($\mathcal{B} \leq 300$) | N/A | 62.61 | 65.08 | 74.38 | 99.93 | 69.50 | 66.67 | 77.34 | 62.33 | 69.33 | 73.20 | 71.93 | 99.93 | 69.43 | 72.46 | 80.39 |
| SimATTA ($\mathcal{B} \leq 500$) | N/A | 63.52 | 68.01 | 76.13 | 99.51 | 70.56 | 73.10 | 78.35 | 62.29 | 70.45 | 73.50 | 72.02 | 99.43 | 70.29 | 72.55 | 80.18 |

the source domain and arrange the samples from the other domains to form the test data stream. For DomainBed datasets, we adopt two stream order strategies. The first order uses a domain-wise data stream, *i.e.*, we finish streaming samples from one domain before starting streaming another domain. The second order is random, where we shuffle samples from all target domains and partition them into four splits 1, 2, 3, and 4, as shown in Tab. 2. More dataset details are provided in Appx. G.1.

**Baselines.** For baseline models, we start with the common source-only models, which either utilize pre-calculated batch statistics (BN w/o adapt) or test batch statistics (BN w/ adapt). For comparison with other TTA methods, we consider four state-of-the-art TTA methods: Tent (Wang et al., 2021), EATA (Niu et al., 2022), CoTTA (Wang et al., 2022a), and SAR (Niu et al., 2023). The three of them except Tent provide extra design to avoid CF. To compare with ADA methods, we select algorithms that are partially comparable with our method, *i.e.*, they should be efficient (*e.g.*, uncertainty-based) without the requirements of additional networks. Therefore, we adopt random, entropy (Wang and Shang, 2014), k-means (Krishna and Murty, 1999), and CLUE (Prabhu et al., 2021) for comparisons.

**Settings.** For TTA, we compare with general TTA baselines in streaming adaptation using the two aforementioned data streaming orders, domain-wise and random. We choose P in PACS and C in VLCS as source domains. For domain-wise data stream, we use order A → C → S for PACS and L → S → V for VLCS. We report the real-time adaptation accuracy results for each split of the data stream, as well as the accuracy on each domain after all adaptations through the data stream (under "post-adaptation" columns). Enhanced TTA is built on TTA with access to extra random sample labels. TTA baselines are further fine-tuned with these random samples. To further improve enhanced TTA, we use long-term label storage and larger unlabeled sample pools. To its extreme where the model can access the whole test set samples, the setting becomes similar to ADA, thus we also use ADA methods for comparisons. ADA baselines have access to all samples in the pre-collected target datasets but not source domain data, whereas our method can only access the streaming test data.

## 5.1 The failure of Test-Time Adaptation

The failure of TTA methods on domain distribution shifts is one of the main motivations of the ATTA setting. As shown in Tab. 2, TTA methods cannot consistently outperform even *the simplest baseline "BN w/ adapt"* which uses test time batch statistics to make predictions, evidencing that current TTA methods cannot solve domain distribution shifts (RQ1). Additionally, Tent (step=10) exhibits significant CF issues, where "step=10" indicates 10 test-time training updates, *i.e.*, 10 gradient backpropagation iterations. This failure of TTA methods necessitates the position of ATTA. In contrast, SimATTA, with a budget $\mathcal{B}$ less than 300, outperforms all TTA methods on both source and target domains by substantial margins. Moreover, compared to the source-only baselines, our method improves the target domain performances significantly with negligible source performance loss, showing that ATTA is a more practically effective setting for real-world distribution shifts.

## 5.2 Efficiency & Enhanced TTA Setting Comparisons

To validate the efficiency of ATTA and broaden the dataset choice, we conduct this study on Tiny-ImageNet-C which, though does not focus on domain shifts, is much larger than PACS and VLCS. we

Table 3: **Comparisons with Enhanced TTA on Tiny-ImageNet-C (severity level 5).**

| Tiny-ImageNet-C | Time (sec) | Noise | | | Blur | | | | Weather | | | Digital | | | | |
|---|---|---|---|---|---|---|---|---|---|---|---|---|---|---|---|---|
| | | Gauss. | Shot | Impul. | Defoc. | Glass | Motion | Zoom | Snow | Frost | Fog | Contr. | Elastic | Pixel | JPEG | Avg. |
| Tent (step=1) | 68.83 | 9.32 | 11.97 | 8.86 | 10.43 | 7.00 | 12.20 | 14.34 | 13.58 | 15.46 | 13.55 | 3.99 | 13.31 | 17.79 | 18.61 | 12.17 |
| Tent (step=10) | 426.90 | 0.86 | 0.63 | 0.52 | 0.52 | 0.55 | 0.54 | 0.50 | 0.50 | 0.50 | 0.50 | 0.50 | 0.50 | 0.50 | 0.50 | 0.54 |
| EATA | 93.14 | 3.98 | 3.33 | 2.18 | 4.80 | 2.37 | 11.02 | 11.41 | 14.06 | 15.26 | 9.65 | 1.36 | 9.88 | 14.24 | 12.12 | 8.26 |
| CoTTA | 538.78 | 5.63 | 7.12 | 6.31 | 8.05 | 5.74 | 9.68 | 10.55 | 11.75 | 12.00 | 11.15 | **4.17** | 5.35 | 7.82 | 8.90 | 8.16 |
| SAR (step=1) | 113.76 | 8.90 | 3.11 | 1.67 | 1.55 | 1.47 | 1.35 | 1.19 | 1.03 | 1.04 | 0.93 | 0.83 | 1.00 | 0.74 | 0.77 | 1.83 |
| SAR (step=10) | 774.11 | 2.67 | 3.26 | 2.38 | 1.64 | 1.85 | 2.49 | 3.16 | 3.81 | 2.72 | 3.12 | 0.81 | 3.47 | 4.04 | 1.76 | 2.66 |
| SimATTA (step=10) | 736.28 | **9.68** | **19.40** | **12.14** | **30.28** | **17.03** | **42.36** | **43.10** | **31.96** | **40.08** | **29.24** | 3.21 | **34.56** | **45.24** | **45.74** | **28.86** |

enhance the TTA setting by fine-tuning baselines on randomly selected labeled samples. Specifically, the classifier of ResNet18-BN is pre-adapted to the brightness corruption (source domain) before test-time adapting. SimATTA's label budget is around 4,000, while all other TTA methods have budget 4,500 for randomly selected labeled samples. The data stream order is shown in Tab. 3. Time is measured across all corrupted images in the Noise and Blur noise types, and the values represent the average time cost for adapting 10,000 images. The results clearly evidence the efficiency of ATTA (RQ2), while substantially outperforming all enhanced TTA baselines. Simply accessing labeled samples cannot benefit TTA methods to match ATTA. With 10 training updates (step=10) for each batch, FTTA methods would suffer from severe CF problem. In contrast, ATTA covers a statistically significant distribution, achieving stronger performances with 10 training updates or even more steps till approximate convergences. In fact, longer training on Tent (step=10) leads to worse results (compared to step=1), which further motivates the design of the ATTA setting. The reason for higher absolute time cost in Tab. 3 is due to differences in training steps. In this experiment, SimATTA has a training step of 10, and similar time cost as SAR per step.

Note that if the enhanced TTA setting is further improved to maintain distributions with a balanced CF mitigation strategy and an incremental clustering design, the design approaches ATTA. Specifically, we compare SimATTA with its variants as the ablation study (RQ3) in Appx. I.2.

## 5.3 COMPARISONS TO A STRONGER SETTING: ACTIVE DOMAIN ADAPTATION

In addtion to the above comparisons with (enhanced) TTA, which necessitate the requirement of extra information in the ATTA setting, we compare ATTA with a stronger setting Active Domain Adaptation (ADA) to demonstrate another superiority of ATTA, *i.e.*, weaker requirements for comparable performances (RQ4). ADA baselines are able to choose the global best active samples, while ATTA has to choose samples from a small sample buffer (*e.g.*, a size of 100) and discard the rest. Tab. 4 presents the post-adaptation model performance results. All ADA results are averaged

Table 4: **Comparisons to ADA baselines.** Source domains are denoted as "(S)". Results are average accuracies (with standard deviations).

| PACS | P (S) | A | C | S |
|---|---|---|---|---|
| Random ($\mathcal{B} = 300$) | 96.21 (0.80) | 81.19 (0.48) | 80.75 (1.27) | 84.34 (0.18) |
| Entropy ($\mathcal{B} = 300$) | 96.31 (0.64) | **88.00 (1.46)** | 82.48 (1.71) | 80.55 (1.01) |
| Kmeans ($\mathcal{B} = 300$) | 93.71 (1.50) | 79.31 (4.01) | 79.64 (1.44) | 83.92 (0.65) |
| CLUE ($\mathcal{B} = 300$) | 96.69 (0.17) | 83.97 (0.57) | **84.77 (0.88)** | **86.91 (0.26)** |
| SimATTA ($\mathcal{B} \leq 300$) | **98.89 (0.09)** | 84.69 (0.22) | 83.09 (0.83) | 83.76 (2.24) |

| VLCS | C (S) | L | S | V |
|---|---|---|---|---|
| Random ($\mathcal{B} = 300$) | 96.21 (1.65) | 66.67 (1.70) | 70.72 (0.30) | 72.14 (1.71) |
| Entropy ($\mathcal{B} = 300$) | 97.74 (1.56) | 69.29 (2.26) | 69.25 (4.77) | 75.26 (3.07) |
| Kmeans ($\mathcal{B} = 300$) | 98.61 (0.27) | 67.57 (1.64) | **70.77 (0.01)** | 74.49 (0.97) |
| CLUE ($\mathcal{B} = 300$) | 85.70 (10.09) | 65.29 (1.49) | 69.42 (2.64) | 69.09 (6.05) |
| SimATTA ($\mathcal{B} \leq 300$) | **99.93 (0.00)** | **69.47 (0.03)** | 69.57 (2.90) | **78.87 (1.53)** |

from 3 random runs, while ATTA results are the post-adaptation performances averaged from the two data stream orders. As can be observed, despite the lack of a pre-collected target dataset, SimATTA produces better or competitive results against ADA methods. Moreover, without source data access, SimATTA's design for CF allows it to maintain superior source domain performances over ADA methods. Further experimental studies including the Office-Home dataset are provided in Appx. I.

In conclusion, the significant improvement compared to weaker settings (TTA, enhanced TTA) and the comparable performance with the stronger setting, ADA, rendering ATTA a setting that is as efficient as TTA and as effective as ADA. This implies its potential is worthy of future explorations.

## 6 CONCLUSION AND DISCUSSION

There's no denying that OOD generalization can be extremely challenging without certain information, often relying on various assumptions easily compromised by different circumstances. Thus, it's prudent to seek methods to achieve significant improvements with minimal cost, *e.g.*, DG methods leveraging environment partitions and ATTA methods using budgeted annotations. As justified in our theoretical and experimental studies, ATTA stands as a robust approach to achieve real-time OOD generalization. Although SimATTA sets a strong baseline for ATTA, there's considerable scope for further investigation within the ATTA setting. One potential direction involves developing alternatives to prevent CF in ATTA scenarios. While selective entropy minimization on low-entropy samples has prove to be empirically effective, it relies on the quality of the pre-trained model and training on incorrectly predicted low-entropy samples may reinforce the errors. It might not be cost-effective to expend annotation budgets on low-entropy samples, but correcting them could be a viable alternative solution. We anticipate that our work will spur numerous further explorations in this field.

ACKNOWLEDGMENTS

This work was supported in part by National Science Foundation grant IIS-2006861 and National Institutes of Health grant U01AG070112.

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

# Active Test-Time Adaptation: Foundational Analyses and An Algorithm
# Supplementary Material

## A   BROADER IMPACTS

The field of domain generalization primarily concentrates on enhancing a model's generalization abilities by preparing it thoroughly before deployment. However, it is equally important for deep learning applications to have the capacity for real-time adaptation, as no amount of preparation can account for all possible scenarios. Consequently, domain generalization and test-time adaptation are complementary strategies: the former is more weighty and extensive, while the latter is more agile, lightweight and privacy-friendly.

This work delves into the development of a real-time model adaptation strategy that can be applied to any pre-trained models, including large language models, to enhance their adaptive capabilities. Our research does not involve any human subjects or dataset releases, nor does it raise any ethical concerns. Since this work does not directly tie to specific applications, we do not foresee any immediate negative societal impacts. Nonetheless, we acknowledge that any technological advancement may carry potential risks, and we encourage the continued assessment of the broader impacts of real-time adaptation methodologies in various contexts.

## B   FAQ & DISCUSSIONS

To facilitate the reviewing process, we summarize the answers to the questions that arose during the discussion of an earlier version of this paper.

The major updates of this version are **reorganized theoretical studies, incremental clustering details, experimental reorganization, and additional datasets and settings**. We include more related field comparisons to distinguish different settings. We also cover the position of this paper in literature and the main claims of this paper. Finally, we will frankly acknowledge the limitations of this paper, explain and justify the scope of coverage, and provide possible future directions.

**Q1: What is the relationship between the proposed ATTA protocol and stream based active learning (Saran et al., 2023)?**

**A:** We would like to discuss the difference between our work and the referenced work.

1. **Real-time Training Distinction**: Saran et al. (2023) doesn't operate in real-time capacity. This is evident from their experiments, where their model is trained only after completing a round. In contrast, our work involves training the model post each batch. This positions Saran et al. (2023)'s work as an intrinsic active learning technique, while our approach leans towards TTA methods.

2. **Continual Training Nuance**: Following the point above, Saran et al. (2023) stands out of the scope of continual training. As they mentioned 'each time new data are acquired, the ResNet is reset to the ImageNet pre-trained weights before being updated', Saran et al. (2023) starts afresh with each iteration and is out of scope for CF discussions. Contrarily, our model is continuously trained on varying distributions, compelling us to address the CF issue while preserving advantages derived from various stored distributions.

3. **Comparative Complexity**: Given the aforementioned distinctions, it's evident that our task presents a greater challenge compared to theirs. In addition, we have included comparisons with stronger active learning settings in Sec. 5.3.

**Q2: What are the insights from the theoretically foundational analysis?**

**A:**

1. It sets a well-defined formulation and grounded theoretical framework for the ATTA setting.

2. While entropy minimizations can cause CF, **balancing the learning rate and number of high/low entropy samples is conversely the key solution to both distribution shifts and**

**CF by corresponding benefits**. Though adding low-entropy data is intuitive, it is crucial in that this simple operation can make methods either too conservative or too aggressive without the correct balancing conditions.

3. The studies in Sec. 3.1 **directly present a feasible and guaranteed solution for implementing ATTA** to tackle shifts while avoiding CF. The aligned empirical validations of Sec. 3.2 also instruct the implementation of SimATTA.

**Q3: In test-time adaptation, one important issue is that the number of testing samples in a batch may be small, which means the sample size $m$ will also be very small. May it affect the theorem and make them become very loose?**

**A:** We consider this issue jointly from theoretical and empirical validations.

1. It is true that the theoretical bounds can be loose given a small size of $m$ unlabeled test samples. This situation of the error bound is mathematically ascribed to the quotient between the VC-dimension $d$ of the hypothesis class and $m$. Under the VC-dimension theory, the ResNet18 model we adopt should have $d \gg m$. However, practically we perform fine-tuning on pre-trained models instead of training from scratch, which significantly reduces the scale of parameter update. In this case, an assumption can be established that fine-tuning a model is roughly equivalent to learning a model with a relatively small $d$ (Appx. H). This assumption is potentially underpinned by the empirical alignment of our validation experiments with the theoretical framework (Fig. 1). To this end, experiments indicate that $d$ and $m$ are practically of similar scale for our settings. This prevents our theoretical bounds from being very loose and meaningless in reality.

2. Regarding cases that our assumption does not apply, this issue would appear inevitable, since it is rigorously inherent in the estimation error of our streaming and varying test distributions. The distribution of a test stream can be hardly monitored when only a limited batch is allowed, which we consider as a limitation of TTA settings. Moreover, this issue directly implies the necessity of using a buffer for unlabeled samples. A good practice is to maintain a relatively comparable sample buffer scale.

**Q4: What distribution shifts can ATTA solve?**

**A:** We would like to follow (but not limited to) the work (Zhao et al., 2023b) to discuss the distribution shifts ATTA can solve.

1. As elucidated in Sec. 3.1 and Sec. 5, ATTA can solve domain generalization shifts. Domain generalization shifts include complex shifts on the joint data distribution $P(X, Y)$, given $X$ as the covariates and $Y$ as the label variable. Since $P(X, Y) = P(X)P(Y|X)$, ATTA can handle covariate shift ($P(X)$), label shift ($P(Y)$), and conditional shift ($P(Y|X)$). The shifts on both covariate and conditional distributions can cover the shift on labels, but they (covariate + conditional shifts) are more complicated than pure label shifts, where only the marginal label distribution changes while the conditional distribution remains. Note that the conditional shifts are generally caused by spurious correlations, where the independent causal mechanism assumption (Pearl, 2009) holds or no concept drifts exist.

2. In our framework, the distribution support of $X$ at different time steps can be different, but we don't cover the situation where the support of $Y$ changes, i.e., class-incremental problems.

**Q5: It is unclear how many samples are selected in each minibatch of testing samples. How the total budget is distributed across the whole testing data stream?**

**A:** The number of selected samples for each minibatch is decided jointly by the incremental clustering and the cluster centroid number $NC(t)$. Intuitively, this sample selection is a dynamic process, with $NC(t)$ **restricting the budget and incremental clustering performing sample selection**. For each batch, we increase applicable clustering centroids as a maximum limit, while the exact number of the selected samples is given by the incremental clustering by how many clusters are located in the scope of *new* distributions. *e.g.*, if the incoming batch does not introduce new data distributions, then we select zero samples even with increased $NC(t)$. In contrast, if the incoming batch contains data located in multiple new distributions, the incremental clustering tends to select more samples than the $NC(t)$ limit, thus forcing to merging of multiple previous clusters into one new cluster.

The incremental clustering is detailed in Sec. 4.2, and $NC(t)$ is naively increased by a constant hyper-parameter $k$. Therefore, the budget is **adaptively distributed** according to the data streaming distribution with budgets controlled by $k$, which is also the reason why we compare methods under a budget limit.

**Q6: Could compared methods have access to a few ground-truth labels as well? Making other algorithms be able to use the same amount of ground-truth labels randomly will produce fairer comparisons.**

**A:**

1. The enhanced TTA setting is exactly the setup we provide to produce fairer comparisons. See Tab. 3 and Tab. 5 for comparison results.

2. ATTA also compares to a stronger setting ADA which can access the whole test datasets multiple times.

Table 5: The table demonstrates the comparisons on PACS where all enhanced TTA baselines have 300 budgets to randomly select labeled samples. The training steps of these labeled samples are the same as the original TTA method training steps. For accumulated sample selection, please refer to our ablation studies.

| Method | | Domain-wise data stream | | | | AVG | | | | Random data stream | | | | AVG | | | |
|---|---|---|---|---|---|---|---|---|---|---|---|---|---|---|---|---|---|
| | | P→ | →A→ | →C→ | →S | P | A | C | S | 1 | 2 | 3 | 4 | P | A | C | S |
| Source only | BN w/o adapt | 99.70 | 59.38 | 28.03 | 42.91 | 99.70 | 59.38 | 28.03 | 42.91 | 43.44 | 43.44 | 43.44 | 43.44 | 99.70 | 59.38 | 28.03 | 42.91 |
| | BN w/ adapt | 98.74 | 68.07 | 64.85 | 54.57 | 98.74 | 68.07 | 64.85 | 54.57 | 62.50 | 62.50 | 62.50 | 62.50 | 98.74 | 68.07 | 64.85 | 54.57 |
| TTA | Tent (steps=1) | N/A | 70.07 | 68.43 | 64.42 | 97.72 | 74.17 | 72.61 | 68.92 | 61.20 | 62.36 | 66.59 | 67.32 | 98.14 | 74.37 | 70.26 | 66.07 |
| | Tent (steps=10) | N/A | 76.27 | 63.78 | 49.35 | 59.46 | 38.62 | 48.46 | 55.03 | 56.20 | 53.22 | 52.55 | 55.55 | 58.32 | 47.56 | 60.75 | 58.00 |
| | EATA | N/A | 69.53 | 66.94 | 61.42 | 98.56 | 69.38 | 66.60 | 64.83 | 60.34 | 59.81 | 64.38 | 65.02 | 98.68 | 73.78 | 68.30 | 59.74 |
| | CoTTA | N/A | 66.55 | 63.14 | 59.91 | 90.12 | 61.67 | 66.68 | 67.68 | 57.26 | 57.36 | 63.46 | 65.64 | 92.22 | 71.53 | 70.44 | 62.41 |
| | SAR (steps=1) | N/A | 66.60 | 63.78 | 50.34 | 98.38 | 67.87 | 64.04 | 49.48 | 57.21 | 56.06 | 56.78 | 57.14 | 98.38 | 68.80 | 64.59 | 53.02 |
| | SAR (steps=10) | N/A | 69.09 | 66.55 | 49.07 | 96.23 | 62.50 | 59.34 | 46.53 | 49.76 | 52.74 | 48.51 | 49.06 | 95.39 | 57.13 | 54.61 | 38.76 |
| | Ours ($\mathcal{B} \leq 300$) | N/A | 76.86 | 70.90 | 75.39 | 98.80 | 84.47 | 82.25 | 81.52 | 69.47 | 76.49 | 82.45 | 82.22 | 98.98 | 84.91 | 83.92 | 86.00 |

**Q7: What is the position of ATTA?**

**A:** Comparisons with different settings are challenging. In this work, the design of our experiments (Sec. 5) is to overcome this challenge by comparing both weaker settings and stronger settings. While the significant performance over weaker settings renders the necessity of extra information, the comparable performance with stronger settings provides the potential to relax restricted requirements. Intuitively, ATTA is the most cost-effective option in the consideration of both efficiency and effectiveness. We further provide the following ATTA summary:

ATTA, which incorporates active learning in FTTA, is the light, real-time, source-free, widely applicable setting to achieve high generalization performances for test-time adaptation.

1. **Necessity:** From the causality perspective, new information is necessary (Lin et al., 2022; Pearl, 2009; Peters et al., 2017) to attain generalizable over distribution shifts which are insurmountable within the current TTA framework.

2. **Effectiveness:** Compared to FTTA methods, ATTA produces substantially better performances, on-par with the costly active domain adaptation (ADA) methods as shown in Table 3 in the paper.

3. **Efficiency:** Relative to ADA methods, ATTA possesses superior efficiency, similar to general FTTA methods, as shown in Tab. 3.

4. **Applicability:** ATTA is a model-agnostic setting. (1) Compared to domain generalization methods, ATTA do not require re-training and has the potential to apply to any pre-trained models. One interesting future direction is designing ATTA methods for large language models (LLMs), where re-trainings are extremely expensive and source data may be inaccessible. (2) Compared to FTTA methods, ATTA can protect model parameters from corrupting while learning new distributions by fine-tuning pre-trained models, rendering it more feasible and practical.

In comparison with existing works, ATTA is motivated to mitigate the limitations of previous settings:

1. FTTA: Limited generalization performance.

2. TTT: Not source-free; limited generalization performance.

3. ADA & domain adaptation/generalization: Expensive re-trainings; limited applicability to pre-trained models.

4. Online active learning: It does not maintain and protect adaptation performances for multiple distributions in one model and does not consider the CF problem.

**Q8: What is the potential practical utility of ATTA?**

**A:**

1. Empirically, our method can generally finish a round of sample selection/training of 100 frames in 5s, *i.e.*, 20 frames per sec, which is more than enough to handle multiple practical situations. Experiments on time complexity are provided in Tab. 3, where SimATTA has comparable time efficiency.

2. As a case analysis, the autopilot system (Hu et al., 2023; Chen et al., 2022a) presents an application scenario requiring high-speed low-latency adaptations, while these adaptations are largely underexplored. When entering an unknown environment, e.g., a construction section, a system of ATTA setting can require the driver to take over the wheel. During the period of manual operation when the driver is handling the wheel, steering signals are generated, and the in-car system quickly adaptations. The system doesn't need to record 60 frames per second, since only the key steering operations and the corresponding dash cam frames are necessary, which can be handled by ATTA algorithms processing at 20 frames per sec. In this case, the human annotations are necessary and indirect. ATTA makes use of this information and adapts in the short term instead of collecting videos and having a long-round fine-tuning (Schafer et al., 2018).

3. In addition, many scenarios applicable for ATTA are less speed-demanding than the case above. One example is a personalized chatbot that subtly prompts and gathers user labels during user interaction. In a home decoration setting, applications can request that users scan a few crucial areas to ensure effective adaptation. Social robots (Mavrogiannis et al., 2023), *e.g.*, vacuum robots, often require users to label critical obstacles they've encountered.

4. Compared with ADA, ATTA stands out as the tailored solution for the above scenarios. It does not require intensive retraining or server-dependent fine-tuning, offering both speed and computational efficiency. Meanwhile, akin to other TTA methods, ATTA also ensures user privacy. While it might marginally exceed the cost of standard TTA methods, the superior generalization ability makes it a compelling choice and justifies the additional expense.

**Q9: What can be covered by this paper?**

**A:** This paper endeavors to establish the foundational framework for a novel setting referred to as ATTA. We target (1) positioning the ATTA setting, (2) solving the two major and basic challenges of ATTA, *i.e.*, the mitigation of distribution shifts and the avoidance of catastrophic forgetting (CF). We achieve the first goal by building the problem formulation and analyses, and further providing extensive qualitative and well-organized experimental comparisons with TTA, enhanced TTA, and ADA settings. These efforts position ATTA as the most cost-effective option between TTA and ADA, where ATTA inherits the efficiency of TTA and the effectiveness of ADA. With our theoretical analyses and the consistent algorithm design, we validate the success of our second goal through significant empirical performances.

**Q10: What are not covered by this paper?**

**A:** Constructing a new setting involves multifaceted complexities. Although there are various potential applications discussed above including scaling this setting up for large models and datasets, we cannot cover them in this single piece of work. There are three main reasons. First, the topics covered by a single paper are limited. Formally establishing ATTA setting and addressing its major challenges of ATTA takes precedence over exploring practical applications. Secondly, given the interrelations between ATTA and other settings, our experimental investigations are predominantly comparative, utilizing the most representative datasets from TTA and domain adaptation to showcase persuasive results. Thirdly, many practical applications necessitate task-specific configurations, rendering them unsuitable for establishing a universal learning setting. While the current focus is on laying down the foundational aspects of ATTA, the exploration of more specialized applications remains a prospective avenue for future work in the ATTA domain.

## C    RELATED WORKS

The development of deep learning witnesses various applications (He et al., 2016; Gui et al., 2020). To tackle OOD problem, various domain generalization works emerge (Krueger et al., 2021; Sagawa et al., 2019).

### C.1    UNSUPERVISED DOMAIN ADAPTATION

Unsupervised Domain Adaptation (UDA) (Pan et al., 2010; Patel et al., 2015; Wilson and Cook, 2020; Wang and Deng, 2018) aims at mitigating distribution shifts between a source domain and a target domain, given labeled source domain samples and unlabeled target samples. UDA methods generally rely on feature alignment techniques to eliminate distribution shifts by aligning feature distributions between source and target domains. Typical feature alignment techniques include discrepancy minimization (Long et al., 2015; Sun and Saenko, 2016; Kang et al., 2019) and adversarial training (Ganin and Lempitsky, 2015; Tsai et al., 2018; Ajakan et al., 2014; Ganin et al., 2016; Tzeng et al., 2015; 2017). Nevertheless, alignments are normally not guaranteed to be correct, leading to the alignment distortion problem as noted by Ning et al. (2021).

Source-free Unsupervised Domain Adaptation (SFUDA) (Fang et al., 2022; Liu et al., 2021b) algorithms aim to adapt a pre-trained model to unlabeled target domain samples without access to source samples. Based on whether the algorithm can access model parameters, these algorithms are categorized into white-box and black-box methods. White-box SFUDA typically considers data recovery (generation) and fine-tuning methods. The former focuses on recovering source-like data (Ding et al., 2022; Yao et al., 2021), e.g., training a Generative Adversarial Network (GAN) (Kurmi et al., 2021; Li et al., 2020), while the latter employs various techniques (Mao et al., 2021), such as knowledge distillation (Chen et al., 2022b; Liu and Yuan, 2022; Yang et al., 2021b; Yu et al., 2022), statistics-based domain alignment (Ishii and Sugiyama, 2021; Liu et al., 2021a; Fan et al., 2022; Eastwood et al., 2021), contrastive learning (Huang et al., 2021; Wang et al., 2022b), and uncertainty-based adaptation (Gawlikowski et al., 2021; Fleuret et al., 2021; Chen et al., 2021; Li et al., 2021b). Black-box SFUDA cannot access model parameters and often relies on self-supervised knowledge distillation (Liang et al., 2022; 2021), pseudo-label denoising (Zhang et al., 2021; Yang et al., 2022), or generative distribution alignment (Yeh et al., 2021; Yang et al., 2021a).

### C.2    TEST-TIME ADAPTATION

Test-time Adaptation (TTA), especially Fully Test-time Adaptation (FTTA) algorithms (Wang et al., 2021; Iwasawa and Matsuo, 2021; Karani et al., 2021; Nado et al., 2020; Schneider et al., 2020; Wang et al., 2022a; Zhao et al., 2023a; Niu et al., 2022; Zhang et al., 2022a; Niu et al., 2023; You et al., 2021; Zhang et al., 2022b), can be considered as realistic and lightweight methods for domain adaptation. Built upon black-box SFUDA, FTTA algorithms eliminate the requirement of a pre-collected target dataset and the corresponding training phase. Instead, they can only access an unlabeled data stream and apply real-time adaptation and training. In addition to FTTA, Test-time Training (TTT) (Sun et al., 2020; Liu et al., 2021c) often relies on appending the original network with a self-supervised task. TTT methods require retraining on the source dataset to transfer information through the self-supervised task. Although they do not access the source dataset during the test-time adaptation phase, TTT algorithms are not off-the-shelf source-free methods. TTA is a promising and critical direction for real-world applications, but current entropy minimization-based methods can be primarily considered as feature calibrations that require high-quality pseudo-labels. This requirement, however, can be easily violated under larger distribution shifts.

Current TTA algorithms, inheriting UDA drawbacks, cannot promise good feature calibration results, which can be detrimental in real-world deployments. For instance, entropy minimization on wrongly predicted target domain samples with relatively low entropy can only exacerbate spurious correlations (Chen et al., 2020). Without extra information, this problem may be analogous to applying causal inference without intervened distributions, which is intrinsically unsolvable (Peters et al., 2016; Pearl, 2009). This paper aims to mitigate this issue with minimal labeled target domain samples. To minimize the cost, we tailor active learning techniques for TTA settings.

It is worth noting that a recent work AdaNPC (Zhang et al., 2023) is essentially a domain generalization method with a TTA phase attached, while our ATTA is built based on the FTTA setting. Specifically, Current FTTA methods and our work cannot access the source domain. In contrast,

AdaNPC accesses source data to build its memory bank, circumventing the catastrophic forgetting problem. Furthermore, AdaNPC requires multiple source domains and training before performing TTA. Thus AdaNPC uses additional information on domain labels and retraining resources for its memory bank, undermining the merits of FTTA. Regarding theoretical bounds, their target domain is bounded by source domain error and model estimations (in big-$O$ expression), while we consider active sample learning and time variables for varying test distributions.

### C.3 CONTINUAL DOMAIN ADAPTATION

Many domain adaptation methods focus on improving target domain performance, neglecting the performance on the source domain, which leads to the CF problem (Kemker et al., 2018; Kirkpatrick et al., 2017; Li and Hoiem, 2017; Lopez-Paz and Ranzato, 2017; De Lange et al., 2021; Wang et al., 2022a; Niu et al., 2022). This issue arises when a neural network, after being trained on a sequence of domains, experiences a significant degradation in its performance on previously learned domains as it continues to learn new domains. Continual learning, also known as lifelong learning, addresses this problem. Recent continual domain adaptation methods have made significant progress by employing gradient regularization, random parameter restoration, buffer sample mixture, and more. Although the CF problem is proposed in the continual learning field, it can occur in any source-free OOD settings since the degradation caused by CF is attributed to the network's parameters being updated to optimize performance on new domains, which may interfere with the representations learned for previous domains.

### C.4 ACTIVE DOMAIN ADAPTATION

Active Domain Adaptation (ADA) (Prabhu et al., 2021; Ning et al., 2021; Su et al., 2020; Ma et al., 2021; Xie et al., 2022) extends semi-supervised domain adaptation with active learning strategies (Cohn et al., 1996; Settles, 2009), aiming to maximize target domain performance with a limited annotation budget. Therefore, the key challenge of active learning algorithms is selecting the most informative unlabeled data in target domains (Kapoor et al., 2007). Sample selection strategies are often based on uncertainty (Lewis and Catlett, 1994; Scheffer et al., 2001), diversity (Jain and Grauman, 2016; Hoi et al., 2009), representativeness (Xu et al., 2003), expected error minimization (Vijaya-narasimhan and Kapoor, 2010), etc. Among these methods, uncertainty and diversity-based methods are simple and computationally efficient, making them the most suitable choices to tailor for TTA settings.

Adapting these strategies is non-trivial because, compared to typical active domain adaptation, our proposed Active Test-time Adaptation (ATTA) setting does not provide access to source data, model parameters, or pre-collected target samples. This requirement demands that our active sample selection algorithm select samples for annotation during data streaming. Consequently, this active sampling selection process is non-regrettable, *i.e.*, we can only meet every sample once in a short period. To avoid possible confusion, compared to the recent Source-free Active Domain Adaptation (SFADA) method SALAD (Kothandaraman et al., 2023), we do not require access to model parameter gradients, training additional neural networks, or pre-collected target datasets. Therefore, our ATTA setting is quite different, much lighter, and more realistic than ADA and SFADA.

### C.5 ACTIVE ONLINE LEARNING

The most related branch of active online learning (AOL) (Cacciarelli and Kulahci, 2023) is active online learning on drifting data stream (Zhou et al., 2021; Baier et al., 2021; Li et al., 2021a). Generally, these methods include two components, namely, detection and adaptation. Compared with ATTA, there are several distinctions. First, this line of studies largely focuses on the distribution shift detection problem, while ATTA focuses on multi-domain adaptations. Second, AOL on drifting data stream aims to detect and adapt to one current distribution in the stream, without considering preserving the adaptation abilities of multiple past distributions by maintaining and fine-tuning the original pre-trained models. In contrast, ATTA's goal is to achieve the OOD generalization optimums adaptable across multiple source and target distributions, leading to the consideration of CF problems. Third, while AOL requires one-by-one data input and discard, ATTA maintains a buffer for incoming data before selection decisions. This is because ATTA targets maintaining the original model without corrupting and replacing it, such that making statistically meaningful and high-quality decisions is

critical for ATTA. In contrast, AOL allows resetting and retraining new models, whose target is more lean to cost saving and one-by-one manner.

# D  FURTHER THEORETICAL STUDIES

In this section, we refine the theoretical studies with supplement analysis and further results.

We use the $\mathcal{H}$-divergence and $\mathcal{H}\Delta\mathcal{H}$-distance definitions following (Ben-David et al., 2010).

**Definition 2** ($\mathcal{H}$-divergence). For a function class $\mathcal{H}$ and two distributions $\mathcal{D}_1$ and $\mathcal{D}_2$ over a domain $\mathcal{X}$, the $\mathcal{H}$-divergence between $\mathcal{D}_1$ and $\mathcal{D}_2$ is defined as

$$d_{\mathcal{H}}(\mathcal{D}_1, \mathcal{D}_2) = \sup_{h \in \mathcal{H}} |P_{x \sim \mathcal{D}_1}[h(x) = 1] - P_{x \sim \mathcal{D}_2}[h(x) = 1]|.$$

The $\mathcal{H}\Delta\mathcal{H}$-distance is defined base on $\mathcal{H}$-divergence. We use the $\mathcal{H}\Delta\mathcal{H}$-distance definition following (Ben-David et al., 2010).

**Definition 3** ($\mathcal{H}\Delta\mathcal{H}$-distance). For two distributions $\mathcal{D}_1$ and $\mathcal{D}_2$ over a domain $\mathcal{X}$ and a hypothesis class $\mathcal{H}$, the $\mathcal{H}\Delta\mathcal{H}$-distance between $\mathcal{D}_1$ and $\mathcal{D}_2$ w.r.t. $\mathcal{H}$ is defined as

$$d_{\mathcal{H}\Delta\mathcal{H}}(\mathcal{D}_1, \mathcal{D}_2) = \sup_{h,h' \in \mathcal{H}} P_{x \sim \mathcal{D}_1}[h(x) \neq h'(x)] + P_{x \sim \mathcal{D}_2}[h(x) \neq h'(x)]. \tag{9}$$

The $\mathcal{H}\Delta\mathcal{H}$-distance essentially provides a measure to quantify the distribution shift between two distributions. It measures the maximum difference of the disagreement between two hypotheses in $\mathcal{H}$ for two distributions, providing a metrics to quantify the distribution shift between $\mathcal{D}_1$ and $\mathcal{D}_2$. $\mathcal{H}$-divergence and $\mathcal{H}\Delta\mathcal{H}$-distance have the advantage that they can be applied between datasets, *i.e.*, estimated from finite samples. Specifically, let $S_1$, $S_2$ be unlabeled samples of size $m$ sampled from $\mathcal{D}_1$ and $\mathcal{D}_2$; then we have estimated $\mathcal{H}\Delta\mathcal{H}$-distance $\hat{d}_{\mathcal{H}}(S_1, S_2)$. This estimation can be bounded based on Theorem 3.4 of Kifer et al. (2004), which we state here for completeness.

**Theorem 5.** *Let $\mathcal{A}$ be a collection of subsets of some domain measure space, and assume that the VC-dimension is some finite $d$. Let $P_1$ and $P_2$ be probability distributions over that domain and $S_1, S_2$ finite samples of sizes $m_1, m_2$ drawn i.i.d. according $P_1, P_2$ respectively. Then*

$$P_{m_1+m_2} \left[ |\phi_{\mathcal{A}}(S_1, S_2) - \phi_{\mathcal{A}}(P_1, P_2)| > \epsilon \right] \quad \leq (2m)^d e^{-m_1 \epsilon^2/16} + (2m)^d e^{-m_2 \epsilon^2/16}, \tag{10}$$

*where $P_{m_1+m_2}$ is the $m_1 + m_2$'th power of $P$ - the probability that $P$ induces over the choice of samples.*

Theorem 5 bounds the probability for relativized discrepancy, and its applications in below lemmas and Theorem 1 help us bound the quantified distribution shifts between domains. The probability, according to a distribution $\mathcal{D}$, that an estimated hypothesis $h$ disagrees with the true labeling function $g : \mathcal{X} \to \{0, 1\}$ is defined as $\epsilon(h(t), g) = \mathbb{E}_{(x) \sim \mathcal{D}}[|h(x, t) - g(x)|]$, which we also refer to as the error or risk $\epsilon(h(t))$. While the source domain dataset is inaccessible under ATTA settings, we consider the existence of the source dataset $D_S$ for the purpose of accurate theoretical analysis. Thus, we initialize $D_{tr}(0)$ as $D_S$, *i.e.*, $D_{tr}(0) = D_S$. For every time step $t$, the test and training data can be expressed as

$$U_{te}(t) \text{ and } D_{tr}(t) = D_S \cup D_{te}(1) \cup D_{te}(2) \cup \cdots \cup D_{te}(t). \tag{11}$$

We use $N$ to denote the total number of samples in $D_{tr}(t)$ and $\boldsymbol{\lambda} = (\lambda_0, \lambda_1, \cdots, \lambda_t)$ to represent the ratio of sample numbers in each component subset. In particular, we have

$$\frac{|D_S|}{|D_{tr}(t)|} = \lambda_0, \frac{|D_{te}(1)|}{|D_{tr}(t)|} = \lambda_1, \cdots, \frac{|D_{te}(t)|}{|D_{tr}(t)|} = \lambda_t, \tag{12}$$

where $\sum_{i=0}^{t} \lambda_i = 1$. Therefore, at time step $t$, the model has been trained on labeled data $D_{tr}(t)$, which contains $t + 1$ components consisting of a combination of data from the source domain and multiple test-time domains. For each domain the model encounters, $D_S, U_{te}(1), U_{te}(2), \cdots, U_{te}(t)$, let $\epsilon_j(h(t))$ denote the error of hypothesis $h$ at time $t$ on the $j$th domain. Specifically, $\epsilon_0(h(t)) = \epsilon_S(h(t))$ represents the error of $h(t)$ on the source data $D_S$, and $\epsilon_j(h(t))$ for $j \geq 1$ denotes the error of $h(t)$ on test data $U_{te}(j)$. Our optimization minimizes a convex combination of training error over the labeled samples from all domains. Formally, given the vector $\boldsymbol{w} = (w_0, w_1, \cdots, w_t)$ of domain error

weights with $\sum_{j=0}^{t} w_j = 1$ and the sample number from each component $N_j = \lambda_j N$, we minimize the empirical weighted error of $h(t)$ as

$$\hat{\epsilon}_{\boldsymbol{w}}(h(t)) = \sum_{j=0}^{t} w_j \hat{\epsilon}_j(h(t)) = \sum_{j=0}^{t} \frac{w_j}{N_j} \sum_{N_j} |h(x,t) - g(x)|. \tag{13}$$

Note that $\boldsymbol{w}, \boldsymbol{\lambda}$ and $N$ are also functions of $t$, which we omit for simplicity.

We now establish two lemmas as the preliminary for Theorem 1. In the following lemma, we bound the difference between the weighted error $\epsilon_{\boldsymbol{w}}(h(t))$ and the domain error $\epsilon_j(h(t))$.

**Lemma 6.** *Let $\mathcal{H}$ be a hypothesis space of VC-dimension $d$. At time step $t$, let the ATTA data domains be $D_S, U_{te}(1), U_{te}(2), \cdots, U_{te}(t)$, and $S_i$ be unlabeled samples of size $m$ sampled from each of the $t+1$ domains respectively. Then for any $\delta \in (0,1)$, for every $h \in \mathcal{H}$ minimizing $\epsilon_{\boldsymbol{w}}(h(t))$ on $D_{tr}(t)$, we have*

$$|\epsilon_{\boldsymbol{w}}(h(t)) - \epsilon_j(h(t))| \leq \sum_{i=0, i \neq j}^{t} w_i \left( \frac{1}{2} \hat{d}_{\mathcal{H}\Delta\mathcal{H}}(S_i, S_j) + 2\sqrt{\frac{2d \log(2m) + \log \frac{2}{\delta}}{m}} + \gamma_i \right),$$

*with probability of at least $1 - \delta$, where $\gamma_i = \min_{h \in \mathcal{H}} \{\epsilon_i(h(t)) + \epsilon_j(h(t))\}$.*

In the following lemma, we provide an upper bound on the difference between the true and empirical weighted errors $\epsilon_{\boldsymbol{w}}(h(t))$ and $\hat{\epsilon}_{\boldsymbol{w}}(h(t))$.

**Lemma 7.** *Let $H$ be a hypothesis class. For $D_{tr}(t) = D_S \cup D_{te}(1) \cup \cdots \cup D_{te}(t)$ at time $t$, if the total number of samples in $D_{tr}(t)$ is $N$, and the ratio of sample numbers in each component is $\lambda_j$, then for any $\delta \in (0,1)$ and $h \in H$, with probability of at least $1 - \delta$, we have*

$$P[|\epsilon_{\boldsymbol{w}}(h(t)) - \hat{\epsilon}_{\boldsymbol{w}}(h(t))| \geq \epsilon] \leq 2 \exp\left( -2N\epsilon^2 / (\sum_{j=0}^{t} \frac{w_j^2}{\lambda_j}) \right).$$

Thus, as $w_j$ deviates from $\lambda_j$, the feasible approximation $\hat{\epsilon}_{\boldsymbol{w}}(h(t))$ with a finite number of labeled samples becomes less reliable. The proofs for both lemmas are provided in Appx. E. Building upon the two preceding lemmas, we proceed to derive bounds on the domain errors under the ATTA setting when minimizing the empirical weighted error using the hypothesis $h$ at time $t$.

Lemma 6 bounds the difference between the weighted error $\epsilon_{\boldsymbol{w}}(h(t))$ and the domain error $\epsilon_j(h(t))$, which is majorly influenced by the estimated $\mathcal{H}\Delta\mathcal{H}$-distance and the quality of discrepancy estimation. During the ATTA process, the streaming test data can form multiple domains and distributions. However, if we consider all data during the test phase as a single test domain, *i.e.*, $\bigcup_{i=1}^{t} U_{te}(i)$, we can simplify Lemma 6 to obtain an upper bound for the test error $\epsilon_T$ as

$$|\epsilon_{\boldsymbol{w}}(h(t)) - \epsilon_T(h(t))| \leq w_0 \left( \frac{1}{2} \hat{d}_{\mathcal{H}\Delta\mathcal{H}}(S_0, S_T) + 2\sqrt{\frac{2d \log(2m) + \log \frac{2}{\delta}}{m}} + \gamma \right), \tag{14}$$

where $\gamma = \min_{h \in \mathcal{H}} \{\epsilon_0(h(t)) + \epsilon_T(h(t))\}$, and $S_T$ is sampled from $\bigcup_{i=1}^{t} U_{te}(i)$. To understand Lamma 7, we need to understand Hoeffding's Inequality, which we state below as a Proposition for completeness.

**Proposition 8** (Hoeffding's Inequality). *Let $X$ be a set, $D_1, \ldots, D_t$ be probability distributions on $X$, and $f_1, \ldots, f_t$ be real-valued functions on $X$ such that $f_i : X \to [a_i, b_i]$ for $i = 1, \ldots, t$. Then for any $\epsilon > 0$,*

$$\mathbb{P}\left( \left| \frac{1}{t} \sum_{i=1}^{t} f_i(x) - \frac{1}{t} \sum_{i=1}^{t} \mathbb{E}_{x \sim D_i}[f_i(x)] \right| \geq \epsilon \right) \leq 2 \exp\left( -\frac{2t^2\epsilon^2}{\sum_{i=1}^{t} (b_i - a_i)^2} \right) \tag{15}$$

*where $\mathbb{E}[f_i(x)]$ is the expected value of $f_i(x)$.*

Lamma 7 provides an upper bound on the difference between the true and empirical weighted errors $\epsilon_{\boldsymbol{w}}(h(t))$ and $\hat{\epsilon}_{\boldsymbol{w}}(h(t))$. Thus, as $w_j$ deviates from $\lambda_j$, the feasible approximation $\hat{\epsilon}_{\boldsymbol{w}}(h(t))$ with a finite number of labeled samples becomes less reliable. Building upon the two preceding lemmas, we proceed to derive bounds on the domain errors under the ATTA setting when minimizing the empirical weighted error using the hypothesis $h$ at time $t$. Theorem 1 essentially bounds the performance of ATTA on the source and each test domains. The adaptation performance on a test domain is majorly

bounded by the composition of (labeled) training data, estimated distribution shift, and ideal joint hypothesis performance, which correspond to $C$, $\hat{d}_{\mathcal{H}\Delta\mathcal{H}}(S_i, S_j)$, and $\gamma_i$, respectively. The ideal joint hypothesis error $\gamma_i$ gauges the inherent adaptability between domains.

If we consider the multiple data distributions during the test phase as a single test domain, $i.e.$, $\bigcup_{i=1}^{t} U_{te}(i)$, Theorem 1 can be reduced into bounds for the source domain error $\epsilon_S$ and test domain error $\epsilon_T$. With the optimal test/source hypothesis $h_T^*(t) = \arg\min_{h \in \mathcal{H}} \epsilon_T(h(t))$ and $h_S^*(t) = \arg\min_{h \in \mathcal{H}} \epsilon_S(h(t))$,

$$|\epsilon_T(\hat{h}(t)) - \epsilon_T(h_T^*(t))| \leq w_0 A + \sqrt{\frac{w_0^2}{\lambda_0} + \frac{(1-w_0)^2}{1-\lambda_0}} B, \qquad (16a)$$

$$|\epsilon_S(\hat{h}(t)) - \epsilon_S(h_S^*(t))| \leq (1-w_0) A + \sqrt{\frac{w_0^2}{\lambda_0} + \frac{(1-w_0)^2}{1-\lambda_0}} B, \qquad (16b)$$

where the distribution divergence term $A = \hat{d}_{\mathcal{H}\Delta\mathcal{H}}(S_0, S_T) + 4\sqrt{\frac{2d\log(2m) + \log\frac{2}{\delta}}{m}} + 2\gamma$, the empirical gap term $B = 2\sqrt{\frac{d\log(2N) - \log(\delta)}{2N}}$, $S_T$ is sampled from $\bigcup_{i=1}^{t} U_{te}(i)$, and $\gamma = \min_{h \in \mathcal{H}}\{\epsilon_0(h(t)) + \epsilon_T(h(t))\}$. Our learning bounds demonstrates the trade-off between the small amount of budgeted test-time data and the large amount of less relevant source data. Next, we provide an approximation of the condition necessary to achieve optimal adaptation performance, which is calculable from finite samples and can be readily applied in practical ATTA scenarios. Following Eq. (16.a), with approximately $B = c_1\sqrt{d/N}$, the optimal value $w_0^*$ to tighten the test error bound is a function of $\lambda_0$ and $A$:

$$w_0^* = \lambda_0 - \sqrt{\frac{A^2 N}{c_1^2 d - A^2 N \lambda_0 (1-\lambda_0)}}, \quad for \quad \lambda_0 \geq 1 - \frac{d}{A^2 N}, \qquad (17)$$

where $c_1$ is a constant. Note that $\lambda_0 \geq 1 - \frac{d}{A^2 N}$ should be the satisfied condition in practical ATTA settings, where the budget is not sufficiently big while the source data amount is relatively large. When the budget is sufficiently large or the source data amount is not sufficiently large compared to the distribution shift $A$, the optimal $w_0^*$ for the test error bound is $w_0^* = 0$, $i.e.$, using no source data since possible error reduction from the data addition is always less than the error increase caused by large divergence between the source data and the test data.

Theorem 2 offers a direct theoretical guarantee that ATTA reduces the error bound on test domains in comparison to TTA without the integration of active learning. Following Theorem 1, when no active learning is included during TTA, $i.e.$, $w_0 = \lambda_0 = 1$, the upper bound $w_0 A + \sqrt{\frac{w_0^2}{\lambda_0} + \frac{(1-w_0)^2}{1-\lambda_0}} B \geq A + B$; when enabling ATTA, with $w_0 = \lambda_0 \neq 1$, we can easily achieve an upper bound $w_0 A + B < A + B$. Therefore, the incorporation of labeled test instances in ATTA theoretically enhances the overall performance across test domains, substantiating the significance of the ATTA setting in addressing distribution shifts.

Entropy quantifies the amount of information contained in a probability distribution. In the context of a classification model, lower entropy indicates that the model assigns high probability to one of the classes, suggesting a high level of certainty or confidence in its prediction. When a model assigns low entropy to a sample, this high confidence can be interpreted as the sample being well-aligned or fitting closely with the model's learned distribution. In other words, the model "recognizes" the sample as being similar to those it was trained on, hence the high confidence in its prediction. While entropy is not a direct measure of distributional distance, it can be used as an indicator of how closely a sample aligns with the model's learned distribution. This interpretation is more about model confidence and the implied proximity rather than a strict mathematical measure of distributional distance. The pre-trained model is well-trained on abundant source domain data, and thus the model distribution is approximately the source distribution. Selecting low-entropy samples using essentially provides an estimate of sampling from the source dataset. Thus, $D_{\phi,S}(t)$, based on well-aligned with the model's learned distribution is an approximation of $D_S$.

When we consider the CF problem and feasibly include the source-like dataset $D_{\phi,S}(t)$ into the ATTA training data in place of the inaccessible $D_S$ in Eq. (11), we can also derive bounds on the domain errors under this practical ATTA setting when minimizing the empirical weighted error $\epsilon'_w(h(t))$ using the hypothesis $h$ at time $t$, similar to Theorem 1. Let $H$ be a hypothesis class of VC-dimension $d$. At time step $t$, for ATTA data domains $D_{\phi,S}(t), U_{te}(1), U_{te}(2), \cdots, U_{te}(t)$, $S_i$ are unlabeled samples of size $m$ sampled from each of the $t+1$ domains respectively. The total number of samples in $D_{tr}(t)$ is

$N$ and the ratio of sample numbers in each component is $\lambda_i$. If $\hat{h}(t) \in \mathcal{H}$ minimizes the empirical weighted error $\hat{\epsilon}'_{\boldsymbol{w}}(h(t))$ with the weight vector $\boldsymbol{w}$ on $D_{tr}(t)$, and $h_j^*(t) = \arg\min_{h \in \mathcal{H}} \epsilon_j(h(t))$ is the optimal hypothesis on the $j$th domain, then for any $\delta \in (0,1)$, we have

$$\epsilon_j(\hat{h}(t)) \leq \epsilon_j(h_j^*(t)) + 2 \sum_{i=0, i \neq j}^{t} w_i \left( \frac{1}{2} \hat{d}_{\mathcal{H}\Delta\mathcal{H}}(S_i, S_j) + 2\sqrt{\frac{2d \log(2m) + \log \frac{2}{\delta}}{m}} + \gamma_i \right) + 2C$$

with probability of at least $1 - \delta$, where $C = \sqrt{\left( \sum_{i=0}^{t} \frac{w_i^2}{\lambda_i} \right) \left( \frac{d \log(2N) - \log(\delta)}{2N} \right)}$ and $\gamma_i = \min_{h \in \mathcal{H}} \{\epsilon_i(h(t)) + \epsilon_j(h(t))\}$. Other derived results following Theorem 1 also apply for this practical ATTA setting. Further empirical validations for our theoretical results are provided in Appx. H.

## E    PROOFS

This section presents comprehensive proofs for all the lemmas, theorems, and corollaries mentioned in this paper, along with the derivation of key intermediate results.

**Lemma 6.** *Let $\mathcal{H}$ be a hypothesis space of VC-dimension $d$. At time step $t$, let the ATTA data domains be $D_S, U_{te}(1), U_{te}(2), \cdots, U_{te}(t)$, and $S_i$ be unlabeled samples of size $m$ sampled from each of the $t+1$ domains respectively. Then for any $\delta \in (0,1)$, for every $h \in \mathcal{H}$ minimizing $\epsilon_{\boldsymbol{w}}(h(t))$ on $D_{tr}(t)$, we have*

$$|\epsilon_{\boldsymbol{w}}(h(t)) - \epsilon_j(h(t))| \leq \sum_{i=0, i \neq j}^{t} w_i \left( \frac{1}{2} \hat{d}_{\mathcal{H}\Delta\mathcal{H}}(S_i, S_j) + 2\sqrt{\frac{2d \log(2m) + \log \frac{2}{\delta}}{m}} + \gamma_i \right),$$

*with probability of at least $1 - \delta$, where $\gamma_i = \min_{h \in \mathcal{H}} \{\epsilon_i(h(t)) + \epsilon_j(h(t))\}$.*

*Proof.* First we prove that given unlabeled samples of size $m$ $S_1$, $S_2$ sampled from two distributions $\mathcal{D}_1$ and $\mathcal{D}_2$, we have

$$d_{\mathcal{H}\Delta\mathcal{H}}(\mathcal{D}_1, \mathcal{D}_2) \leq \hat{d}_{\mathcal{H}\Delta\mathcal{H}}(S_1, S_2) + 4\sqrt{\frac{2d \log(2m) + \log \frac{2}{\delta}}{m}}. \tag{18}$$

We start with Theorem 3.4 of Kifer et al. (2004):

$$P_{m_1+m_2} \left[ |\phi_{\mathcal{A}}(S_1, S_2) - \phi_{\mathcal{A}}(P_1, P_2)| > \epsilon \right] \leq (2m)^d e^{-m_1 \epsilon^2 / 16} + (2m)^d e^{-m_2 \epsilon^2 / 16}. \tag{19}$$

In Eq. 19, '$d$' is the VC-dimension of a collection of subsets of some domain measure space $\mathcal{A}$, while in our case, $d$ is the VC-dimension of hypothesis space $\mathcal{H}$. Following (Ben-David et al., 2010), the $\mathcal{H}\Delta\mathcal{H}$ space is the set of disagreements between every two hypotheses in $\mathcal{H}$, which can be represented as a linear threshold network of depth 2 with 2 hidden units. Therefore, the VC-dimension of $\mathcal{H}\Delta\mathcal{H}$ is at most twice the VC-dimension of $\mathcal{H}$, and the VC-dimension of our domain measure space is $2d$ for Eq. 19 to hold.

Given $\delta \in (0,1)$, we set the upper bound of the inequality to $\delta$, and solve for $\epsilon$:

$$\delta = (2m)^{2d} e^{-m_1 \epsilon^2 / 16} + (2m)^{2d} e^{-m_2 \epsilon^2 / 16}.$$

We rewrite the inequality as

$$\frac{\delta}{(2m)^{2d}} = e^{-m_1 \epsilon^2 / 16} + e^{-m_2 \epsilon^2 / 16};$$

taking the logarithm of both sides, we get

$$\log \frac{\delta}{(2m)^{2d}} = -m_1 \frac{\epsilon^2}{16} + \log(1 + e^{-(m_1 - m_2)\frac{\epsilon^2}{16}}).$$

Assuming $m_1 = m_2 = m$ and defining $a = \frac{\epsilon^2}{16}$, we have

$$\log \frac{\delta}{(2m)^{2d}} = -ma + \log 2;$$

rearranging the equation, we then get

$$ma + \log(\delta/2) = 2d \log(2m).$$

Now, we can solve for $a$:

$$a = \frac{2d \log(2m) + \log \frac{2}{\delta}}{m}.$$

Recall that $a = \frac{\epsilon^2}{16}$, so we get:

$$\epsilon = 4\sqrt{a}$$

$$\epsilon = 4\sqrt{\frac{2d \log(2m) + \log \frac{2}{\delta}}{m}}.$$

With probability of at least $1 - \delta$, we have

$$|\phi_{\mathcal{A}}(S_1, S_2) - \phi_{\mathcal{A}}(P_1, P_2)| \leq 4\sqrt{\frac{2d \log(2m) + \log \frac{2}{\delta}}{m}};$$

therefore,

$$d_{\mathcal{H}\Delta\mathcal{H}}(\mathcal{D}_1, \mathcal{D}_2) \leq \hat{d}_{\mathcal{H}\Delta\mathcal{H}}(S_1, S_2) + 4\sqrt{\frac{2d \log(2m) + \log \frac{2}{\delta}}{m}}. \tag{20}$$

Now we prove Lemma 6. We use the triangle inequality for classification error in the derivation. For the domain error of hypothesis $h$ at time $t$ on the $j$th domain $\epsilon_j(h(t))$, given the definition of $\epsilon_{\boldsymbol{w}}(h(t))$,

$$|\epsilon_{\boldsymbol{w}}(h(t)) - \epsilon_j(h(t))| = |\sum_{i=0}^{t} w_i \epsilon_i(h(t)) - \epsilon_j(h(t))|$$

$$\leq \sum_{i=0}^{t} w_i |\epsilon_i(h(t)) - \epsilon_j(h(t))|$$

$$\leq \sum_{i=0}^{t} w_i (|\epsilon_i(h(t)) - \epsilon_i(h(t), h_i^*(t))| + |\epsilon_i(h(t), h_i^*(t)) - \epsilon_j(h(t), h_i^*(t))|$$

$$+ |\epsilon_j(h(t), h_i^*(t)) - \epsilon_j(h(t))|)$$

$$\leq \sum_{i=0}^{t} w_i (\epsilon_i(h_i^*(t)) + |\epsilon_i(h(t), h_i^*(t)) - \epsilon_j(h(t), h_i^*(t))| + \epsilon_j(h_i^*(t)))$$

$$\leq \sum_{i=0}^{t} w_i (\gamma_i + |\epsilon_i(h(t), h_i^*(t)) - \epsilon_j(h(t), h_i^*(t))|),$$

where $\gamma_i = \min_{h \in \mathcal{H}} \{\epsilon_i(h(t)) + \epsilon_j(h(t))\}$. By the definition of $\mathcal{H}\Delta\mathcal{H}$-distance and our proved Eq. 20,

$$|\epsilon_i(h(t), h_i^*(t)) - \epsilon_j(h(t), h_i^*(t))| \leq \sup_{h, h' \in \mathcal{H}} |\epsilon_i(h(t), h'(t)) - \epsilon_j(h(t), h'(t))|$$

$$= \sup_{h, h' \in \mathcal{H}} P_{x \sim \mathcal{D}_i}[h(x) \neq h'(x)] + P_{x \sim \mathcal{D}_j}[h(x) \neq h'(x)]$$

$$= \frac{1}{2} d_{\mathcal{H}\Delta\mathcal{H}}(\mathcal{D}_i, \mathcal{D}_j)$$

$$\leq \frac{1}{2} \hat{d}_{\mathcal{H}\Delta\mathcal{H}}(S_i, S_j) + 2\sqrt{\frac{2d \log(2m) + \log \frac{2}{\delta}}{m}},$$

where $\mathcal{D}_i, \mathcal{D}_j$ denote the $i$th and $j$th domain. Therefore,

$$
\begin{aligned}
|\epsilon_{\boldsymbol{w}}(h(t)) - \epsilon_j(h(t))| &\leq \sum_{i=0}^{t} w_i(\gamma_i + |\epsilon_i(h(t), h_i^*(t)) - \epsilon_j(h(t), h_i^*(t))|) \\
&\leq \sum_{i=0}^{t} w_i(\gamma_i + \frac{1}{2}d_{\mathcal{H}\Delta\mathcal{H}}(\mathcal{D}_i, \mathcal{D}_j)) \\
&\leq \sum_{i=0}^{t} w_i(\gamma_i + \frac{1}{2}\hat{d}_{\mathcal{H}\Delta\mathcal{H}}(S_i, S_j) + 2\sqrt{\frac{2d\log(2m) + \log\frac{2}{\delta}}{m}}).
\end{aligned}
$$

Since $\epsilon_i(h(t)) - \epsilon_j(h(t)) = 0$ when $i = j$, we derive

$$
|\epsilon_{\boldsymbol{w}}(h(t)) - \epsilon_j(h(t))| \leq \sum_{i=0, i\neq j}^{t} w_i \left( \frac{1}{2}\hat{d}_{\mathcal{H}\Delta\mathcal{H}}(S_i, S_j) + 2\sqrt{\frac{2d\log(2m) + \log\frac{2}{\delta}}{m}} + \gamma_i \right),
$$

with probability of at least $1 - \delta$, where $\gamma_i = \min_{h\in\mathcal{H}}\{\epsilon_i(h(t)) + \epsilon_j(h(t))\}$.

This completes the proof. $\square$

**Lemma 7.** *Let $H$ be a hypothesis class. For $D_{tr}(t) = D_S \cup D_{te}(1) \cup \cdots \cup D_{te}(t)$ at time $t$, if the total number of samples in $D_{tr}(t)$ is $N$, and the ratio of sample numbers in each component is $\lambda_j$, then for any $\delta \in (0, 1)$ and $h \in H$, with probability of at least $1 - \delta$, we have*

$$
P[|\epsilon_{\boldsymbol{w}}(h(t)) - \hat{\epsilon}_{\boldsymbol{w}}(h(t))| \geq \epsilon] \leq 2\exp\left(-2N\epsilon^2/(\sum_{j=0}^{t}\frac{w_j^2}{\lambda_j})\right).
$$

*Proof.* We apply Hoeffding's Inequality in our proof:

$$
\mathbb{P}\left(\left|\frac{1}{t}\sum_{i=1}^{t} f_i(x) - \frac{1}{t}\sum_{i=1}^{t}\mathbb{E}_{x\sim D_i}[f_i(x)]\right| \geq \epsilon\right) \leq 2\exp\left(-\frac{2t^2\epsilon^2}{\sum_{i=1}^{t}(b_i - a_i)^2}\right). \tag{21}
$$

In the $j$th domain, there are $\lambda_j N$ samples. With the true labeling function $g(x)$, for each of the $\lambda_j N$ samples $x$, let there be a real-valued function $f_i(x)$

$$
f_i(x) = \frac{w_j}{\lambda_j}|h(x, t) - g(x)|,
$$

where $f_i(x) \in [0, \frac{w_j}{\lambda_j}]$. Incorporating all the domains, we get

$$
\hat{\epsilon}_{\boldsymbol{w}}(h(t)) = \sum_{j=0}^{t} w_j\hat{\epsilon}_j(h(t)) = \sum_{j=0}^{t}\frac{w_j}{\lambda_j N}\sum_{\lambda_j N}|h(x, t) - g(x)| = \frac{1}{N}\sum_{j=0}^{t}\sum_{i=1}^{\lambda_j N} f_i(x),
$$

which corresponds to the $\frac{1}{t}\sum_{i=1}^{t} f_i(x)$ part in Hoeffding's Inequality.

Due to the linearity of expectations, we can calculate the sum of expectations as

$$
\frac{1}{N}\sum_{j=0}^{t}\sum_{i=1}^{\lambda_j N}\mathbb{E}[f_i(x)] = \frac{1}{N}(\sum_{j=0}^{t}\lambda_j N\frac{w_j}{\lambda_j}\epsilon_j(h(t))) = \sum_{j=0}^{t} w_j\epsilon_j(h(t)) = \epsilon_{\boldsymbol{w}}(h(t)),
$$

which corresponds to the $\frac{1}{t}\sum_{i=1}^{t}\mathbb{E}_{x\sim D_i}[f_i(x)]$ part in Hoeffding's Inequality. Therefore, we can apply Hoeffding's Inequality as

$$
\begin{aligned}
P[|\epsilon_{\boldsymbol{w}}(h(t)) - \hat{\epsilon}_{\boldsymbol{w}}(h(t))| \geq \epsilon] &\leq 2\exp\left(-2N^2\epsilon^2/(\sum_{i=0}^{N} range^2(f_i(x)))\right) \\
&= 2\exp\left(-2N^2\epsilon^2/(\sum_{j=0}^{t}\lambda_j N(\frac{w_j}{\lambda_j})^2)\right) \\
&= 2\exp\left(-2N\epsilon^2/(\sum_{j=0}^{t}\frac{w_j^2}{\lambda_j})\right).
\end{aligned}
$$

This completes the proof. $\square$

**Theorem 1.** *Let $H$ be a hypothesis class of VC-dimension $d$. At time step $t$, for ATTA data domains $D_S, U_{te}(1), U_{te}(2), \cdots, U_{te}(t)$, $S_i$ are unlabeled samples of size $m$ sampled from each of the $t + 1$ domains respectively. The total number of samples in $D_{tr}(t)$ is $N$ and the ratio of sample numbers in each component is $\lambda_i$. If $\hat{h}(t) \in \mathcal{H}$ minimizes the empirical weighted error $\hat{\epsilon}_{\boldsymbol{w}}(h(t))$ with the weight vector $\boldsymbol{w}$ on $D_{tr}(t)$, and $h_j^*(t) = \arg\min_{h \in \mathcal{H}} \epsilon_j(h(t))$ is the optimal hypothesis on the $j$th domain, then for any $\delta \in (0, 1)$, with probability of at least $1 - \delta$, we have*

$$\epsilon_j(\hat{h}(t)) \leq \epsilon_j(h_j^*(t)) + 2 \sum_{i=0,i\neq j}^{t} w_i \left( \frac{1}{2} \hat{d}_{\mathcal{H}\Delta\mathcal{H}}(S_i, S_j) + 2\sqrt{\frac{2d\log(2m) + \log\frac{2}{\delta}}{m}} + \gamma_i \right) + 2C,$$

*where $C = \sqrt{\left(\sum_{i=0}^{t} \frac{w_i^2}{\lambda_i}\right)\left(\frac{d\log(2N) - \log(\delta)}{2N}\right)}$ and $\gamma_i = \min_{h \in \mathcal{H}}\{\epsilon_i(h(t)) + \epsilon_j(h(t))\}$. For future test domains $j = t + k$ ($k > 0$), assuming $k' = \arg\min_{k' \in \{0,1,\dots t\}} d_{\mathcal{H}\Delta\mathcal{H}}(D(k'), U_{te}(t + k))$ and $\min d_{\mathcal{H}\Delta\mathcal{H}}(D(k'), U_{te}(t + k)) \leq \delta_D$, where $0 \leq \delta_D \ll +\infty$, then $\forall \delta$, with probability of at least $1 - \delta$, we have*

$$\epsilon_{t+k}(\hat{h}(t)) \leq \epsilon_{t+k}(h_{t+k}^*(t)) + \sum_{i=0}^{t} w_i \left( \hat{d}_{\mathcal{H}\Delta\mathcal{H}}(S_i, S_{k'}) + 4\sqrt{\frac{2d\log(2m) + \log\frac{2}{\delta}}{m}} + \delta_D + 2\gamma_i \right) + 2C.$$

*Proof.* First we prove that for any $\delta \in (0, 1)$ and $h \in H$, with probability of at least $1 - \delta$, we have

$$|\epsilon_{\boldsymbol{w}}(h(t)) - \hat{\epsilon}_{\boldsymbol{w}}(h(t))| \leq \sqrt{\left(\sum_{i=0}^{t} \frac{w_i^2}{\lambda_i}\right)\left(\frac{d\log(2N) - \log(\delta)}{2N}\right)}. \tag{22}$$

We apply Theorem 3.2 of Kifer et al. (2004) and Lemma 7,

$$P[|\epsilon_{\boldsymbol{w}}(h(t)) - \hat{\epsilon}_{\boldsymbol{w}}(h(t))| \geq \epsilon] \leq (2N)^d \exp\left(-2N\epsilon^2 / (\sum_{j=0}^{t} \frac{w_j^2}{\lambda_j})\right).$$

Given $\delta \in (0, 1)$, we set the upper bound of the inequality to $\delta$, and solve for $\epsilon$:

$$\delta = (2N)^d \exp\left(-2N\epsilon^2 / (\sum_{j=0}^{t} \frac{w_j^2}{\lambda_j})\right).$$

We rewrite the inequality as

$$\frac{\delta}{(2N)^d} = e^{-2N\epsilon^2 / (\sum_{j=0}^{t} \frac{w_j^2}{\lambda_j})},$$

taking the logarithm of both sides, we get

$$\log\frac{\delta}{(2N)^d} = -2N\epsilon^2 / (\sum_{j=0}^{t} \frac{w_j^2}{\lambda_j}).$$

Rearranging the equation, we then get

$$\epsilon^2 = (\sum_{j=0}^{t} \frac{w_j^2}{\lambda_j})\frac{d\log(2N) - \log(\delta)}{2N}.$$

Therefore, with probability of at least $1 - \delta$, we have

$$|\epsilon_{\boldsymbol{w}}(h(t)) - \hat{\epsilon}_{\boldsymbol{w}}(h(t))| \leq \sqrt{\left(\sum_{i=0}^{t} \frac{w_i^2}{\lambda_i}\right)\left(\frac{d\log(2N) - \log(\delta)}{2N}\right)}. \tag{23}$$

Based on Eq. 23, we now prove Theorem 1. For the empirical domain error of hypothesis $h$ at time $t$ on the $j$th domain $\epsilon_j(\hat{h}(t))$, applying Lemma 6, Eq. 23, and the definition of $h_j^*(t)$, we get

$$\epsilon_j(\hat{h}(t)) \leq \epsilon_{\boldsymbol{w}}(\hat{h}(t)) + \sum_{i=0,i\neq j}^{t} w_i \left( \frac{1}{2}\hat{d}_{\mathcal{H}\Delta\mathcal{H}}(S_i,S_j) + 2\sqrt{\frac{2d\log(2m)+\log\frac{2}{\delta}}{m}} + \gamma_i \right)$$

$$\leq \hat{\epsilon}_{\boldsymbol{w}}(\hat{h}(t)) + \sqrt{\left(\sum_{i=0}^{t}\frac{w_i^2}{\lambda_i}\right)\left(\frac{d\log(2N)-\log(\delta)}{2N}\right)}$$

$$+ \sum_{i=0,i\neq j}^{t} w_i \left( \frac{1}{2}\hat{d}_{\mathcal{H}\Delta\mathcal{H}}(S_i,S_j) + 2\sqrt{\frac{2d\log(2m)+\log\frac{2}{\delta}}{m}} + \gamma_i \right)$$

$$\leq \hat{\epsilon}_{\boldsymbol{w}}(h_j^*(t)) + \sqrt{\left(\sum_{i=0}^{t}\frac{w_i^2}{\lambda_i}\right)\left(\frac{d\log(2N)-\log(\delta)}{2N}\right)}$$

$$+ \sum_{i=0,i\neq j}^{t} w_i \left( \frac{1}{2}\hat{d}_{\mathcal{H}\Delta\mathcal{H}}(S_i,S_j) + 2\sqrt{\frac{2d\log(2m)+\log\frac{2}{\delta}}{m}} + \gamma_i \right)$$

$$\leq \epsilon_{\boldsymbol{w}}(h_j^*(t)) + 2\sqrt{\left(\sum_{i=0}^{t}\frac{w_i^2}{\lambda_i}\right)\left(\frac{d\log(2N)-\log(\delta)}{2N}\right)}$$

$$+ \sum_{i=0,i\neq j}^{t} w_i \left( \frac{1}{2}\hat{d}_{\mathcal{H}\Delta\mathcal{H}}(S_i,S_j) + 2\sqrt{\frac{2d\log(2m)+\log\frac{2}{\delta}}{m}} + \gamma_i \right)$$

$$\leq \epsilon_j(h_j^*(t)) + 2\sqrt{\left(\sum_{i=0}^{t}\frac{w_i^2}{\lambda_i}\right)\left(\frac{d\log(2N)-\log(\delta)}{2N}\right)}$$

$$+ 2\sum_{i=0,i\neq j}^{t} w_i \left( \frac{1}{2}\hat{d}_{\mathcal{H}\Delta\mathcal{H}}(S_i,S_j) + 2\sqrt{\frac{2d\log(2m)+\log\frac{2}{\delta}}{m}} + \gamma_i \right)$$

$$= \epsilon_j(h_j^*(t)) + 2\sum_{i=0,i\neq j}^{t} w_i \left( \frac{1}{2}\hat{d}_{\mathcal{H}\Delta\mathcal{H}}(S_i,S_j) + 2\sqrt{\frac{2d\log(2m)+\log\frac{2}{\delta}}{m}} + \gamma_i \right) + 2C$$

with probability of at least $1-\delta$, where $C = \sqrt{\left(\sum_{i=0}^{t}\frac{w_i^2}{\lambda_i}\right)\left(\frac{d\log(2N)-\log(\delta)}{2N}\right)}$ and $\gamma_i = \min_{h\in\mathcal{H}}\{\epsilon_i(h(t))+\epsilon_j(h(t))\}$.

For future test domains $j = t + k$ where $k > 0$, we have the assumption that $k' = \operatorname{argmin}_{k'\in\{0,1,\dots t\}} d_{\mathcal{H}\Delta\mathcal{H}}(D(k'), U_{te}(t+k))$ and $\min d_{\mathcal{H}\Delta\mathcal{H}}(D(k'), U_{te}(t+k)) \leq \delta_D$. Here, we slightly abuse the notation $D(k')$ to represent $D_s$ if $k' = 0$ and $U_{te}(k')$ if $k' > 0$. Then we get

$$\epsilon_{t+k}(\hat{h}(t)) \leq \epsilon_{\boldsymbol{w}}(\hat{h}(t)) + \sum_{i=0}^{t} w_i \left( \frac{1}{2}\hat{d}_{\mathcal{H}\Delta\mathcal{H}}(S_i,S_{t+k}) + 2\sqrt{\frac{2d\log(2m)+\log\frac{2}{\delta}}{m}} + \gamma_i \right)$$

$$\leq \epsilon_{\boldsymbol{w}}(\hat{h}(t)) + \sum_{i=0}^{t} w_i \left( \frac{1}{2}(\hat{d}_{\mathcal{H}\Delta\mathcal{H}}(S_i,S_{k'}) + \hat{d}_{\mathcal{H}\Delta\mathcal{H}}(S_{k'},S_{t+k})) + 2\sqrt{\frac{2d\log(2m)+\log\frac{2}{\delta}}{m}} + \gamma_i \right)$$

$$\leq \epsilon_{\boldsymbol{w}}(\hat{h}(t)) + \sum_{i=0}^{t} w_i \left( \frac{1}{2}\hat{d}_{\mathcal{H}\Delta\mathcal{H}}(S_i,S_{k'}) + \frac{1}{2}\delta_D + 2\sqrt{\frac{2d\log(2m)+\log\frac{2}{\delta}}{m}} + \gamma_i \right)$$

$$\leq \hat{\epsilon}_{\boldsymbol{w}}(\hat{h}(t)) + \sqrt{\left(\sum_{i=0}^{t}\frac{w_i^2}{\lambda_i}\right)\left(\frac{d\log(2N)-\log(\delta)}{2N}\right)}$$

$$+ \sum_{i=0}^{t} w_i \left( \frac{1}{2}\hat{d}_{\mathcal{H}\Delta\mathcal{H}}(S_i,S_{k'}) + \frac{1}{2}\delta_D + 2\sqrt{\frac{2d\log(2m)+\log\frac{2}{\delta}}{m}} + \gamma_i \right)$$

$$\leq \hat{\epsilon}_{\boldsymbol{w}}(h_{t+k}^*(t)) + \sqrt{\left(\sum_{i=0}^{t} \frac{w_i^2}{\lambda_i}\right)\left(\frac{d\log(2N)-\log(\delta)}{2N}\right)}$$

$$+ \sum_{i=0}^{t} w_i \left(\frac{1}{2}\hat{d}_{\mathcal{H}\Delta\mathcal{H}}(S_i, S_{k'}) + \frac{1}{2}\delta_D + 2\sqrt{\frac{2d\log(2m)+\log\frac{2}{\delta}}{m}} + \gamma_i\right)$$

$$\leq \epsilon_{\boldsymbol{w}}(h_{t+k}^*(t)) + 2\sqrt{\left(\sum_{i=0}^{t} \frac{w_i^2}{\lambda_i}\right)\left(\frac{d\log(2N)-\log(\delta)}{2N}\right)}$$

$$+ \sum_{i=0}^{t} w_i \left(\frac{1}{2}\hat{d}_{\mathcal{H}\Delta\mathcal{H}}(S_i, S_{k'}) + \frac{1}{2}\delta_D + 2\sqrt{\frac{2d\log(2m)+\log\frac{2}{\delta}}{m}} + \gamma_i\right)$$

$$\leq \epsilon_{t+k}(h_{t+k}^*(t)) + 2\sqrt{\left(\sum_{i=0}^{t} \frac{w_i^2}{\lambda_i}\right)\left(\frac{d\log(2N)-\log(\delta)}{2N}\right)}$$

$$+ 2\sum_{i=0}^{t} w_i \left(\frac{1}{2}\hat{d}_{\mathcal{H}\Delta\mathcal{H}}(S_i, S_{k'}) + \frac{1}{2}\delta_D + 2\sqrt{\frac{2d\log(2m)+\log\frac{2}{\delta}}{m}} + \gamma_i\right)$$

$$= \epsilon_{t+k}(h_{t+k}^*(t)) + \sum_{i=0}^{t} w_i \left(\hat{d}_{\mathcal{H}\Delta\mathcal{H}}(S_i, S_{k'}) + 4\sqrt{\frac{2d\log(2m)+\log\frac{2}{\delta}}{m}} + \delta_D + 2\gamma_i\right) + 2C.$$

with probability of at least $1-\delta$, where $C = \sqrt{\left(\sum_{i=0}^{t} \frac{w_i^2}{\lambda_i}\right)\left(\frac{d\log(2N)-\log(\delta)}{2N}\right)}$, $\gamma_i = \min_{h\in\mathcal{H}}\{\epsilon_i(h(t)) + \epsilon_{t+k}(h(t))\}$, and $0 \leq \delta_D \ll +\infty$.

This completes the proof. $\qquad\square$

**Theorem 2.** *Let $H$ be a hypothesis class of VC-dimension $d$. For ATTA data domains $D_S, U_{te}(1)$, $U_{te}(2), \cdots, U_{te}(t)$, considering the test-time data as a single test domain $\bigcup_{i=1}^{t} U_{te}(i)$, if $\hat{h}(t) \in \mathcal{H}$ minimizes the empirical weighted error $\hat{\epsilon}_{\boldsymbol{w}}(h(t))$ with the weight vector $\boldsymbol{w}$ on $D_{tr}(t)$, let the test error be upper-bounded with $|\epsilon_T(\hat{h}(t)) - \epsilon_T(h_T^*(t))| \leq EB_T(\boldsymbol{w}, \boldsymbol{\lambda}, N, t)$. Let $\boldsymbol{w'}$ and $\boldsymbol{\lambda'}$ be the weight and sample ratio vectors when no active learning is included, i.e., $\boldsymbol{w'}$ and $\boldsymbol{\lambda'}$ s.t. $w_0' = \lambda_0' = 1$ and $w_i' = \lambda_i' = 0$ for $i \geq 1$, then for any $\boldsymbol{\lambda} \neq \boldsymbol{\lambda'}$, there exists $\boldsymbol{w}$ s.t.*

$$EB_T(\boldsymbol{w}, \boldsymbol{\lambda}, N, t) < EB_T(\boldsymbol{w'}, \boldsymbol{\lambda'}, N, t). \tag{24}$$

*Proof.* From Theorem 1, we can derive the bound for the test error where the test-time data are considered as a single test domain:

$$|\epsilon_T(\hat{h}(t)) - \epsilon_T(h_T^*(t))| \leq EB_T(\boldsymbol{w}, \boldsymbol{\lambda}, N, t)$$

$$= w_0(\hat{d}_{\mathcal{H}\Delta\mathcal{H}}(S_0, S_T) + 4\sqrt{\frac{2d\log(2m)+\log\frac{2}{\delta}}{m}} + 2\gamma)$$

$$+ 2\sqrt{\frac{w_0^2}{\lambda_0} + \frac{(1-w_0)^2}{1-\lambda_0}}\sqrt{\frac{d\log(2N)-\log(\delta)}{2N}};$$

and we simplify the above equation as

$$|\epsilon_T(\hat{h}(t)) - \epsilon_T(h_T^*(t))| \leq w_0 A + \sqrt{\frac{w_0^2}{\lambda_0} + \frac{(1-w_0)^2}{1-\lambda_0}}B, \tag{25}$$

where the distribution divergence term $A = \hat{d}_{\mathcal{H}\Delta\mathcal{H}}(S_0, S_T) + 4\sqrt{\frac{2d\log(2m)+\log\frac{2}{\delta}}{m}} + 2\gamma$, the empirical gap term $B = 2\sqrt{\frac{d\log(2N)-\log(\delta)}{2N}}$, $S_T$ is sampled from $\bigcup_{i=1}^{t} U_{te}(i)$, and $\gamma = \min_{h\in\mathcal{H}}\{\epsilon_0(h(t)) + \epsilon_T(h(t))\}$.

Since we have

$$\sqrt{\frac{w_0^2}{\lambda_0} + \frac{(1-w_0)^2}{1-\lambda_0}} = \sqrt{\frac{(w_0-\lambda_0)^2}{\lambda_0(1-\lambda_0)} + 1} \geq 1, \tag{26}$$

where Formula 26 obtains the minimum value if and only if $w_0 = \lambda_0$; when enabling ATTA with any $\lambda_0 \neq 1$, we can get

$$EB_T(\boldsymbol{w}, \boldsymbol{\lambda}, N, t) = w_0 A + \sqrt{\frac{w_0^2}{\lambda_0} + \frac{(1-w_0)^2}{1-\lambda_0}} B \geq w_0 A + B, \tag{27}$$

where the minimum value $EB_T(\boldsymbol{w}, \boldsymbol{\lambda}, N, t)_{min} = w_0 A + B$ can be obtained with condition $w_0 = \lambda_0 \neq 1$. When no active learning is included, *i.e.*, for weight and sample ratio vectors $\boldsymbol{w}'$ and $\boldsymbol{\lambda}'$, $w_0' = \lambda_0' = 1$ and $w_i' = \lambda_i' = 0$ for $i \geq 1$, we have

$$EB_T(\boldsymbol{w}', \boldsymbol{\lambda}', N, t) = w_0' A + \sqrt{\frac{w_0'^2}{\lambda_0'} + \frac{(1-w_0')^2}{1-\lambda_0'}} B = A + B. \tag{28}$$

Since for $EB_T(\boldsymbol{w}, \boldsymbol{\lambda}, N, t)_{min} = w_0 A + B$, $w_0 < 1$ and $A, B > 0$ hold, we derive

$$EB_T(\boldsymbol{w}, \boldsymbol{\lambda}, N, t)_{min} = w_0 A + B < A + B = EB_T(\boldsymbol{w}', \boldsymbol{\lambda}', N, t). \tag{29}$$

This completes the proof. $\qquad\square$

**Corollary 3.** *At time step $t$, for ATTA data domains $D_{\phi,S}(t), U_{te}(1), U_{te}(2), \cdots, U_{te}(t)$, $S_i$ are unlabeled samples of size $m$ sampled from each of the $t+1$ domains respectively, and $S_S$ is unlabeled samples of size $m$ sampled from $D_S$. If $\hat{h}(t) \in \mathcal{H}$ minimizes $\hat{\epsilon}'_{\boldsymbol{w}}(h(t))$ while other conditions remain identical to Theorem 1, then*

$$\epsilon_S(\hat{h}(t)) \leq \epsilon_S(h_S^*(t)) + \sum_{i=0}^{t} w_i \left( \hat{d}_{\mathcal{H}\Delta\mathcal{H}}(S_i, S_S) + 4\sqrt{\frac{2d\log(2m) + \log\frac{2}{\delta}}{m}} + 2\gamma_i \right) + 2C,$$

*with probability at least $1 - \delta$, where $C$ follows Theorem 1 and $\gamma_i = \min_{h \in \mathcal{H}}\{\epsilon_i(h(t)) + \epsilon_S(h(t))\}$.*

*Proof.* For the empirical source error on $D_S$ of hypothesis $h$ at time $t$, similar to Theorem 1, we apply Lemma 6, Eq. 23 to get

$$\epsilon_S(\hat{h}(t)) \leq \epsilon_{\boldsymbol{w}}(\hat{h}(t)) + \sum_{i=0}^{t} w_i \left( \frac{1}{2}\hat{d}_{\mathcal{H}\Delta\mathcal{H}}(S_i, S_S) + 2\sqrt{\frac{2d\log(2m) + \log\frac{2}{\delta}}{m}} + \gamma_i \right)$$

$$\leq \hat{\epsilon}_{\boldsymbol{w}}(\hat{h}(t)) + \sqrt{\left(\sum_{i=0}^{t}\frac{w_i^2}{\lambda_i}\right)\left(\frac{d\log(2N) - \log(\delta)}{2N}\right)}$$

$$+ \sum_{i=0}^{t} w_i \left( \frac{1}{2}\hat{d}_{\mathcal{H}\Delta\mathcal{H}}(S_i, S_S) + 2\sqrt{\frac{2d\log(2m) + \log\frac{2}{\delta}}{m}} + \gamma_i \right)$$

$$\leq \hat{\epsilon}_{\boldsymbol{w}}(h_S^*(t)) + \sqrt{\left(\sum_{i=0}^{t}\frac{w_i^2}{\lambda_i}\right)\left(\frac{d\log(2N) - \log(\delta)}{2N}\right)}$$

$$+ \sum_{i=0}^{t} w_i \left( \frac{1}{2}\hat{d}_{\mathcal{H}\Delta\mathcal{H}}(S_i, S_S) + 2\sqrt{\frac{2d\log(2m) + \log\frac{2}{\delta}}{m}} + \gamma_i \right)$$

$$\leq \epsilon_{\boldsymbol{w}}(h_S^*(t)) + 2\sqrt{\left(\sum_{i=0}^{t}\frac{w_i^2}{\lambda_i}\right)\left(\frac{d\log(2N) - \log(\delta)}{2N}\right)}$$

$$+ \sum_{i=0}^{t} w_i \left( \frac{1}{2}\hat{d}_{\mathcal{H}\Delta\mathcal{H}}(S_i, S_S) + 2\sqrt{\frac{2d\log(2m) + \log\frac{2}{\delta}}{m}} + \gamma_i \right)$$

$$\leq \epsilon_S(h_S^*(t)) + 2\sqrt{\left(\sum_{i=0}^{t}\frac{w_i^2}{\lambda_i}\right)\left(\frac{d\log(2N) - \log(\delta)}{2N}\right)}$$

$$+ 2\sum_{i=0}^{t} w_i \left( \frac{1}{2}\hat{d}_{\mathcal{H}\Delta\mathcal{H}}(S_i, S_S) + 2\sqrt{\frac{2d\log(2m) + \log\frac{2}{\delta}}{m}} + \gamma_i \right)$$

$$= \epsilon_S(h_S^*(t)) + \sum_{i=0}^{t} w_i \left( \hat{d}_{\mathcal{H}\Delta\mathcal{H}}(S_i, S_S) + 4\sqrt{\frac{2d\log(2m) + \log\frac{2}{\delta}}{m}} + 2\gamma_i \right) + 2C$$

with probability of at least $1 - \delta$, where $C = \sqrt{\left(\sum_{i=0}^{t} \frac{w_i^2}{\lambda_i}\right)\left(\frac{d\log(2N) - \log(\delta)}{2N}\right)}$ and $\gamma_i = \min_{h \in \mathcal{H}}\{\epsilon_i(h(t)) + \epsilon_S(h(t))\}$.

This completes the proof. $\qquad\square$

**Corollary 4.** *At time step $t$, for ATTA data domains $D_{\phi,S}(t), U_{te}(1), U_{te}(2), \cdots, U_{te}(t)$, suppose that $\hat{h}(t) \in \mathcal{H}$ minimizes $\hat{\epsilon}\boldsymbol{w}'(h(t))$ under identical conditions to Theorem 2. Let's denote the source error upper bound with $|\epsilon_S(\hat{h}(t)) - \epsilon_S(h_S^*(t))| \leq EB_S(\boldsymbol{w}, \boldsymbol{\lambda}, N, t)$. Let $\boldsymbol{w}'$ and $\boldsymbol{\lambda}'$ be the weight and sample ratio vectors when $D_{\phi,S}(t)$ is not included, i.e., $\boldsymbol{w}'$ and $\boldsymbol{\lambda}'$ s.t. $w_0' = \lambda_0' = 0$. If $\hat{d}_{\mathcal{H}\Delta\mathcal{H}}(D_S, D_{\phi,S}(t)) < \hat{d}_{\mathcal{H}\Delta\mathcal{H}}(D_S, \bigcup_{i=1}^{t} U_{te}(i))$, then for any $\boldsymbol{\lambda} \neq \boldsymbol{\lambda}'$, there exists $\boldsymbol{w}$ s.t.*

$$EB_S(\boldsymbol{w}, \boldsymbol{\lambda}, N, t) < EB_S(\boldsymbol{w}', \boldsymbol{\lambda}', N, t). \tag{30}$$

*Proof.* From Theorem 1, considering the test-time data as a single test domain, we can derive the bound for the source error on $D_S$:

$$|\epsilon_S(\hat{h}(t)) - \epsilon_S(h_S^*(t))| \leq EB_S(\boldsymbol{w}, \boldsymbol{\lambda}, N, t)$$

$$= w_0(\hat{d}_{\mathcal{H}\Delta\mathcal{H}}(S_0, S_S) + 4\sqrt{\frac{2d\log(2m) + \log\frac{2}{\delta}}{m}} + 2\gamma)$$

$$+ (1 - w_0)(\hat{d}_{\mathcal{H}\Delta\mathcal{H}}(S_S, S_T) + 4\sqrt{\frac{2d\log(2m) + \log\frac{2}{\delta}}{m}} + 2\gamma')$$

$$+ 2\sqrt{\frac{w_0^2}{\lambda_0} + \frac{(1 - w_0)^2}{1 - \lambda_0}}\sqrt{\frac{d\log(2N) - \log(\delta)}{2N}},$$

where $S_T$ is sampled from $\bigcup_{i=1}^{t} U_{te}(i)$, $\gamma = \min_{h \in \mathcal{H}}\{\epsilon_0(h(t)) + \epsilon_S(h(t))\}$, and $\gamma' = \min_{h \in \mathcal{H}}\{\epsilon_T(h(t)) + \epsilon_S(h(t))\}$. We have

$$\sqrt{\frac{w_0^2}{\lambda_0} + \frac{(1 - w_0)^2}{1 - \lambda_0}} = \sqrt{\frac{(w_0 - \lambda_0)^2}{\lambda_0(1 - \lambda_0)} + 1} \geq 1, \tag{31}$$

where the equality and the minimum value are obtained if and only if $w_0 = \lambda_0$.

When $D_{\phi,S}(t)$ is not included, *i.e.*, with the weight and sample ratio vectors $\boldsymbol{w}'$ and $\boldsymbol{\lambda}'$ s.t. $w_0' = \lambda_0' = 0$, using the empirical gap term $B = 2\sqrt{\frac{d\log(2N) - \log(\delta)}{2N}}$, we have

$$EB_S(\boldsymbol{w}', \boldsymbol{\lambda}', N, t) = \hat{d}_{\mathcal{H}\Delta\mathcal{H}}(S_S, S_T) + 4\sqrt{\frac{2d\log(2m) + \log\frac{2}{\delta}}{m}} + 2\gamma' + \sqrt{\frac{w_0^2}{\lambda_0} + \frac{(1 - w_0)^2}{1 - \lambda_0}}B$$

$$= \hat{d}_{\mathcal{H}\Delta\mathcal{H}}(S_S, S_T) + 4\sqrt{\frac{2d\log(2m) + \log\frac{2}{\delta}}{m}} + 2\gamma' + B.$$

When $D_{\phi,S}(t)$ is included with $\lambda_0 \neq 0$,

$$EB_S(\boldsymbol{w}, \boldsymbol{\lambda}, N, t) = w_0(\hat{d}_{\mathcal{H}\Delta\mathcal{H}}(S_0, S_S) + 4\sqrt{\frac{2d\log(2m) + \log\frac{2}{\delta}}{m}} + 2\gamma)$$

$$+ (1 - w_0)(\hat{d}_{\mathcal{H}\Delta\mathcal{H}}(S_S, S_T) + 4\sqrt{\frac{2d\log(2m) + \log\frac{2}{\delta}}{m}} + 2\gamma')$$

$$+ \sqrt{\frac{w_0^2}{\lambda_0} + \frac{(1 - w_0)^2}{1 - \lambda_0}}B$$

$$\leq w_0(\hat{d}_{\mathcal{H}\Delta\mathcal{H}}(S_0, S_S) + 4\sqrt{\frac{2d\log(2m) + \log\frac{2}{\delta}}{m}} + 2\gamma)$$

$$+ (1 - w_0)(\hat{d}_{\mathcal{H}\Delta\mathcal{H}}(S_S, S_T) + 4\sqrt{\frac{2d\log(2m) + \log\frac{2}{\delta}}{m}} + 2\gamma') + B,$$

---

**Algorithm 2** INCREMENTAL CLUSTERING (IC)

---

**Require:** Given previously selected anchors, new unlabeled samples, and the cluster budget as $D_{\text{anc}}$, $U_{\text{new}}$, and $NC$. Global anchor weights $\mathbf{w}^{\text{anc}} = (w_1^{\text{anc}}, \ldots, w_{|D_{\text{anc}}|}^{\text{anc}})^\top$.

1: For simplicity, we consider anchor weights $\mathbf{w}^{\text{anc}}$ as a global vector.
2: **function** IC($D_{\text{anc}}$, $U_{\text{new}}$, $NC$)
3:     $\mathbf{w}^{\text{sp}} \leftarrow \text{Concat}(\mathbf{w}^{\text{anc}}, \mathbf{1}_{|U_{\text{new}}|}^\top)$                   ▷ Assign all new samples with weight 1.
4:     $\Phi \leftarrow$ Extract the features from the penultimate layer of model $f$ on $x \in D_{\text{anc}} \cup U_{\text{new}}$ in order.
5:     clusters $\leftarrow$ Weighted-K-Means($\Phi$, $\mathbf{w}^{\text{sp}}$, $NC$)
6:     new_clusters $\leftarrow \{\text{cluster}_i \mid \forall \text{cluster}_i \in \text{clusters}, \forall x \in D_{\text{anc}}, x \notin \text{clusters}_i\}$
7:     $X_{\text{new\_anchors}} \leftarrow \{\text{the closest sample } x \text{ to the centroid of cluster}_i \mid \forall \text{cluster}_i \in \text{new\_clusters}\}$
8:     $X_{\text{anchors}} \leftarrow \{x \in D_{\text{anc}}\} \cup X_{\text{new\_anchors}}$
9:     $\mathbf{w}^{\text{anc}} \leftarrow \text{Concat}(\mathbf{w}^{\text{anc}}, \mathbf{0}_{|X_{\text{new\_anchors}}|}^\top)$              ▷ Initialize new anchor weights.
10:     **for** $w_i^{\text{anc}} \in \mathbf{w}^{\text{anc}}$, $w_i^{\text{anc}} \leftarrow w_i^{\text{anc}} + \dfrac{\text{\# sample of cluster}_j}{\text{\# anchor in cluster}_j}$, $w_i^{\text{anc}} \in \text{cluster}_j$    ▷ Weight accumulation.
11:     **Return** $X_{\text{anchors}}$
12: **end function**

---

where the minimum value can be obtained with condition $w_0 = \lambda_0 \neq 0$.

In practical learning scenarios, we generally assume adaptation tasks are solvable; therefore, there should be a prediction function that performs well on two distinct domains. In this case, $\gamma$ and $\gamma'$ should be relatively small, so we can assume $\gamma \approx \gamma'$. If $\hat{d}_{\mathcal{H}\Delta\mathcal{H}}(S_0, S_S) < \hat{d}_{\mathcal{H}\Delta\mathcal{H}}(S_S, S_T)$, then we have

$$
\begin{aligned}
EB_S(\boldsymbol{w}, \boldsymbol{\lambda}, N, t)_{min} &= w_0(\hat{d}_{\mathcal{H}\Delta\mathcal{H}}(S_0, S_S) + 4\sqrt{\frac{2d\log(2m) + \log\frac{2}{\delta}}{m}} + 2\gamma) \\
&\quad + (1 - w_0)(\hat{d}_{\mathcal{H}\Delta\mathcal{H}}(S_S, S_T) + 4\sqrt{\frac{2d\log(2m) + \log\frac{2}{\delta}}{m}} + 2\gamma') + B \\
&< \hat{d}_{\mathcal{H}\Delta\mathcal{H}}(S_S, S_T) + 4\sqrt{\frac{2d\log(2m) + \log\frac{2}{\delta}}{m}} + 2\gamma' + B \\
&= EB_S(\boldsymbol{w'}, \boldsymbol{\lambda'}, N, t).
\end{aligned}
$$

Therefore, we derive

$$
EB_S(\boldsymbol{w}, \boldsymbol{\lambda}, N, t)_{min} < EB_S(\boldsymbol{w'}, \boldsymbol{\lambda'}, N, t). \tag{32}
$$

This completes the proof.          $\square$

## F    INCREMENTAL CLUSTERING

### F.1    ALGORITHM DETAILS

We provide the detailed algorithm for incremental clustering as Alg. 2.

### F.2    VISUALIZATION

To better illustrate the incremental clustering algorithm, we provide visualization results on PACS to demonstrate the process. As shown in Fig. 3, the initial step of IC is a normal K-Means clustering step, and ten anchors denoted as "X" are selected. The weights of all samples in a clusters is aggregated into the corresponding anchor's weight. Therefore, these ten samples (anchors) are given larger sizes visually (*i.e.*, larger weights) than that of other new test samples in the first IC step (Fig. 4). During the first IC step, several distributions are far away from the existed anchors and form clusters 1,7,9 and 10, which leads to 4 new selected anchors. While the number of cluster centroid is only increased by 1, 4 of the existing anchors are clustered into the same cluster 8 (purple). Thus IC produces 4 new anchors instead of 1. Similarly, in the second IC step (Fig. 5), the new streaming-in test samples introduce a new distribution; IC produces 3 new clusters (4, 8, and 11) and the corresponding number of anchors to cover them. The number of centroid is only increased by 1, which implies that there are two original-cluster-merging events. More IC step visualization results are provided in Fig. 6 and 7.

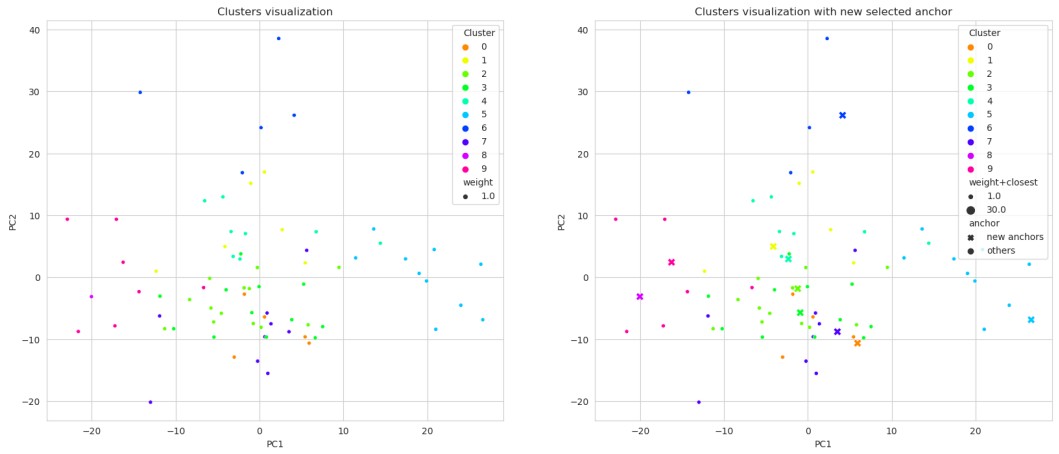

Figure 3: **Initial IC step: normal clustering.** Left: Clustering results. Right: Selecting new anchors.

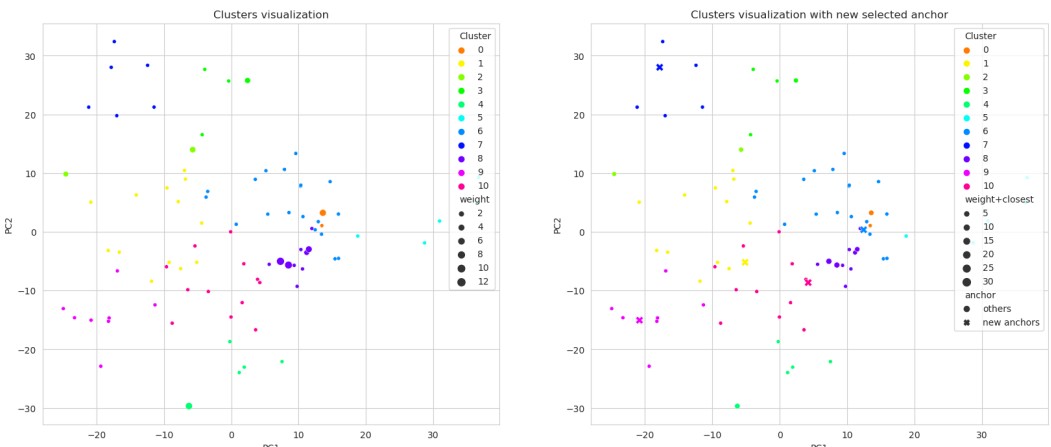

Figure 4: **The first IC step.** Left: Weighted clustering results. Right: Selecting new anchors.

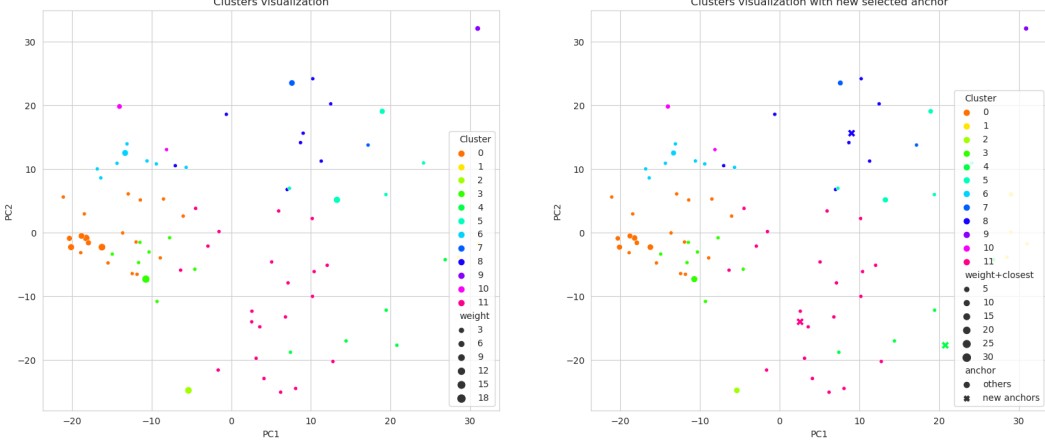

Figure 5: **The second IC step.** Left: Weighted clustering results. Right: Selecting new anchors.

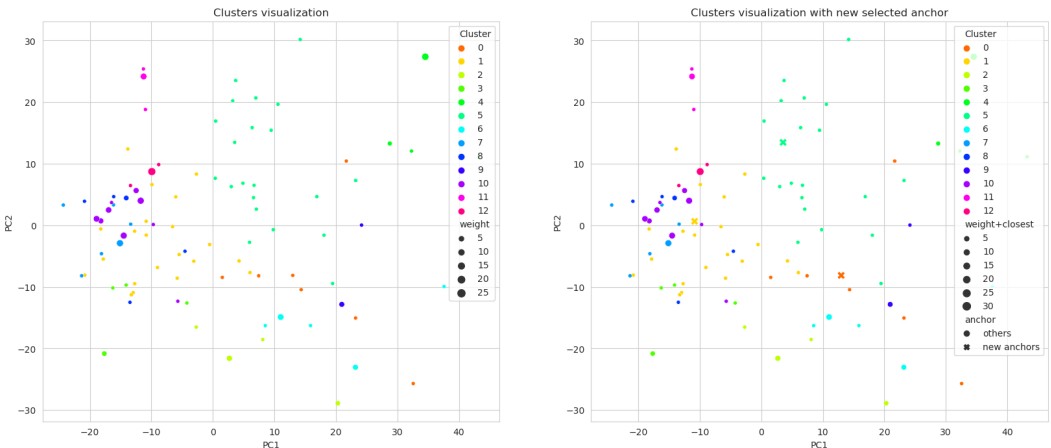

Figure 6: **The third IC step.** Left: Weighted clustering results. Right: Selecting new anchors.

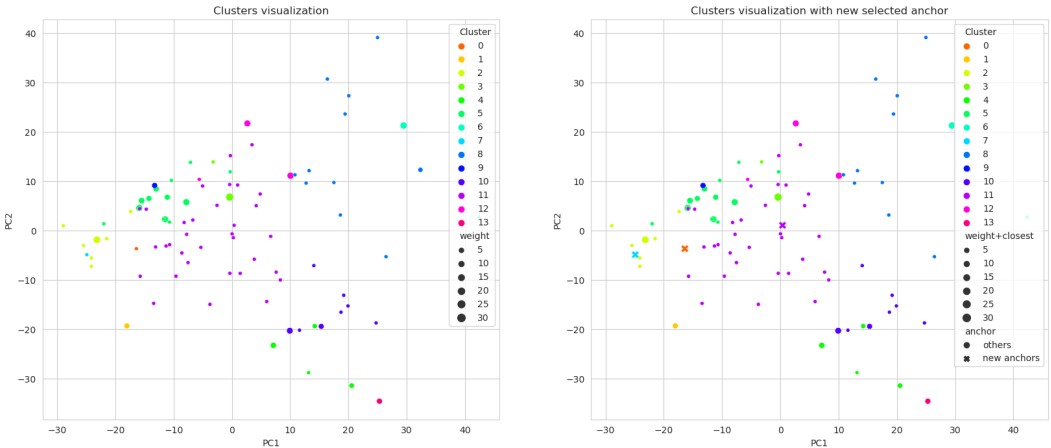

Figure 7: **The fourth IC step.** Left: Weighted clustering results. Right: Selecting new anchors.

## G    EXPERIMENT DETAILS

In this section, we provide more experimental details including the details of the datasets and training settings.

### G.1    DETAILS ABOUT THE DATASETS

We adopt datasets PACS, VLCS, and Office-Home from DomainBed (Gulrajani and Lopez-Paz, 2020) with the same domain splits. All available licenses are mentioned below.

- **PACS (Li et al., 2017)** includes four domains: art, cartoons, photos, and sketches. PACS is a 7-class classification dataset with 9,991 images of dimension (3, 224, 224).

- **VLCS (Fang et al., 2013)** contains photographic domains: Caltech101, LabelMe, SUN09, and VOC2007. This dataset includes 10,729 images of dimension (3, 224, 224) with 5 classes.

- **Office-Home (Venkateswara et al., 2017)** is a 65-class dataset, including domains: art, clipart, product, and real. VLCS includes 10,729 images of dimension (3, 224, 244). (License)

- **Tiny-ImageNet-C** is a 200-class dataset, including 15 corrupt types. Tiny-ImageNet-C includes 150,000 images of dimension (3, 224, 244). Since the class number 200 is less than ImageNet (1000), the model's last layer classifier needs to be adapted. In this work, we use the brightness corruption domain to adapt.

In the source pretraining phase, we adopt the most ImageNet-like domain as our source domain. For PACS and Office-Home, we use domains "photos" and "real" as the source domains, respectively, while for VLCS, Caltech101 is assigned to apply the source pretraining. We freeze the random seeds to generate the sample indices order for the two test data streams, namely, the domain-wise data stream and the random data stream.

For PACS, the domain-wise data stream inputs samples from domain art, cartoons, to sketches, while we shuffle all samples from these three domains in the random data stream. For VLCS, we stream the domains in the order: LabelMe, SUN09, and VOC2007, as the domain-wise data stream. For Office-Home, the domain-wise data stream order becomes art, clipart, and product.

### G.2    TRAINING AND OPTIMIZATION SETTINGS

In this section, we extensively discuss the model architectures, optimization settings, and method settings.

#### G.2.1    ARCHITECTURES

**PACS & VLCS.** We adopt ResNet-18 as our model encoder followed by a linear classifier. The initial parameters of ResNet-18 are ImageNet pre-trained weights. In our experiment, we remove the Dropout layer since we empirically found that using the Dropout layer might degrade the optimization process when the sample number is small. The specific implementation of the network is closely aligned with the implementation in DomainBed (Gulrajani and Lopez-Paz, 2020).

**Office-Home.** We employ ResNet-50 as our model encoder for Office-Home. Except for the architecture, the other model settings are aligned with the ResNet-18.

**Tiny-ImageNet-C** ResNet-18 is adapted from ImageNet to Tiny-ImageNet-C by training the last linear layer.

#### G.2.2    TRAINING & OPTIMIZATION

In this section, we describe the training configurations for both the source domain pre-training and test-time adaptation procedures.

**Source domain pre-training.** For the PACS and VLCS datasets, models are fine-tuned on the selected source domains for 3,000 iterations. The Adam optimizer is utilized with a learning rate

of $10^{-4}$. In contrast, for the Office-Home dataset, the model is fine-tuned for a longer duration of 10,000 iterations with a slightly adjusted learning rate of $5 \times 10^{-5}$.

**Test-time adaptation.** For test-time adaptation across PACS and VLCS, the pre-trained source model is further fine-tuned using the SGD optimizer with a learning rate of $10^{-3}$. While on Office-Home and Tiny-ImageNet-C, a learning rate of $10^{-4}$ is adopted. For all TTA baselines, barring specific exceptions, we faithfully adhere to the original implementation settings. A noteworthy exception is the EATA method, which requires a cosine similarity threshold. The default threshold of the original EATA implementation was not suitable for the three datasets used in our study, necessitating an adjustment. We empirically set this threshold to $0.5$ for training. Unlike Tent and SAR, which only require the optimization of batch normalization layers (Santurkar et al., 2018), SimATTA allows the training of all parameters in the networks. In experiments, we use a tolerance count (tol) to control the training process. SimATTA will stop updating once the loss does not descrease for more than 5 steps. However, for Tiny-ImageNet-C, SimATTA uses 'steps=10' for time comparisons since other methods apply at most 10 steps.

### G.2.3 METHOD SETTINGS

**Tent.** In our experiments, we apply the official implementation of Tent[1]. Specifically, we evaluate Tent with 1 test-time training step and 10 steps, respectively.

**EATA.** Our EATA implementation follows its official code[2]. In our experiments, EATA has 2000 fisher training samples, $E_0 = 0.4 \times \log(\# \text{class})$, $\epsilon < 0.5$.

**CoTTA.** For CoTTA, we strictly follow all the code and settings from its official implementation[3].

**SAR.** With SAR's official implementation[4], we set $E_0 = 0.4 \times \log(\# \text{class})$ and $e_0 = 0.1$ in our experiments.

**ADA baselines.** For ADA baselines, we follow the architecture of the official implementation of CLUE (Prabhu et al., 2021)[5].

**SimATTA Implementation.** Our implementation largely involves straightforward hyperparameter settings. The higher entropy bound $e_h = 10^{-2}$ should exceed the lower entropy bound $e_l$, but equal values are acceptable. Empirically, the lower entropy bound $e_l$ can be set to $10^{-3}$ for VLCS and Office-Home, or $10^{-4}$ for PACS. The choice of $e_l$ is largely dependent on the number of source-like samples obtained. A lower $e_l$ may yield higher-accuracy low-entropy samples, but this could lead to unstable training due to sample scarcity.

Though experimentation with different hyperparameters is encouraged, our findings suggest that maintaining a non-trivial number of low-entropy samples and setting an appropriate $\lambda_0$ are of primary importance. If $\lambda_0 < 0.5$, CF may ensue, which may negate any potential improvement.

Regarding the management of budgets, numerous strategies can be adopted. In our experiments, we utilized a simple hyperparameter $k$, varying from 1 to 3, to regulate the increasing rate of budget consumption. This strategy is fairly elementary and can be substituted by any adaptive techniques.

### G.3 SOFTWARE AND HARDWARE

We conduct our experiments with PyTorch (Paszke et al., 2019) and scikit-learn (Pedregosa et al., 2011) on Ubuntu 20.04. The Ubuntu server includes 112 Intel(R) Xeon(R) Gold 6258R CPU @2.70GHz, 1.47TB memory, and NVIDIA A100 80GB PCIe graphics cards. The training process costs graphics memory less than 10GB, and it requires CPU computational resources for scikit-learn K-Means clustering calculations. Our implementation also includes a GPU-based PyTorch K-Means method for transferring calculation loads from CPUs to GPUs. However, for consistency, the results of our experiments are obtained with the original scikit-learn K-Means implementation.

---

[1] https://github.com/DequanWang/tent
[2] https://github.com/mr-eggplant/EATA
[3] https://github.com/qinenergy/cotta
[4] https://github.com/mr-eggplant/SAR
[5] https://github.com/virajprabhu/CLUE

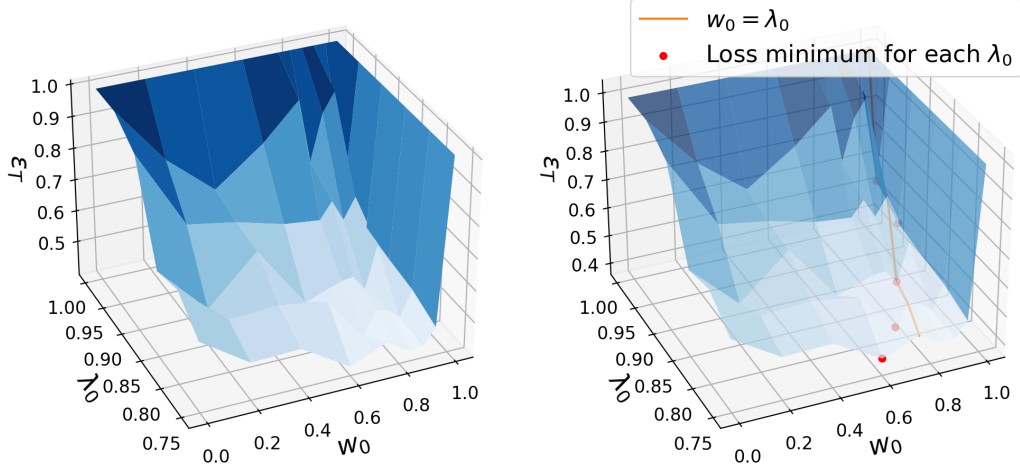

Figure 8: **Target loss surface on 2000 samples without source pre-training.** The red points denote the loss minimum for a fixed $\lambda_0$. The orange line denote the place where $w_0 = \lambda_0$.

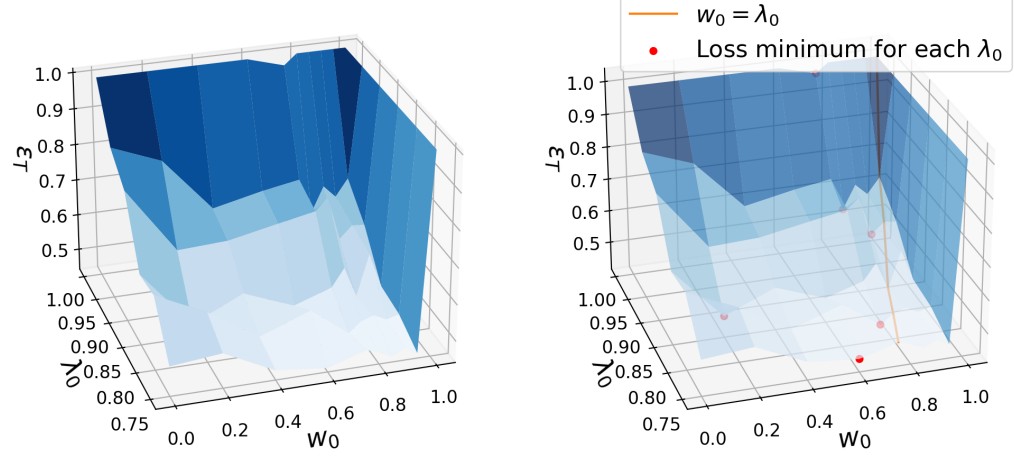

Figure 9: **Target loss surface on 2000 samples with source pre-training.**

## H EMPIRICAL VALIDATIONS FOR THEORETICAL ANALYSIS

In this section, we undertake empirical validation of our learning theory, which encompasses multiple facets awaiting verification. In contemporary computer vision fields, pre-trained models play a pivotal role, and performance would significantly decline without the use of pre-trained features. The learning theory suggests that given the vast VC-dimension of complete ResNets, without substantial data samples, the training error cannot be theoretically tight-bounded. However, we show empirically in the following experiments that fine-tuning pre-trained models is behaviorally akin to training a model with a low VC-dimension.

**Training on 2000 Samples Without Source Domain Pre-training.** For an ImageNet pre-trained ResNet-18 model, we trained it using 2000 samples from the PACS dataset. To ascertain the optimal value $w_0^*$ in Equation 4, we trained multiple models for different $w_0$ and $\lambda_0$ pairings. For each pair, we derived the target domain loss (from art, cartoons, and sketches) post-training and plotted this loss on the z-axis. With $w_0$ and $\lambda_0$ serving as the xy-axes, we drafted the target domain loss $\epsilon_T$ surface in Figure 8. As the results show, given a $\lambda_0$, the optimal $w_0^*$ typically aligns with the line $\lambda_0 = w_0$, with a slight downward shift, which aligns with Equation 4.

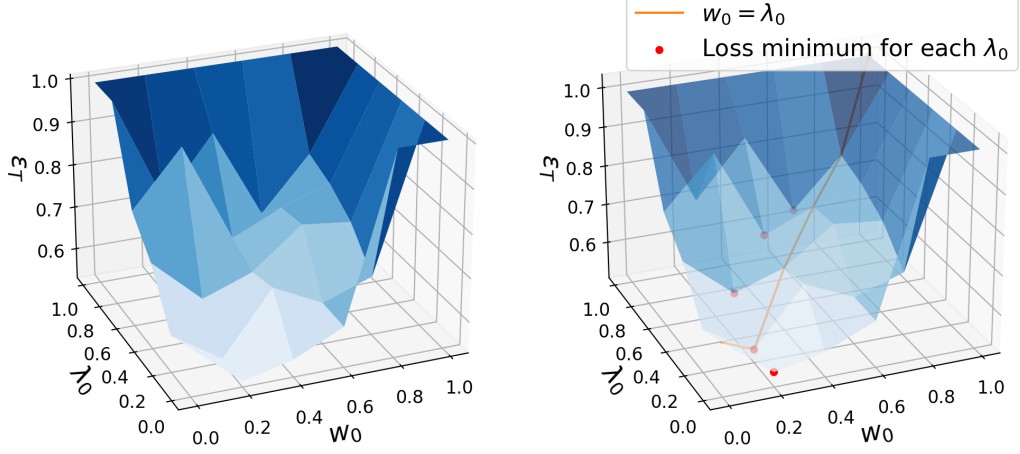

Figure 10: **Target loss surface on 500 samples with source pre-training.**

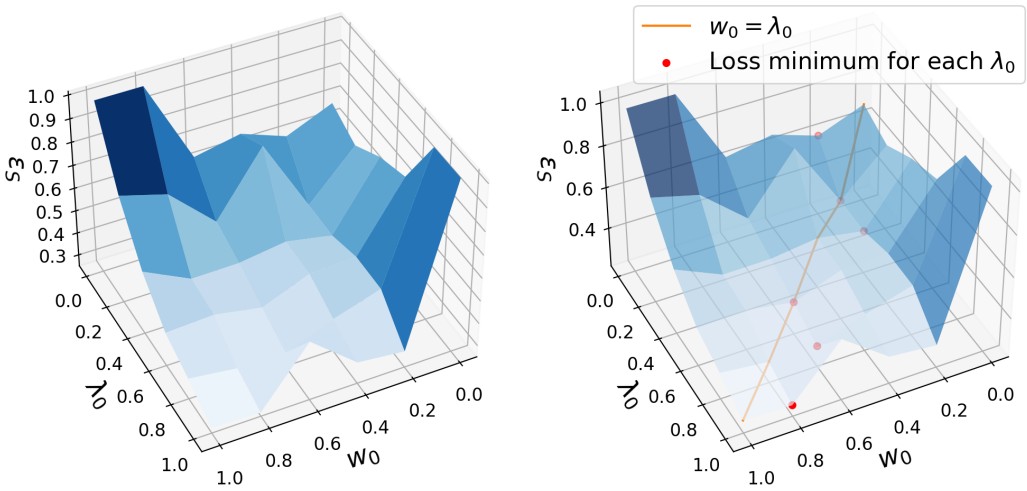

Figure 11: **Source loss surface on 500 samples with source pre-training.**

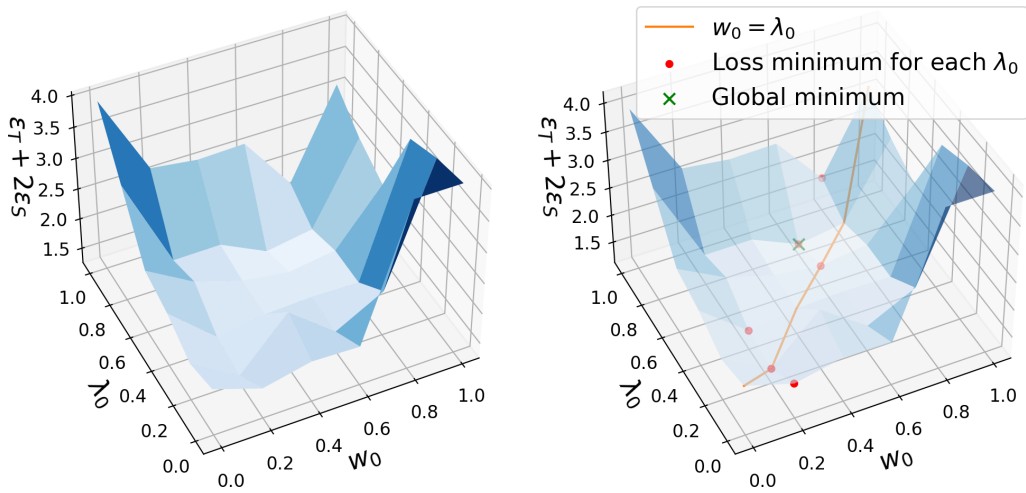

Figure 12: **Target and source loss surface on 500 samples with source pre-training.**

Table 6: **TTA comparisons on Office-Home.** This table includes the two data stream settings mentioned in the dataset setup and reports performances in accuracy. Results that outperform all TTA baselines are highlighted in **bold** font. N/A denotes the adaptations are not applied on the source domain.

| Office-Home | Domain-wise data stream | | | | Post-adaptation | | | | Random data stream | | | | Post-adaptation | | | |
|---|---|---|---|---|---|---|---|---|---|---|---|---|---|---|---|---|
| | R | →A→ | →C→ | →P | R | A | C | P | 1 | 2 | 3 | 4 | R | A | C | P |
| BN w/o adapt | 93.78 | 42.93 | 37.62 | 59.90 | 93.78 | 42.93 | 37.62 | 59.90 | 46.82 | 46.82 | 46.82 | 46.82 | 93.78 | 42.93 | 37.62 | 59.90 |
| BN w/ adapt | 92.38 | 49.69 | 39.43 | 63.53 | 92.38 | 49.69 | 39.43 | 63.53 | 50.88 | 50.88 | 50.88 | 50.88 | 92.38 | 49.69 | 39.43 | 63.53 |
| Tent (steps=1) | N/A | 49.61 | 39.31 | 63.87 | 92.47 | 49.57 | 39.89 | 63.89 | 49.95 | 50.27 | 50.23 | 52.06 | 92.40 | 49.24 | 39.68 | 63.98 |
| Tent (steps=10) | N/A | 49.61 | 39.04 | 61.41 | 87.08 | 44.79 | 38.37 | 60.49 | 50.05 | 49.31 | 48.74 | 47.79 | 85.31 | 42.85 | 37.89 | 58.71 |
| EATA | N/A | 49.65 | 39.04 | 63.53 | 91.60 | 49.61 | 38.65 | 63.48 | 49.73 | 50.27 | 49.45 | 51.07 | 91.05 | 49.11 | 38.26 | 62.99 |
| CoTTA | N/A | 49.61 | 38.76 | 61.84 | 87.81 | 44.95 | 35.92 | 59.04 | 49.84 | 49.84 | 48.95 | 50.43 | 86.99 | 43.68 | 34.73 | 57.56 |
| SAR (steps=1) | N/A | 49.65 | 39.24 | 63.53 | 92.45 | 49.73 | 39.36 | 63.69 | 49.84 | 50.05 | 49.91 | 51.67 | 92.38 | 49.57 | 39.50 | 63.87 |
| SAR (steps=10) | N/A | 49.53 | 38.81 | 61.50 | 88.94 | 46.15 | 37.04 | 59.41 | 50.09 | 50.30 | 49.77 | 49.22 | 89.14 | 46.23 | 36.31 | 59.45 |
| SimATTA ($\mathcal{B} \leq 300$) | N/A | **56.20** | **48.38** | **71.66** | **95.75** | **60.07** | **52.62** | **74.70** | **58.57** | **60.88** | **62.91** | **63.67** | **95.89** | **62.01** | **54.98** | **74.70** |
| SimATTA ($\mathcal{B} \leq 500$) | N/A | **58.71** | **51.11** | **74.36** | **96.03** | **62.05** | **57.41** | **76.98** | **58.85** | **62.63** | **63.41** | **64.31** | **95.91** | **63.78** | **57.87** | **77.09** |

**Training on 2000 Samples with Source Domain Pre-training.** To further assess the effects of source pre-training, we repeated the same experiment on a source pre-trained ResNet-18. The results are depicted in Figure 9. This experiment provides empirical guidance on selecting $w_0$ in source domain pre-trained situations. The findings suggest that the optimal $w_0^*$ non-trivially shifts away from the line $\lambda_0 = w_0$ towards lower-value regions. Considering the source pre-training process as using a greater quantity of source domain samples, it implies that when the number of source samples greatly exceeds target samples, a lower $w_0$ can enhance target domain results.

**Training on 500 Samples with Source Domain Pre-training.** We proceed to fine-tune the source domain pre-trained ResNet-18 using only 500 samples, thereby simulating active TTA settings. We train models with various $w_0$ and $\lambda_0$ pairings, then graph the target domain losses, source domain losses, and the combined losses. As shown in Figure 10, the target losses still comply with our theoretical deductions where the local minima are close to the line $\lambda_0 = w_0$ and marginally shift towards lower values. Considering the challenge of CF, the source domain results in Figure 11 suggest a reverse trend compared to the target domain, where lower $\lambda_0$ and $w_0$ values yield superior target domain results but inferior source domain results. Thus, to curb CF, the primary strategy is to maintain a relatively higher $\lambda_0$. When considering both target and source domains, a balance emerges as depicted in Figure 12. The global minimum is located in the middle region, demonstrating the trade-off between the target domain and source domain performance.

# I   ADDITIONAL EXPERIMENT RESULTS

In this section, we provide additional experiment results. The Office-Home results and ablation studies will be presented in a similar way as the main paper. In the full results Sec. I.3, we will post more detailed experimental results with specific budget numbers and intermediate performance during the test-time adaptation.

Table 7: **Comparisons to ADA baselines on Office-Home.** The source domain is denoted as "(S)" in the table. Results are average accuracies with standard deviations).

| Office-Home | R (S) | A | C | P |
|---|---|---|---|---|
| Random ($\mathcal{B} = 300$) | 95.04 (0.20) | 57.54 (1.16) | 53.43 (1.17) | 73.46 (0.97) |
| Entropy ($\mathcal{B} = 300$) | 94.39 (0.49) | 61.21 (0.71) | 56.53 (0.71) | 72.31 (0.28) |
| Kmeans ($\mathcal{B} = 300$) | 95.09 (0.14) | 57.37 (0.90) | 51.74 (1.34) | 71.81 (0.39) |
| CLUE ($\mathcal{B} = 300$) | 95.20 (0.23) | 60.18 (0.98) | 58.05 (0.43) | 73.72 (0.70) |
| Ours ($\mathcal{B} \leq 300$) | 95.82 (0.07) | 61.04 (0.97) | 53.80 (1.18) | 74.70 (0.00) |

## I.1 RESULTS ON OFFICE-HOME

We conduct experiments on Office-Home and get the test-time performances and post-adaptation performances for two data streams. As shown in Tab. 6, SimATTA can outperform all TTA baselines with huge margins. Compared to ADA baselines under the source-free settings, as shown in Tab. 7, SimATTA obtains comparable results.

## I.2 ABLATION STUDIES

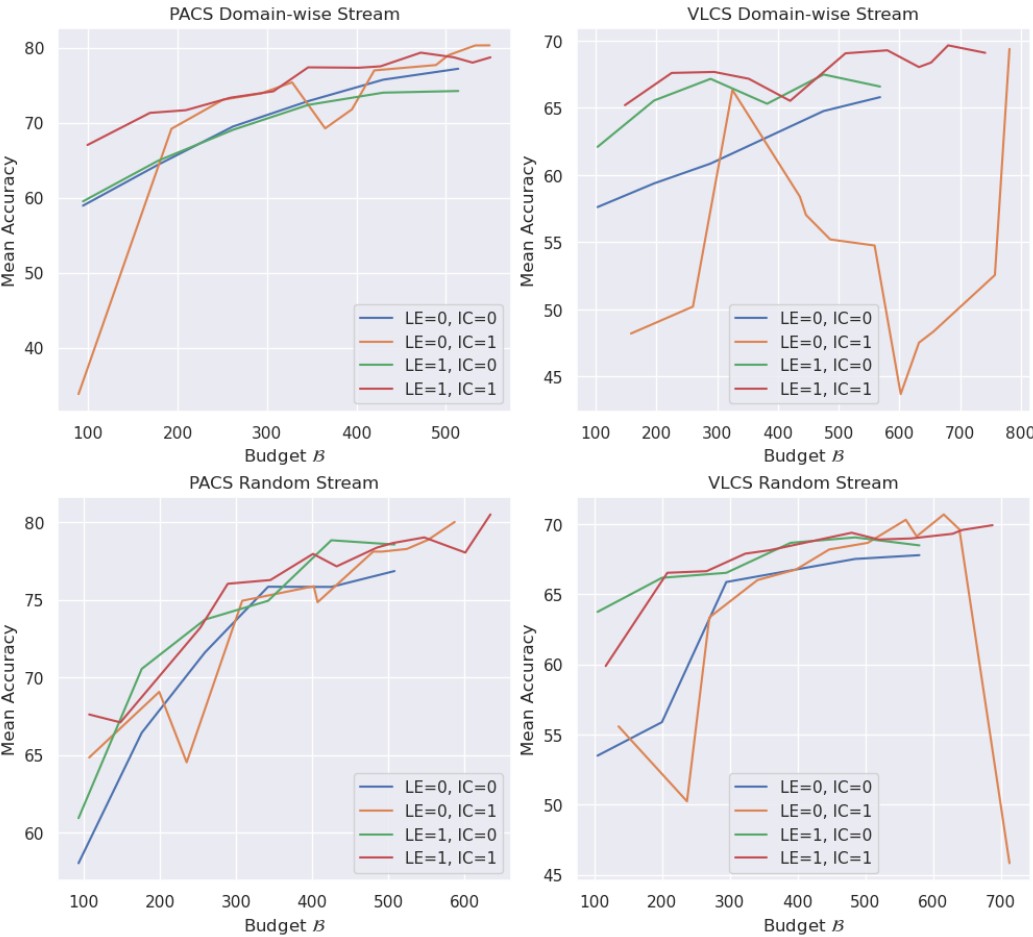

Figure 13: **Ablation study on PACS and VLCS.** "IC=0" denotes removing incremental clustering (IC) selection. "LE=0" denotes removing the low-entropy (LE) sample training. Domain-wise stream and random stream are applied on first and second rows, respectively. The accuracy values are averaged across all splits/domains.

In this section, we explore three variations of our method to examine the individual impacts of its components. The first variant replaces the incremental clustering selection with entropy selection,

where only the samples with the highest entropy are chosen. The second variant eliminates low-entropy sample training. The third variation combines the first and second variants. We perform this ablation study on the PACS and VLCS as outlined in Fig. 13. We denote the use of incremental clustering (IC) and low-entropy training (LE) respectively as IC=1 and LE=1.

The experiments essentially reveals the effectiveness of incremental clustering and low-entropy-sample training. As we have detailed in Sec. 3.2, these techniques are designed to to select informative samples, increase distribution coverage, and mitigate catastrophic forgetting. These designs appositely serve the ATTA setting where the oracle has costs and the budget is limited. Therefore, their effectiveness is prominent particularly when the budget is small. As the results show, when the budget $\mathcal{B} \leq 100$ or $\mathcal{B} \leq 300$, removing the components observably impairs performances. When $\mathcal{B}$ gets large, more active samples cover a larger distribution; thus the performance gap from random selection and informative selection gets smaller. In the extreme case where $\mathcal{B} \to \infty$, all samples are selected and thus the superiority of our meticulously-designed techniques are not manifested.

Specifically, our analysis yields several insights. First, SimATTA (LE=1, IC=1) comprehensively outperforms other variants on both datasets, different streams, and different budgets. Second, variants without low-entropy training (LE=0, IC=0/1) easily fail to produce stable results (e.g., domain-wise stream in VLCS). Third, SimATTA's performance surpasses this variant on PACS's domain-wise stream clearly especially when the budgets are low. This indicates these variants fail to retrieve the most informative style shift (PACS's shifts) samples, which implies the advantage of incremental clustering when the budget is tight.

In addition, these results show that IC has its unique advantage on *domain-wise streams* where distributions change abruptly instead of *random streams*. Therefore, compared to PACS's domain-wise stream results, the reason for the smaller performance improvement of SimATTA over the variant (LE=1, IC=0) on VLCS's domain-wise stream is that images in VLCS are all photos that do not include those severe style shifts in PACS (*i.e.*, art, cartoons, and sketches). That is, when the shift is not severe, we don't need IC to cover very different distributions, and selecting samples using entropy can produce good results. In brief, IC is extraordinary for severe distribution shifts and quick adaptation.

It is worth mentioning that low budget comparison is essential to show the informative sample retrieval ability, since as the budget increases, all AL techniques will tend to perform closely.

## I.3 COMPLETE EXPERIMENT RESULTS

We provide complete experimental results in this section. As shown in Tab. 8, we present the full results for two data streams. The test-time adaptation accuracies are shown in the "Current domain" row, while the "Budgets" row denotes the used budget by the end of the domain. The rest four rows denote the four domain test results by the end of the real-time adaptation of the current domain, where the first column results are the test accuracy before the test-time adaptation phase. N/A represents "do not apply".

Table 8: **Tent (steps=1) on PACS.**

| Tent (steps=1) | Domain-wise data stream | | | | Random data stream | | | |
|---|---|---|---|---|---|---|---|---|
| | P | →A→ | →C→ | →S | 1→ | →2→ | →3→ | →4→ |
| Current domain | N/A | 67.29 | 64.59 | 44.67 | 56.35 | 54.09 | 51.83 | 48.58 |
| Budgets | N/A | N/A | N/A | N/A | N/A | N/A | N/A | N/A |
| P | 99.70 | 98.68 | 98.38 | 97.60 | 98.56 | 98.08 | 97.72 | 97.19 |
| A | 59.38 | 69.09 | 68.95 | 66.85 | 68.07 | 67.33 | 65.58 | 63.53 |
| C | 28.03 | 64.04 | 65.19 | 64.08 | 64.85 | 65.19 | 62.97 | 60.75 |
| S | 42.91 | 53.65 | 47.39 | 42.58 | 54.57 | 49.83 | 44.13 | 41.56 |

## J CHALLENGES AND PERSPECTIVES

Despite advancements, test-time adaptation continues to pose considerable challenges. As previously discussed, without supplementary information and assumptions, the ability to guarantee model generalization capabilities is limited. However, this is not unexpected given that recent progress

Table 9: **Tent (steps=10) on PACS.**

| Tent (steps=10) | Domain-wise data stream | | | | Random data stream | | | |
|---|---|---|---|---|---|---|---|---|
| | P | →A→ | →C→ | →S | 1→ | →2→ | →3→ | →4→ |
| Current domain | N/A | 67.38 | 57.85 | 20.23 | 47.36 | 31.01 | 22.84 | 20.33 |
| Budgets | N/A | N/A | N/A | N/A | N/A | N/A | N/A | N/A |
| P | 99.70 | 95.45 | 87.43 | 62.63 | 93.83 | 81.32 | 65.39 | 50.78 |
| A | 59.38 | 64.94 | 55.03 | 34.52 | 55.32 | 40.28 | 28.27 | 23.68 |
| C | 28.03 | 55.89 | 56.70 | 40.57 | 54.52 | 39.68 | 27.22 | 20.95 |
| S | 42.91 | 36.96 | 26.27 | 13.59 | 32.25 | 23.16 | 20.95 | 19.62 |

Table 10: **EATA on PACS.**

| EATA | Domain-wise data stream | | | | Random data stream | | | |
|---|---|---|---|---|---|---|---|---|
| | P | →A→ | →C→ | →S | 1→ | →2→ | →3→ | →4→ |
| Current domain | N/A | 67.04 | 64.72 | 50.27 | 57.31 | 56.06 | 58.17 | 59.78 |
| Budgets | N/A | N/A | N/A | N/A | N/A | N/A | N/A | N/A |
| P | 99.70 | 98.62 | 98.50 | 98.62 | 98.68 | 98.62 | 98.50 | 98.62 |
| A | 59.38 | 68.90 | 68.16 | 66.50 | 68.65 | 68.95 | 69.34 | 69.63 |
| C | 28.03 | 63.74 | 65.36 | 62.46 | 65.19 | 66.00 | 65.57 | 65.70 |
| S | 42.91 | 54.01 | 52.89 | 48.18 | 55.71 | 55.64 | 54.09 | 54.26 |

Table 11: **CoTTA on PACS.**

| CoTTA | Domain-wise data stream | | | | Random data stream | | | |
|---|---|---|---|---|---|---|---|---|
| | P | →A→ | →C→ | →S | 1→ | →2→ | →3→ | →4→ |
| Current domain | N/A | 65.48 | 62.12 | 53.17 | 56.06 | 54.33 | 57.16 | 57.42 |
| Budgets | N/A | N/A | N/A | N/A | N/A | N/A | N/A | N/A |
| P | 99.70 | 98.68 | 98.62 | 98.62 | 98.62 | 98.62 | 98.56 | 98.62 |
| A | 59.38 | 65.82 | 65.87 | 65.48 | 66.02 | 65.87 | 66.31 | 65.97 |
| C | 28.03 | 62.63 | 63.05 | 63.10 | 63.01 | 62.88 | 63.01 | 62.97 |
| S | 42.91 | 53.88 | 54.03 | 53.78 | 54.67 | 55.31 | 55.10 | 54.62 |

Table 12: **SAR (steps=1) on PACS.**

| SAR (steps=1) | Domain-wise data stream | | | | Random data stream | | | |
|---|---|---|---|---|---|---|---|---|
| | P | →A→ | →C→ | →S | 1→ | →2→ | →3→ | →4→ |
| Current domain | N/A | 66.75 | 63.82 | 49.58 | 56.78 | 56.35 | 56.68 | 56.70 |
| Budgets | N/A | N/A | N/A | N/A | N/A | N/A | N/A | N/A |
| P | 99.70 | 98.68 | 98.50 | 98.32 | 98.74 | 98.56 | 98.50 | 98.44 |
| A | 59.38 | 68.02 | 68.07 | 66.94 | 67.87 | 68.65 | 68.55 | 68.16 |
| C | 28.03 | 62.84 | 64.97 | 62.93 | 63.82 | 64.89 | 64.46 | 64.38 |
| S | 42.91 | 53.47 | 52.07 | 45.74 | 54.92 | 55.46 | 53.68 | 52.53 |

Table 13: **SAR (steps=10) on PACS.**

| SAR (steps=10) | Domain-wise data stream | | | | Random data stream | | | |
|---|---|---|---|---|---|---|---|---|
| | P | →A→ | →C→ | →S | 1→ | →2→ | →3→ | →4→ |
| Current domain | N/A | 69.38 | 68.26 | 49.02 | 53.51 | 51.15 | 51.78 | 45.60 |
| Budgets | N/A | N/A | N/A | N/A | N/A | N/A | N/A | N/A |
| P | 99.70 | 98.20 | 95.39 | 96.47 | 97.13 | 97.78 | 97.72 | 94.13 |
| A | 59.38 | 72.36 | 66.60 | 62.16 | 62.74 | 64.94 | 66.11 | 56.64 |
| C | 28.03 | 63.44 | 68.30 | 56.19 | 59.77 | 61.73 | 62.03 | 56.02 |
| S | 42.91 | 53.37 | 44.59 | 54.62 | 41.00 | 49.66 | 48.79 | 36.37 |

Table 14: **SimATTA** ($\mathcal{B} \leq 300$) **on PACS.**

| SimATTA ($\mathcal{B} \leq 300$) | Domain-wise data stream | | | | Random data stream | | | |
|---|---|---|---|---|---|---|---|---|
| | P | →A→ | →C→ | →S | 1→ | →2→ | →3→ | →4→ |
| Current domain | N/A | 76.86 | 70.90 | 75.39 | 69.47 | 76.49 | 82.45 | 82.22 |
| Budgets | N/A | 75 | 145 | 223 | 66 | 142 | 203 | 267 |
| P | 99.70 | 98.44 | 98.86 | 98.80 | 97.96 | 98.68 | 99.04 | 98.98 |
| A | 59.38 | 80.71 | 82.32 | 84.47 | 73.97 | 80.52 | 81.10 | 84.91 |
| C | 28.03 | 48.12 | 82.00 | 82.25 | 72.35 | 81.06 | 83.36 | 83.92 |
| S | 42.91 | 32.78 | 56.25 | 81.52 | 79.49 | 83.10 | 84.78 | 86.00 |

Table 15: **SimATTA** ($\mathcal{B} \leq 500$) **on PACS.**

| SimATTA ($\mathcal{B} \leq 500$) | Domain-wise data stream | | | | Random data stream | | | |
|---|---|---|---|---|---|---|---|---|
| | P | →A→ | →C→ | →S | 1→ | →2→ | →3→ | →4→ |
| Current domain | N/A | 77.93 | 76.02 | 76.30 | 68.46 | 78.22 | 80.91 | 85.49 |
| Budgets | N/A | 121 | 230 | 358 | 102 | 221 | 343 | 425 |
| P | 99.70 | 98.92 | 98.86 | 98.62 | 98.20 | 99.46 | 99.10 | 99.16 |
| A | 59.38 | 87.01 | 87.60 | 88.33 | 73.39 | 79.20 | 84.91 | 86.67 |
| C | 28.03 | 54.78 | 83.96 | 83.49 | 68.43 | 74.40 | 84.22 | 84.77 |
| S | 42.91 | 46.37 | 63.53 | 83.74 | 81.34 | 81.04 | 86.66 | 87.71 |

Table 16: **Tent (steps=1) on VLCS.**

| Tent (steps=1) | Domain-wise data stream | | | | Random data stream | | | |
|---|---|---|---|---|---|---|---|---|
| | C | →L→ | →S→ | →V | 1→ | →2→ | →3→ | →4→ |
| Current domain | N/A | 38.55 | 34.40 | 53.88 | 44.85 | 44.29 | 47.38 | 44.98 |
| Budgets | N/A | N/A | N/A | N/A | N/A | N/A | N/A | N/A |
| C | 100.00 | 84.81 | 85.44 | 84.73 | 84.95 | 85.16 | 85.80 | 85.30 |
| L | 33.55 | 40.02 | 43.11 | 43.86 | 39.68 | 41.98 | 43.11 | 43.49 |
| S | 41.10 | 33.39 | 35.41 | 33.61 | 36.29 | 37.90 | 38.27 | 37.81 |
| V | 49.08 | 53.20 | 54.06 | 53.11 | 53.76 | 54.18 | 53.76 | 53.35 |

Table 17: **Tent (steps=10) on VLCS.**

| Tent (steps=10) | Domain-wise data stream | | | | Random data stream | | | |
|---|---|---|---|---|---|---|---|---|
| | C | →L→ | →S→ | →V | 1→ | →2→ | →3→ | →4→ |
| Current domain | N/A | 45.41 | 31.44 | 32.32 | 46.13 | 42.31 | 43.51 | 39.48 |
| Budgets | N/A | N/A | N/A | N/A | N/A | N/A | N/A | N/A |
| C | 100.00 | 73.07 | 48.34 | 42.54 | 74.13 | 62.19 | 56.54 | 52.01 |
| L | 33.55 | 46.61 | 38.44 | 37.65 | 44.88 | 45.93 | 43.41 | 40.32 |
| S | 41.10 | 31.75 | 28.82 | 27.79 | 35.37 | 36.14 | 35.28 | 33.64 |
| V | 49.08 | 48.05 | 40.14 | 33.12 | 50.50 | 44.49 | 42.48 | 40.37 |

Table 18: **EATA on VLCS.**

| EATA | Domain-wise data stream | | | | Random data stream | | | |
|---|---|---|---|---|---|---|---|---|
| | C | →L→ | →S→ | →V | 1→ | →2→ | →3→ | →4→ |
| Current domain | N/A | 37.24 | 33.15 | 52.58 | 43.77 | 42.48 | 43.34 | 41.55 |
| Budgets | N/A | N/A | N/A | N/A | N/A | N/A | N/A | N/A |
| C | 100.00 | 85.16 | 85.02 | 84.10 | 84.73 | 84.52 | 84.10 | 83.32 |
| L | 33.55 | 37.16 | 37.24 | 37.69 | 37.09 | 36.78 | 36.90 | 36.67 |
| S | 41.10 | 33.39 | 33.49 | 32.39 | 33.33 | 32.54 | 31.84 | 31.47 |
| V | 49.08 | 51.87 | 52.16 | 52.49 | 52.07 | 52.43 | 52.64 | 52.55 |

Table 19: **CoTTA on VLCS.**

| CoTTA | Domain-wise data stream | | | | Random data stream | | | |
|---|---|---|---|---|---|---|---|---|
| | C | →L→ | →S→ | →V | 1→ | →2→ | →3→ | →4→ |
| Current domain | N/A | 37.39 | 32.54 | 52.25 | 43.69 | 42.14 | 43.21 | 42.32 |
| Budgets | N/A | N/A | N/A | N/A | N/A | N/A | N/A | N/A |
| C | 100.00 | 81.55 | 81.98 | 82.12 | 82.61 | 82.47 | 82.12 | 81.98 |
| L | 33.55 | 37.20 | 37.91 | 37.65 | 38.48 | 38.22 | 38.40 | 37.99 |
| S | 41.10 | 30.71 | 32.78 | 33.12 | 34.00 | 33.70 | 33.97 | 33.52 |
| V | 49.08 | 52.01 | 52.64 | 52.90 | 53.64 | 53.14 | 53.08 | 53.23 |

Table 20: **SAR (steps=1) on VLCS.**

| SAR (steps=1) | Domain-wise data stream | | | | Random data stream | | | |
|---|---|---|---|---|---|---|---|---|
| | C | →L→ | →S→ | →V | 1→ | →2→ | →3→ | →4→ |
| Current domain | N/A | 36.18 | 34.43 | 52.46 | 43.64 | 43.04 | 44.20 | 41.93 |
| Budgets | N/A | N/A | N/A | N/A | N/A | N/A | N/A | N/A |
| C | 100.00 | 84.31 | 84.17 | 83.96 | 85.09 | 85.23 | 85.23 | 85.09 |
| L | 33.55 | 35.62 | 38.29 | 39.72 | 38.55 | 39.34 | 40.21 | 40.70 |
| S | 41.10 | 33.24 | 36.41 | 36.53 | 34.37 | 35.62 | 36.29 | 36.44 |
| V | 49.08 | 51.75 | 52.61 | 52.37 | 52.90 | 52.75 | 53.05 | 53.02 |

Table 21: **SAR (steps=10) on VLCS.**

| SAR (steps=10) | Domain-wise data stream | | | | Random data stream | | | |
|---|---|---|---|---|---|---|---|---|
| | C | →L→ | →S→ | →V | 1→ | →2→ | →3→ | →4→ |
| Current domain | N/A | 35.32 | 34.10 | 51.66 | 43.56 | 42.05 | 42.53 | 41.16 |
| Budgets | N/A | N/A | N/A | N/A | N/A | N/A | N/A | N/A |
| C | 100.00 | 83.96 | 83.04 | 82.12 | 84.03 | 84.24 | 85.23 | 85.09 |
| L | 33.55 | 34.07 | 35.92 | 41.49 | 39.53 | 38.37 | 37.65 | 37.58 |
| S | 41.10 | 31.93 | 34.89 | 33.94 | 35.19 | 32.94 | 33.88 | 33.12 |
| V | 49.08 | 51.33 | 51.51 | 53.08 | 52.78 | 52.34 | 51.78 | 52.01 |

Table 22: **SimATTA ($\mathcal{B} \leq 300$) on VLCS.**

| SimATTA ($\mathcal{B} \leq 300$) | Domain-wise data stream | | | | Random data stream | | | |
|---|---|---|---|---|---|---|---|---|
| | C | →L→ | →S→ | →V | 1→ | →2→ | →3→ | →4→ |
| Current domain | N/A | 62.61 | 65.08 | 74.38 | 62.33 | 69.33 | 73.20 | 71.93 |
| Budgets | N/A | 79 | 175 | 272 | 71 | 135 | 208 | 262 |
| C | 100.00 | 99.51 | 98.52 | 99.93 | 99.86 | 99.79 | 100.00 | 99.93 |
| L | 33.55 | 68.11 | 69.92 | 69.50 | 62.61 | 66.64 | 68.45 | 69.43 |
| S | 41.10 | 55.24 | 68.89 | 66.67 | 65.54 | 69.29 | 71.79 | 72.46 |
| V | 49.08 | 66.08 | 70.94 | 77.34 | 73.79 | 76.87 | 78.82 | 80.39 |

Table 23: **SimATTA ($\mathcal{B} \leq 500$) on VLCS.**

| SimATTA ($\mathcal{B} \leq 500$) | Domain-wise data stream | | | | Random data stream | | | |
|---|---|---|---|---|---|---|---|---|
| | C | →L→ | →S→ | →V | 1→ | →2→ | →3→ | →4→ |
| Current domain | N/A | 63.52 | 68.01 | 76.13 | 62.29 | 70.45 | 73.50 | 72.02 |
| Budgets | N/A | 113 | 266 | 446 | 107 | 203 | 283 | 356 |
| C | 100.00 | 99.29 | 98.59 | 99.51 | 99.93 | 99.86 | 99.86 | 99.43 |
| L | 33.55 | 62.95 | 70.63 | 70.56 | 66.57 | 67.09 | 67.24 | 70.29 |
| S | 41.10 | 51.31 | 73.83 | 73.10 | 65.33 | 71.79 | 72.91 | 72.55 |
| V | 49.08 | 59.36 | 71.65 | 78.35 | 73.58 | 77.84 | 80.01 | 80.18 |

Table 24: **Tent (steps=1) on Office-Home.**

| Tent (steps=1) | Domain-wise data stream | | | | Random data stream | | | |
|---|---|---|---|---|---|---|---|---|
| | R | →A→ | →C→ | →P | 1→ | →2→ | →3→ | →4→ |
| Current domain | N/A | 49.61 | 39.31 | 63.87 | 49.95 | 50.27 | 50.23 | 52.06 |
| Budgets | N/A | N/A | N/A | N/A | N/A | N/A | N/A | N/A |
| R | 96.44 | 92.33 | 92.36 | 92.47 | 92.38 | 92.45 | 92.45 | 92.40 |
| A | 57.07 | 49.73 | 49.73 | 49.57 | 49.69 | 49.73 | 49.57 | 49.24 |
| C | 44.97 | 39.27 | 39.54 | 39.89 | 39.45 | 39.68 | 39.73 | 39.68 |
| P | 73.15 | 63.60 | 63.66 | 63.89 | 63.60 | 63.82 | 63.93 | 63.98 |

Table 25: **Tent (steps=10) on Office-Home.**

| Tent (steps=10) | Domain-wise data stream | | | | Random data stream | | | |
|---|---|---|---|---|---|---|---|---|
| | R | →A→ | →C→ | →P | 1→ | →2→ | →3→ | →4→ |
| Current domain | N/A | 49.61 | 39.04 | 61.41 | 50.05 | 49.31 | 48.74 | 47.79 |
| Budgets | N/A | N/A | N/A | N/A | N/A | N/A | N/A | N/A |
| R | 96.44 | 91.99 | 89.14 | 87.08 | 92.08 | 90.80 | 88.59 | 85.31 |
| A | 57.07 | 49.94 | 46.77 | 44.79 | 49.44 | 48.21 | 45.69 | 42.85 |
| C | 44.97 | 38.58 | 39.11 | 38.37 | 40.18 | 40.02 | 38.63 | 37.89 |
| P | 73.15 | 63.28 | 61.03 | 60.49 | 64.36 | 63.64 | 61.12 | 58.71 |

Table 26: **EATA on Office-Home.**

| EATA | Domain-wise data stream | | | | Random data stream | | | |
|---|---|---|---|---|---|---|---|---|
| | R | →A→ | →C→ | →P | 1→ | →2→ | →3→ | →4→ |
| Current domain | N/A | 49.65 | 39.04 | 63.53 | 49.73 | 50.27 | 49.45 | 51.07 |
| Budgets | N/A | N/A | N/A | N/A | N/A | N/A | N/A | N/A |
| R | 96.44 | 92.36 | 92.17 | 91.60 | 92.38 | 92.22 | 91.71 | 91.05 |
| A | 57.07 | 49.57 | 49.53 | 49.61 | 49.69 | 49.40 | 49.36 | 49.11 |
| C | 44.97 | 39.08 | 39.01 | 38.65 | 39.27 | 39.01 | 38.42 | 38.26 |
| P | 73.15 | 63.42 | 63.42 | 63.48 | 63.51 | 63.37 | 63.33 | 62.99 |

Table 27: **CoTTA on Office-Home.**

| CoTTA | Domain-wise data stream | | | | Random data stream | | | |
|---|---|---|---|---|---|---|---|---|
| | R | →A→ | →C→ | →P | 1→ | →2→ | →3→ | →4→ |
| Current domain | N/A | 49.61 | 38.76 | 61.84 | 49.84 | 49.84 | 48.95 | 50.43 |
| Budgets | N/A | N/A | N/A | N/A | N/A | N/A | N/A | N/A |
| R | 96.44 | 90.38 | 88.02 | 87.81 | 90.48 | 89.37 | 88.00 | 86.99 |
| A | 57.07 | 48.58 | 45.53 | 44.95 | 47.34 | 46.35 | 44.62 | 43.68 |
| C | 44.97 | 36.66 | 35.58 | 35.92 | 37.55 | 36.40 | 35.44 | 34.73 |
| P | 73.15 | 60.40 | 57.74 | 59.04 | 61.12 | 59.63 | 58.35 | 57.56 |

Table 28: **SAR (steps=1) on Office-Home.**

| SAR (steps=1) | Domain-wise data stream | | | | Random data stream | | | |
|---|---|---|---|---|---|---|---|---|
| | R | →A→ | →C→ | →P | 1→ | →2→ | →3→ | →4→ |
| Current domain | N/A | 49.65 | 39.24 | 63.53 | 49.84 | 50.05 | 49.91 | 51.67 |
| Budgets | N/A | N/A | N/A | N/A | N/A | N/A | N/A | N/A |
| R | 96.44 | 92.38 | 92.31 | 92.45 | 92.40 | 92.36 | 92.36 | 92.38 |
| A | 57.07 | 49.65 | 49.57 | 49.73 | 49.69 | 49.61 | 49.57 | 49.57 |
| C | 44.97 | 39.34 | 39.22 | 39.36 | 39.34 | 39.56 | 39.47 | 39.50 |
| P | 73.15 | 63.51 | 63.51 | 63.69 | 63.60 | 63.71 | 63.71 | 63.87 |

Table 29: **SAR (steps=10) on Office-Home.**

| SAR (steps=10) | Domain-wise data stream | | | | Random data stream | | | |
|---|---|---|---|---|---|---|---|---|
| | R | →A→ | →C→ | →P | 1→ | →2→ | →3→ | →4→ |
| Current domain | N/A | 49.53 | 38.81 | 61.50 | 50.09 | 50.30 | 49.77 | 49.22 |
| Budgets | N/A | N/A | N/A | N/A | N/A | N/A | N/A | N/A |
| R | 96.44 | 92.20 | 92.06 | 88.94 | 92.40 | 92.47 | 91.53 | 89.14 |
| A | 57.07 | 49.40 | 49.77 | 46.15 | 49.81 | 50.02 | 48.91 | 46.23 |
| C | 44.97 | 39.20 | 38.63 | 37.04 | 39.50 | 39.29 | 38.65 | 36.31 |
| P | 73.15 | 63.53 | 62.69 | 59.41 | 64.18 | 64.18 | 62.83 | 59.45 |

Table 30: **SimATTA ($\mathcal{B} \leq 300$) on Office-Home.**

| SimATTA ($\mathcal{B} \leq 300$) | Domain-wise data stream | | | | Random data stream | | | |
|---|---|---|---|---|---|---|---|---|
| | R | →A→ | →C→ | →P | 1→ | →2→ | →3→ | →4→ |
| Current domain | N/A | 56.20 | 48.38 | 71.66 | 58.57 | 60.88 | 62.91 | 63.67 |
| Budgets | N/A | 75 | 187 | 277 | 79 | 147 | 216 | 278 |
| R | 96.44 | 95.43 | 95.43 | 95.75 | 95.91 | 95.96 | 96.01 | 95.89 |
| A | 57.07 | 57.56 | 59.50 | 60.07 | 58.34 | 59.91 | 61.15 | 62.01 |
| C | 44.97 | 42.25 | 52.46 | 52.62 | 51.66 | 52.30 | 54.75 | 54.98 |
| P | 73.15 | 68.84 | 70.13 | 74.70 | 72.45 | 73.10 | 74.50 | 74.70 |

Table 31: **SimATTA ($\mathcal{B} \leq 500$) on Office-Home.**

| SimATTA ($\mathcal{B} \leq 500$) | Domain-wise data stream | | | | Random data stream | | | |
|---|---|---|---|---|---|---|---|---|
| | R | →A→ | →C→ | →P | 1→ | →2→ | →3→ | →4→ |
| Current domain | N/A | 58.71 | 51.11 | 74.36 | 58.85 | 62.63 | 63.41 | 64.31 |
| Budgets | N/A | 107 | 284 | 440 | 126 | 248 | 361 | 467 |
| R | 96.44 | 95.69 | 95.71 | 96.03 | 96.26 | 96.19 | 95.87 | 95.91 |
| A | 57.07 | 61.43 | 61.43 | 62.05 | 58.18 | 61.15 | 61.52 | 63.78 |
| C | 44.97 | 46.41 | 57.73 | 57.41 | 53.17 | 55.14 | 56.79 | 57.87 |
| P | 73.15 | 70.74 | 71.98 | 76.98 | 73.51 | 74.18 | 75.78 | 77.09 |

in deep learning heavily relies on large-scale data. Consequently, two promising paths emerge: establishing credible assumptions and leveraging additional information.

Firstly, developing credible assumptions can lead to comprehensive comparisons across various studies. Given that theoretical guarantees highlight the inherent differences between methods primarily based on the application limits of their assumptions, comparing these assumptions becomes critical. Without such comparative studies, empirical evaluations may lack precise guidance and explanation.

Secondly, while we acknowledge the value of real-world data (observations), discussions surrounding the use of extra information remain pertinent. Considerations include the strategies to acquire this supplementary information and the nature of the additional data needed.

Despite the myriad of works on domain generalization, domain adaptation, and test-time adaptation, a comprehensive survey or benchmark encapsulating the aforementioned comparisons remains an unmet need. Moreover, potential future directions for out-of-distribution generalization extend beyond domain generalization and test-time adaptation. One promising avenue is bridging the gap between causal inference and deep learning, for instance, through causal representation learning.

In conclusion, our hope is that this work not only offers a novel practical setting and algorithm but also illuminates meaningful future directions and research methodologies that can benefit the broader scientific community.

