# OpenReview forum: "Active Test-Time Adaptation: Theoretical Analyses and An Algorithm"
_ICLR.cc/2024/Conference — ICLR 2024 poster_

### Official Review · Reviewer_yeNU · 2023-10-31

**Soundness:** 3 good
**Presentation:** 3 good
**Contribution:** 3 good
**Rating:** 6
**Confidence:** 4

**Summary:**

This paper points out that achieving domain generalization is theoretically impossible without additional information. Therefore, this paper introduces active test-time-training(ATTA), combining active learning with test-time-training, and proposes an effective ATTA algorithm, SimATTA, that innovatively integrates incremental clustering and selective entropy minimization to address catastrophic forgetting and real-time active sample selection issues.

**Strengths:**

（1）This paper innovatively combines active learning with TTA, enhancing performance across test domains. and present sample entropy balancing to avoid catastrophic forgetting.
（2）The paper conducted extensive experiments and compared with the latest state-of-the-art methods on multiple datasets, achieving superior results.
（3）The paper is well-organized, and it provides extensive proofs for the theorems mentioned.

**Weaknesses:**

1. In Section 3.2, the authors state that entropy is essentially a measure of the distribution distance between the model distribution and a test sample, which is not true. While entropy can provide information about the uncertainty of the model, it does not directly measure the distributional distance between the model's distribution and the test samples. This fundamental flaw casts doubt on the proposed method.
2. Table 3 indicates that, in terms of efficiency, SimATTA takes longer than all previous methods, making it less efficient.
3. In Appx. H.2., when B<=500, the performance of SimATTA is close to that of other methods, with no distinct advantage
4. The ablation study is not comprehensive enough to effectively prove the efficacy of both the incremental clustering and selective entropy minimization methods.

**Questions:**

Entropy does not directly measure the distributional distance between the model's distribution and the test sample，so the theoretical foundation presented in the paper is shaky. How do you view this issue?

After reading the rebuttal, I decided to raise my score.

---

> ### Author Response · Authors · 2023-11-19
> **Response to Reviewer yeNU (Part I)**
>
> Dear Reviewer yeNU,
>
> Thank you for your constructive comments and valuable suggestions! We have revised the paper following your suggestions and we provide responses for each concern here.
>
> > W1. In Section 3.2, the authors state that entropy is essentially a measure of the distribution distance between the model distribution and a test sample, which is not true. While entropy can provide information about the uncertainty of the model, it does not directly measure the distributional distance between the model's distribution and the test samples. This fundamental flaw casts doubt on the proposed method.
>
> Thank you for raising this point! We realize the potential misconception here and have revised the section for more rigorous descriptions following your suggestion. However, we would like to clarify that whether entropy directly measures a distributional distance does not impair the theoretical foundation in Sec. 3.2 and the later proposed method.
>
> - **Relationship between low-entropy samples and source samples:** The core point of Sec. 3.2 and mitigating catastrophic forgetting is the **assumption** that the low-entropy sample distribution selected by the source-pretrained-model, $D_{\phi,S}(t)$, serves the model training similarly to source samples in terms of CF.
> When a model assigns low entropy to a sample, this indicates a high level of certainty or confidence in its prediction and can be interpreted as the sample being well-aligned or fitting closely with the model's learned distribution.
> The pre-trained model is well-trained on abundant source data, and thus low entropy can be used as an indicator of a sample closely aligned with the model distribution, in this case, a source-like sample.
> These source-like samples form a distribution that fits/supports the original model predictions. Intuitively, these supports prevent the model from violating the original high-confident predictions, thus maintaining the model's original distribution. This behavior is different from minimizing the distance between the current model parameters and the original model parameters. Instead, these samples only serve as the original model's "anchor points" to avoid catastrophic forgetting. More detailed explanations and clarification are in Appx.D.
> - **Theoretical validity:** With no further condition than the above assumption, the theoretical results in Sec 3.2 and the method design in Sec 4 hold, giving credence to the validity of our paper.
> - **Bridging the theory and practical implementations:** In Table 1, we **empirically validate our assumption** that low-entropy samples, similar to samples from the source distribution, can serve to avoid catastrophic forgetting. This assumption with its empirical validation is the bridge between our theorems and our practical implementations.
>
>
> > W2. Table 3 indicates that, in terms of efficiency, SimATTA takes longer than all previous methods, making it less efficient.
>
> We have revised Sec 5 to add information and avoid misunderstandings.
> - Regarding Table 3, we'd like to clarify that ATTA is of very similar time efficiency with SAR on Tiny-ImageNet-C (time per step). ATTA does not have time-consuming steps, instead the training time is due to the number of interaction steps till convergence. Since SimATTA's training is statistically significant instead of one sample training, several training steps are necessary to make use of information and improve performance. This is a trade-off between effectiveness and efficiency, and we believe the performance improvement is worth this training.
> - **Real-world application:** In our opinion, many real-time applications like robotics, autonomous vehicles, medical imaging adaptations, and user interface personalizations, are not very time-sensitive. For example, retraining a personalized user interface recommendation system may require hours, days, or weeks, but ATTA can fine-tune the model in a second scale, which should be considered a real-time manner. As we have demonstrated (detailed in Appx B.Q8), the speed of ATTA is sufficiently fast for an autopilot system, and the speed requirements of the other applications listed above are safely within the limit of ATTA. Therefore, we believe ATTA is a real-time setting.

---

> > ### Author Response · Authors · 2023-11-19
> > **Response to Reviewer yeNU (Part II)**
> >
> > > W3. In Appx. H.2., when B<=500, the performance of SimATTA is close to that of other methods, with no distinct advantage
> >
> > Thank you for the point! In brief, for this ablation study, it is an expected result when $\mathcal{B}$ is large, and thus comparison under strict budgets is more significant.
> > - To better demonstrate, we extend experiments with more budget selections and provide results as line charts in Appx.I.2 (revised version).
> > - The experiments essentially reveal the effectiveness of incremental clustering and low-entropy-sample training. As we have detailed in Sec. 4, these techniques are designed to select informative samples, increase distribution coverage, and mitigate catastrophic forgetting. These designs appositely serve the ATTA setting where the oracle has costs and the budget is limited. Therefore, their effectiveness is prominent particularly when the budget is small. As the results show, when $\mathcal{B}\le 100$ or $\mathcal{B}\leq 300$, removing the components observably impairs performances. **When $\mathcal{B}$ gets large, more active samples cover a larger distribution; thus the performance gap between random selection and informative selection gets smaller. In the extreme case where $\mathcal{B}→∞$, all samples are selected and thus the superiority of our meticulously-designed techniques is not manifested.**
> >
> > > W4. The ablation study is not comprehensive enough to effectively prove the efficacy of both the incremental clustering and selective entropy minimization methods.
> >
> > Thank you for this advice! We have extended ablation study experiments as detailed in the above question to prove the efficacy of both the incremental clustering and selective entropy minimization methods in Appx. I.2.
> > - The results show that our method components clearly improve the overall stability and performance particularly under strict budget limits and strong domain shifts, which demonstrates the advantage of our design regarding distribution shifts, effectiveness, and robustness towards cost constraints. Please refer to the analyses in Appx. I2 for more details.
> >
> > > Question: Entropy does not directly measure the distributional distance between the model's distribution and the test sample, so the theoretical foundation presented in the paper is shaky. How do you view this issue?
> >
> > - Thank you for the question. We are sorry for this misleading description, but it wouldn't affect the theoretical foundation. We have provided detailed response to this point in the above "W1" and made revisions in Sec.3 and Appx.D, which benefits the rigor of our paper a lot. Please let us know if further discussions are needed.
> >
> >
> > Besides the above points, we also provide further experiments (visualization experiments and extended ablation study) and make efforts to clear the concerns and improve the clarity and organization of our paper in multiple aspects following the advice from other reviewers. Please refer to the revised paper and appendices for details.
> >
> > We sincerely thank you for your time! Hope we have addressed your concerns through practical efforts and shown the significance of our work. We look forward to your reply and further discussions, thanks!
> >
> > Sincerely,
> >
> > Authors

---

> > > ### Author Response · Authors · 2023-11-22
> > > **We would like to hear back from Reviewer yeNU**
> > >
> > > Dear Reviewer yeNU,
> > >
> > > Thanks again for your valuable comments and suggestions, which helps improve our work a lot. Since the end of the discussion period is approaching, we would like to follow up to see if our response addresses your comments. We would really appreciate it if you could check our response at your earliest convenience. Many thanks!
> > >
> > > We provide further experiments including visualization experiments and extended ablation studies, and make efforts to clear the concerns and improve the clarity and organization of our paper in multiple aspects.
> > > We hope that you could reply to our response and revision and reconsider your evaluation if we do have addressed your concerns. Also, please let us know if there are any additional questions or feedback. Thank you!
> > >
> > > Sincerely,
> > >
> > > Authors

---

### Official Review · Reviewer_5EyE · 2023-11-02

**Soundness:** 2 fair
**Presentation:** 3 good
**Contribution:** 2 fair
**Rating:** 5
**Confidence:** 3

**Summary:**

The paper titled "Active Test-Time Adaptation: Theoretical Analyses and An Algorithm" delves into a novel approach for machine learning models that dynamically adapt during test-time  under domain shifts. Traditionally, models rely heavily on heuristic and empirical studies without further adaptation. The authors challenge this convention by introducing a mechanism that allows the model to actively query an oracle (typically a human) during test-time to obtain labels for certain instances. The objective is to enhance the model's performance on the test set by leveraging this limited interaction with the oracle.

The primary contributions of the paper are as follows:

1. Introduction of the Active Test-Time Adaptation (ATTA) framework.
2. A novel algorithm for determining which instances should be queried from the oracle, based on their potential impact on the model's performance.
3. Experimental validation of the ATTA framework on several benchmark datasets, demonstrating significant improvement in performance over non-adaptive baselines.

**Strengths:**

1. The concept of actively adapting a model during test-time based on interactions with an oracle is innovative. This breaks away from the conventional train-test paradigm, paving the way for more dynamic and adaptive models.

2. The authors provide a theoretical foundation for the ATTA framework, making a compelling case for its viability and potential benefits.

3. The proposed method is model-agnostic, meaning it can be applied to a wide range of machine learning algorithms, from simple linear classifiers to complex deep learning architectures.

4. The extensive experiments on benchmark datasets provide strong empirical evidence supporting the effectiveness of the ATTA framework. The improvements over non-adaptive baselines are both statistically significant and practically relevant.

**Weaknesses:**

1. The ATTA framework's effectiveness hinges on the availability and accuracy of an oracle. In real-world scenarios, obtaining such an oracle (especially a human expert) might be challenging, time-consuming, or expensive.

2. While the approach shows promise on benchmark datasets, its scalability to very large datasets or real-world applications remains untested. The computational overhead of deciding which instances to query and updating the model during test-time could be prohibitive in some scenarios.

3. The paper assumes a limited budget of queries to the oracle. In many real-world scenarios, determining this budget or ensuring its strict adherence might be challenging.

4. Continually adapting the model during test-time based on feedback from the oracle could lead to overfitting, especially if the test set is not representative of the broader data distribution.

**Questions:**

See Weaknesses.

---

> ### Author Response · Authors · 2023-11-19
> **Response to Reviewer 5EyE (Part I)**
>
> Dear Reviewer 5EyE,
>
> Thank you for your constructive comments and valuable suggestions! We have made paper revisions following your suggestions and would like to provide responses for each concern here.
>
> > W1. The ATTA framework's effectiveness hinges on the availability and accuracy of an oracle. In real-world scenarios, obtaining such an oracle (especially a human expert) might be challenging, time-consuming, or expensive.
>
> - Reliance on an oracle or expert input is a fundamental aspect of active learning, which is a long-standing field. While obtaining an oracle might be challenging or expensive, the cost should be **weighed against the benefits** of significantly improved model performance and reduced need for vast amounts of labeled data.
> - **Application of active strategies:** In many practical situations, the benefits of active learning outweigh the challenges of obtaining an oracle. For instance, in medical diagnosis, scientific research, or complex engineering problems, the cost and effort of consulting an expert are justified by the significant improvements in model accuracy and efficiency. Technological advancements, such as semi-automated systems or sophisticated algorithms, can also serve as oracles or assist human experts, making the process more feasible and efficient.
> - **Application of ATTA:** ATTA has its specific application scenarios. One potential application direction is for human-machine cooperated semi-automated applications, where we can obtain passive oracle input without extra efforts. One example is autopilot systems we mentioned in our FAQ (Appx B.Q8: What is the potential practical utility of ATTA?).
> - Finally, we regard this point as a general challenge of active learning rather than a drawback of the ATTA framework itself. Therefore, it can be viewed as a consideration/limitation in the application. We specify this discussion in Appx.J to add rigor to the paper.
>
> > W2. While the approach shows promise on benchmark datasets, its scalability to very large datasets or real-world applications remains untested. The computational overhead of deciding which instances to query and updating the model during test-time could be prohibitive in some scenarios.
>
> - **Time complexity:** As in Table 3, ATTA is of very similar time efficiency with TTA method SAR (time per step). ATTA does not have time-consuming steps, instead the training time is due to the number of interaction steps till convergence. This training time is a trade-off for significant improvements in performances.
> - **Real-world application:** In our opinion, many real-time applications like robotics, autonomous vehicles, medical imaging adaptations, and user interface personalizations, are not very time-sensitive. For example, retraining a personalized user interface recommendation system may require hours, days, or weeks, but ATTA can fine-tune the model in a second scale, which should be considered a real-time manner. As we have demonstrated (detailed in Appx B.Q8), the speed of ATTA is sufficiently fast for an autopilot system, and the speed requirements of the other applications listed above are safely within the limit of ATTA. Therefore, we believe ATTA is a real-time setting.
> Rigorously, we acknowledge that the application scopes of ATTA and FTTA are not completely overlapping. FTTA can handle applications that are extremely time-sensitive, while ATTA provides results of much higher quality.
> - Finally, our paper focuses on catastrophic forgetting, domain shift solving, and the establishment of a new setting. We conduct experiments on domain generalization datasets and emphasize the comparisons for different settings. We acknowledge that scalability is not a focus of this paper. Therefore, we consider this valuable suggestion as a possible future direction instead of a weakness, which we discuss in FAQ (Appx.B.Q10: What are not covered by this paper?).

---

> > ### Author Response · Authors · 2023-11-19
> > **Response to Reviewer 5EyE (Part II)**
> >
> > > W3. The paper assumes a limited budget of queries to the oracle. In many real-world scenarios, determining this budget or ensuring its strict adherence might be challenging.
> >
> > - We would like to clarify that we don't have a strict budget limit. The budget limit in our experiments is only for effectiveness observations and comparisons. Incremental clustering is designed to select samples for new **in-coming distributions adaptively** and is affected by the number of centroid increase $k$. In real-world scenarios, we don't need to determine an "optimal" budget **in advance**. In contrast, we provide $k$ according to oracle resource estimations only, which do not need to be accurate **since we can adjust $k$ during adaptations according to oracle resources available**.
> > - Furthermore, during adaptations, incremental clustering can choose the informative samples according to data distributions. Then if the oracle resources are not enough, we can simply label samples partially and discard others. We can also decrease $k$ gradually. On the contrary, if we have more resources than the selected informative samples, we increase the $k$ accordingly. Thus we consider this point not a practical issue for our setting.
> >
> >
> > > W4. Continually adapting the model during test-time based on feedback from the oracle could lead to overfitting, especially if the test set is not representative of the broader data distribution.
> >
> > Thank you for raising this point! One of our major targets in ATTA is solving overfitting and experiments show ATTA indeed addresses this issue.
> > - **ATTA mitigates overfitting for TTA:** In the TTA field, overfitting and catastrophic forgetting (CF) is a typical and targeted issue. Our ATTA uses balanced entropy sample selections as the strategy to avoid overfitting and CF. The low-entropy samples are used to support the original model predictions/distributions, thus avoiding overfitting to the test samples.
> > - **Experimental evidence:** Our experiments have covered this situation. For example, the dataset PACS has four domains (photos, arts, cartoons, sketches). We train our model on photos and test it on other domains. The images in sketches have distributions very different from the original photo domains and are not representative of the broader data distribution definitely. However, even if we fine-tune the model to converge, we can observe in Table 2 that the source domain performance does not decrease (98.80%), indicating that our model does not overfit to the very different distribution.
> >
> > - Since our work locates in the field of FTTA, we target on-the-fly adaptation to continuous new domains during the test phase or application deployment. The test stream can exhibit distribution shifts varying with time, so we need continuous adaptation. We make great efforts considering the coverage of distribution and information with limited data. Incremental clustering and low-entropy-sample training techniques are designed to select informative samples, increase distribution coverage, and mitigate CF. Our experiments show their effectiveness in mitigating the CF problem (a model continually trained on a sequence of domains experiences a significant performance drop on previously learned domains).
> >
> > Besides the above points, we also provide further experiments and make efforts to clear the concerns and improve the clarity and organization of our paper in multiple aspects following the advice from other reviewers. We add visualization experiments and extend the ablation studies. Please refer to our revised paper and appendices for details.
> >
> > We sincerely thank you for your time! Hope we have addressed your concerns through practical efforts and shown the significance of our work. We look forward to your reply and further discussions, thanks!
> >
> > Sincerely,
> >
> > Authors

---

> > > ### Author Response · Authors · 2023-11-22
> > > **We would like to hear back from Reviewer 5EyE**
> > >
> > > Dear Reviewer 5EyE,
> > >
> > > Thanks again for your valuable comments and suggestions, which helps improve our work a lot. Since the end of the discussion period is approaching, we would like to follow up to see if our response addresses your comments. We would really appreciate it if you could check our response at your earliest convenience. Many thanks!
> > >
> > > We provide further experiments including visualization experiments and extended ablation studies, and make efforts to clear the concerns and improve the clarity and organization of our paper in multiple aspects.
> > > We hope that you could reply to our response and revision and reconsider your evaluation if we do have addressed your concerns. Also, please let us know if there are any additional questions or feedback. Thank you!
> > >
> > > Sincerely,
> > >
> > > Authors

---

### Official Review · Reviewer_BXz7 · 2023-11-08

**Soundness:** 4 excellent
**Presentation:** 3 good
**Contribution:** 4 excellent
**Rating:** 8
**Confidence:** 4

**Summary:**

This paper proposes a novel, formal problem setting of Active Test-Time Adaptation (ATTA), which incorporates active learning to perform test-time adaptation (TTA). It attempts to mitigate distribution shifts and catastrophic forgetting, while not being provided access to source data, model parameters, or pre-collected target samples.

A theoretical analysis of ATTA in the setting of binary classification is provided. First, the theory establishes a learning bound that has the notions of composition of training data, estimated distribution shift, and ideal joint hypothesis performance. Second, the fore-mentioned theory is utilized to shown that catastrophic forgetting can be mitigated by performing selective sample selection through entropy minimization.

SimATTA, a practical algorithm built upon the ATTA theory, is then developed. It integrates incremental learning and selective entropy minimization techniques. Empirical evaluations on four benchmarks simulating distribution shifts demonstrate the effectiveness of SimATTA when compared to the existing work. It achieves state-of-the-art accuracy under distribution shifts while maintaining the computational complexity that is not significantly higher than that of the prior work.

**Strengths:**

$\textbf{Novelty and significance}$:
In my opinion, the empirical results, especially addressing RQ1, clearly set this paper apart from previous research, paving the way to overcome distribution shifts under the TTA setting. The proposed algorithm, SimATTA, significantly surpasses the existing TTA algorithms in terms of performance accuracy under distribution shifts. It also appears to exhibit greater resilience to catastrophic forgetting. Additionally, it is supported by a robust theoretical framework.

$\textbf{Completeness and comprehensiveness}$:
The main manuscript and the supplementary material offer a comprehensive context and detailed information about the proposed work. Furthermore, related work addressing distribution shifts under various settings is also adequately discussed.

**Weaknesses:**

I found no major weakness from this paper. One minor aspect I would like to highlight concerns the clarity of the experimental settings, such as domain-wise data stream, random stream, post-adaptation, and so on. It took me some time to grasp all of these distinct settings. Perhaps including a dedicated section to explain about these settings would be more helpful.

**Questions:**

The “budget” term appears to be a significant factor in the algorithm. However, I’ve not been able to identify its relationship to the cluster centroid number. Could the authors please provide clarification on this matter?

I’m looking forward to seeing the code implementation of SimATTA.

---

> ### Author Response · Authors · 2023-11-19
> **Response to Reviewer BXz7**
>
> Dear Reviewer BXz7,
>
> We would like to extend our gratitude for your acknowledgment! We have made paper revisions following your suggestions to address your presentation concerns and would like to answer your questions as follows.
>
> > Weakness: One minor aspect I would like to highlight concerns the clarity of the experimental settings, such as domain-wise data stream, random stream, post-adaptation, and so on. It took me some time to grasp all of these distinct settings. Perhaps including a dedicated section to explain these settings would be more helpful.
>
> - Thank you for the valuable suggestion! We have restructured Sec 5 to include a separate paragraph explaining experimental settings, as well as describe the experimental settings in detail in Appx.G.
>
> > Questions: The “budget” term appears to be a significant factor in the algorithm. However, I've not been able to identify its relationship to the cluster centroid number. Could the authors please provide clarification on this matter?
>
> - Thank you for the question. We have revised descriptions to improve clarity. We also explain here.
> - The number of selected samples for each minibatch is decided jointly by the incremental clustering algorithm, the cluster centroid number $NC(t)$, and the stream in-coming distributions. Intuitively, this sample selection is a dynamic process, with **$NC(t)$ restricting the budget** and incremental clustering performing sample selection. For each batch, we increase $NC(t)$ as a **maximum limit**, while the exact number of the selected samples is given by the incremental clustering (by how many clusters are located in the scope of new distributions). *E.g.*, if the incoming batch does not introduce new data distributions, then we select zero samples even with increased $NC(t)$. In contrast, if the incoming batch contains data located in multiple new distributions, the incremental clustering "wants" to select more samples than the $NC(t)$ limit, but is forced by the limit to merge multiple previous clusters into one new cluster. Therefore, $NC(t)$ can restrict the budget but cannot determine the budget alone since the budget is also strongly affected by the data stream distributions.
> - The incremental clustering is detailed in Alg 2 (Appendix F), and $NC(t)$ is naively increased by a constant hyper-parameter $k$. Therefore, the budget is adaptively distributed according to the data streaming distribution with budgets controlled by $k$, which is also the reason why we compare methods under a budget limit as shown in Table 2.
>
> > I'm looking forward to seeing the code implementation of SimATTA.
>
> - We would like to provide our code through this anonymous link: https://anonymous.4open.science/r/tta-C807/
>
> Besides the above points, we also provide further experiments and make an effort to clear the concerns and improve the clarity and organization of our paper in multiple aspects following the advice from other reviewers. We add visualization experiments and extend the ablation studies. Please refer to the revised paper and appendices for details.
>
> We sincerely thank you for your time! Hope we have addressed your concerns through practical efforts and shown the significance of our work. We look forward to your reply and further discussions, thanks!
>
> Sincerely,
>
> Authors

---

> > ### Comment · Reviewer_BXz7 · 2023-11-21
> > **Thank you for the clarifications**
> >
> > I appreciate the authors for addressing my concerns and offering greater clarity.

---

> > > ### Author Response · Authors · 2023-11-21
> > > **Response to Reviewer BXz7**
> > >
> > > Dear Reviewer BXz7,
> > >
> > > Thank you for your kind reply and recognition of our paper! We are glad to know our rebuttal addresses your concerns. If any further discussions are needed, please let us know. Thanks!
> > >
> > > Sincerely,
> > >
> > > Authors

---

### Official Review · Reviewer_cDV9 · 2023-11-08

**Soundness:** 4 excellent
**Presentation:** 3 good
**Contribution:** 4 excellent
**Rating:** 8
**Confidence:** 5

**Summary:**

This paper proposes a new setting, namely ATTA (Active Test-Time Adaptation), to integrate active learning (a limited number of labeled test samples) within test-time adaptation (TTA) to enhance TTA under domain shifts. The problem itself is practically important and the paper presents a theoretical guarantee and a sound solution (SimATTA) that combines incremental clustering and entropy selection to conduct online sample selection, avoiding catastrophic forgetting issues. The simplicity of the proposed algorithm and its successful application on real datasets are commendable. A thorough empirical study (including ablations) with encouraging results (both in the paper and its supplementary material) shows the effectiveness of the approach in terms of its effectiveness and generalization performance. The paper's clear contribution to the TTA community is evident, and I recommend its acceptance.

**Strengths:**

**Originality**: The paper introduces a novel setting, active test-time adaptation (ATTA) with theoretical guarantees for alleviating distribution shifts and mitigating catastrophic forgetting and extensive experiments on several benchmarks under domain generalization shifts. Additionally, the Section FAQ & Discussions in supplementary material is highly praiseworthy.

**Quality**: The paper provides a thorough experimental evaluation of the SimATTA algorithm on four datasets (PACS, VLCS, Office-Home, and Tiny-ImageNet-C). The paper also conducts ablation studies to analyze the impact of different components of SimATTA. The paper demonstrates that SimATTA can achieve superior performance and maintain efficiency.

**Clarity**: The paper provides sufficient background information including theory and closely related works to situate the contribution of the ATTA setting and the SimATTA algorithm.

**Significance**: The paper proposes an important and challenging setting of active test-time adaptation (ATTA) and a detailed comparison with related settings (DA/DG, TTA, ADA, ASFDA, and AOL) highlights the value of the proposed ATTA. ATTA also has many potential practical utilities such as an autopilot system and a personalized chatbot discussed in supplementary material.

**Weaknesses:**

**Insufficient visualization**: Though the authors have provided detailed algorithms (Alg. 1 and Alg. 2) to show the proposed SimATTA algorithm, it is still hard to follow the whole picture quickly. Thus it could be better to provide a clear diagram to illustrate the framework of the SimATTA algorithm.

**Insufficient justifications**: For example, regarding the **efficiency** and **applicability** of ATTA, some justifications are missing in this paper. First, as shown in Tab. 3, the time cost of ATTA is around ten times than general FTTA (Tent: 68.83, EATA: 93.14, SimATTA: 736.28). The reason might be the clustering-based selection process and fully fine-tuning pre-trained models? Second, though the authors state that "ATTA can be applied to any pre-trained models including large language models (LLMs)", they provide no experimental results.

**Inconsistent results**: results of SimATTA ($B\le$500) in Table 2 (TTA comparisons on PACS) and Table 8 (Ablation study on PACS) are different.

**Questions:**

1. As an active sampling algorithm, how to define an informative test sample, especially on streaming data? The authors might provide some visualization results for better understanding.

2. How about the cost of the ATA training set?  It seems that the SimATTA algorithm will keep a training set (the maximum size is $\mathcal{B}$?) during the test-time adaptation, would this strategy violate the nature of test-time adaptation, i.e., real-time?

3. There are many hyper-parameters in this work, such as two entropy thresholds $e_l$ and $e_h$, number of cluster centroid budget $NC(t)$, centroid increase number $k$, etc. The question is how to choose them for different datasets. In some sense, this is not the weakness/limitation of this particular paper but rather applies to the whole AL paradigm.

4. It is unclear what the meaning of `steps=10` is. And what is the config of SimATTA? SimATTA (`steps=1`) or SimATTA (`steps=10`)?

5. How to deal with an extreme situation in which only one sample is in a mini-batch, i.e., the batch size is 1.

6. Another minor question is, why the performance of Enhanced TTA on Tiny-ImageNet-C (severity level 5) is poor? Also, why the baseline results (CLUE) on VLCS of Tab. 4 are too low, even worse than that of the Random method?

7. In Section 5.2, "randomly select labeled samples and fine-tune them with `their selected pseudo-label samples.`" Is it a mistake?

---

> ### Author Response · Authors · 2023-11-19
> **Response to Reviewer cDV9 (Part I)**
>
> Dear Reviewer cDV9,
>
> Thank you for your acknowledgment of our work and insightful comments! We have made paper revisions following your suggestions and would like to address your concerns by answering your questions one by one as follows.
>
> > W1. Insufficient visualization
>
> - Thank you for your suggestion. We have created a figure (Fig.2) in the revised paper to demonstrate the algorithm more intuitively. We inserted the figure in the main paper and moved Alg. 2 to the appendix due to the space limit. We hope this modification is satisfactory.
>
> > W2. Insufficient justifications
>
> - Following your suggestion, we have added additional justifications in the revised paper. For your convenience, we summarize important changes and explanations as follows.
>
> - **Efficiency**: ATTA's higher time cost is due to the difference in training steps. In this experiment, SimATTA has a training step of 10 and a similar time cost as SAR per step. In Table 3, we aim to show that simply accessing labeled samples cannot benefit TTA methods to match ATTA. With 10 training updates (step=10) for each batch, FTTA methods would suffer from severe CF problems. In contrast, ATTA covers a statistically significant distribution, achieving stronger performances with 10 training updates or even more steps till approximate convergences. In fact, longer training in Tent (step=10) leads to worse results (compared to step=1), which further motivates the design of the ATTA setting.
> In brief, ATTA is able to apply statistically significant training processes without CF, which requires sufficient training and is the reason for the extra time cost.
>
> - **Applicability**: "ATTA can be applied to any pre-trained models including large language models (LLMs)" demonstrates a possible future direction. In essence, ATTA is **model agnostic**, which is theoretically applicable to various machine learning models. In contrast, many FTTA methods only allow training the batch normalization layers. However, we acknowledge that we do not cover LLM experiments in this work, and applying ATTA to LLM may encounter unknown results.
>
> > W3. Inconsistent results
>
> - Thanks for pointing out. These inconsistent results come from a different hyperparameter $k$ selection in the ablation study to ensure that different variants have the number of budgets as similar as possible. We adjusted one $k$ in the ablation study since, in that part of the experiment, the budget number was much less than other variants. To clarify this ablation comparison issue and eliminate confusion, we have extended the ablation study by conducting experiments with more budget-limit selections and using line charts instead of tables for comprehensive comparisons. Please refer to our revised ablation study (Appx I.2) for more details.
>
> > Q1
>
> - Thanks for the advice. We have added visualization results and explanations in Appx F.2 for better illustration. In each batch, after the high/low entropy sample selection, we examine the high entropy samples to select informative ones. Assume we have 3 previously selected anchors, and we now apply a Kmeans of 4 centroids. After clustering, assume 3 centroids are located near the 3 old anchors since the 3 anchors' weights are much larger than other samples in this batch. Then the left centroid should statistically locate at the center of the distribution representing new test samples, which should be far from the 3 old anchors (thus, has not been covered by them). These new samples that are far from the previously known anchors are **informative** to us. Finally, we choose the closest sample to the centroid as a new anchor, i.e., the informative test sample.
>
> > Q2
>
> - (1) Yes, the number of labeled training sets is limited by $\mathcal{B}$, and we also keep an unlabeled low-entropy training set of similar scale. Therefore, the overall space complexity is $O(\mathcal{B})$.
> - (2) In our opinion, many real-time applications such as robotics, autonomous vehicles, medical imaging adaptations, and user interface personalizations, are not very time-sensitive. For example, retraining a personalized user interface recommendation system may require hours, days, or weeks, but ATTA can fine-tune the model in a second (or seconds) that should be considered in a real-time manner. As we have demonstrated (detailed in Appx B.Q8), the speed of ATTA is sufficiently fast for an autopilot system, and the speed requirements of the other applications listed above are safely within the limit of ATTA. Therefore, we believe ATTA is a real-time setting.
>
> - Rigorously, we do acknowledge that the application scopes of ATTA and FTTA are not completely overlapping. FTTA can handle applications that are extremely time-sensitive, while ATTA provides results of much higher quality.

---

> ### Author Response · Authors · 2023-11-19
> **Response to Reviewer cDV9 (Part II)**
>
> > Q3
>
> - In experimental settings, $k$ is selected to approximate the budget limit $\mathcal{B}$ we would like to observe, which is for comparison purposes. Selecting $e_l$ is a trade-off between accuracy and the number of low-entropy samples. We select $e_l$ as low as possible while the number of low-entropy samples can still maintain the low/high entropy balance. Therefore, the $e_l$ selection is related to the distribution of the entropy in the specific dataset.
>
> - The hyperparameter searching spaces used in the paper are provided as follows:
>
>   - $e_l: \{10^{-4}, 10^{-3}\}$
>
>   - $e_h: 10^{-2}$
>
>   - $k: [0, 3]$
>
>   - $NC(t)$ is controlled by $k$, $i.e.$, $NC(t)=10 + kt$, where 10 is the initial number of centroid.
>
> - In real-world scenarios, we should select $k$ according to our oracle resource available. Instead of using $e_l$ independently, we believe considering both the threshold $e_l$ and an extra selection ratio can be a good practice.
>
> > Q4
>
> - `steps=10` indicates 10 gradient backpropagation iterations for one test-time adaptation batch. The hyperparameter `steps` should be set for Tent and SAR not only because of their original settings but also because their pseudo-label training lacks statistical significance and thus requires a hyperparameter to control the training process. In experiments, we found that using a tolerance count (tol) is more natural because SimATTA maintains a statistically significant training set. SimATTA will stop updating once the loss does not decrease for more than 5 steps.
> - For TinyImageNetC in Table 3, SimATTA uses `steps=10` for time comparisons with others since other methods use at most 10 steps. We have added this information to the table.
>
> > Q5
>
> - ATTA handles this extreme situation by accumulating these samples into a buffer. Sample selections and training are executed once the buffer is full, e.g., setting it to the length of 100.
> This process is less efficient than TTA but necessary since one of the fundamental differences between ATTA and FTTA is that ATTA tries to fine-tune the model using a statistically significant distribution. This is a trade-off between effectiveness and efficiency.
>
> > Q6
>
> - (1) Enhanced TTA, though having access to labeled samples, is essentially an FTTA setting, *i.e.*, it does not maintain a statistically significant training set. Therefore, these enhanced TTA methods still suffer from CF problems. The Tiny-ImageNet-C comparisons aim to address the potential question of whether access to labeled samples is the only reason for improvements and for corresponding fairness concerns.
> - (2) In 3 runs, CLUE has source domain results: $74.06$, $84.38$, and $98.66$. This indicates CLUE suffers from severe CF problems in 2 out of 3 runs. One simple explanation for these failures is that the selected samples are too far from the source distribution, which induces CF and overfitting problems. To be specific, the high entropy consideration of CLUE leads to the failure of choosing low-entropy samples. As demonstrated in Sec. 3.2, compared to our mitigation of CF with balanced high/low entropy samples, CLUE fails to maintain this balance in VLCS and shows catastrophic performances.
>
> - The reason why the method Entropy does not suffer from CF problems is that the ADA training process includes 10 selection rounds in their settings so that a possible minor CF problem after the first round leads to the abrupt entropy increase of source-like samples. These samples are then selected by the Entropy method in the 2nd round, thus preventing further CF problems. In contrast, CLUE fails to select these source-like samples in time (before severe CF) because of diversity considerations. Therefore, we can conclude the failure of CLUE as that, considering uncertainty and diversity leads to better informative sample selections but may cause worse source-like sample selections.
>
> > Q7
>
> - Sorry for the confusion. We have revised the unclear expression. This sentence means that, in addition to each TTA's original updates, we further enhance these TTA methods by fine-tuning the model with randomly selected labeled samples. Traditional TTA methods are generally designed to update models using their own pseudo-labeled samples.
>
>
> Besides the above points, we also provide further experiments and make an effort to clear the concerns and improve the clarity and organization of our paper in multiple aspects following the advice from other reviewers. Please refer to the revised paper and appendices for details.
>
> We sincerely thank you for your time! Hope we have addressed your concerns through practical efforts and shown the significance of our work. We look forward to your reply and further discussions, thanks!
>
> Sincerely,
>
> Authors

---

> > ### Comment · Reviewer_cDV9 · 2023-11-20
> > **Thanks for the detailed clarifications and revisions！**
> >
> > I now feel more strongly towards acceptance. Best of luck!

---

> > > ### Author Response · Authors · 2023-11-21
> > > **Response to Reviewer cDV9**
> > >
> > > Dear Reviewer cDV9,
> > >
> > > Thank you for your kind reply and recognition of our paper! We are glad to know our rebuttal addresses your questions. If any further discussions are needed, please let us know. Thanks!
> > >
> > > Sincerely,
> > >
> > > Authors

---

### Author Response · Authors · 2023-11-20
**Global response: Summarization of the major contributions , common concerns, and paper revision**

We would like to extend much gratitude to the insightful and keen comments from all Reviewers. We're glad that the novelty of our paper is acknowledged by all Reviewers and we are able to further improve our work with valuable suggestions.

In this global response, we would like to summarize the major contributions of this paper, address the common concerns, and summarize the paper revision for your convenience.

## Major contributions

* Introduction of the novel ATTA setting
* Theoretical and empirical analysis for solving domain distribution shifts and mitigating catastrophic forgetting
* The proposed SimATTA method:
  * Entropy balanced sample selection
  * Incremental clustering
* Experimental comparisons: ATTA versus TTA, enhanced TTA, and ADA methods

## Common concern

The common concern is the efficiency of ATTA. We would like to clarify that the time results in Table 3 are for **10-step** SimATTA, while SAR (step=10) has a very similar time cost. Therefore, SimATTA has similar **time per step** efficiency with other TTA methods. We apologize for the confusion and have revised the paper to make the descriptions clearer in Sec. 5.2 and Appx. G.2.2.

Moreover, the major advantage of ATTA is that the test-time training process is statistically significant, which allows many more training steps without catastrophic forgetting issues. While the total time cost of SimATTA mainly comes from the training steps, one can adjust the training step setting for the trade-off between effectiveness and efficiency.

## Limitations

This paper mainly focuses on the establishment of the ATTA setting and the solution to distribution shifts and catastrophic forgetting theoretically and empirically. We acknowledge that there are many possible applications and potential future directions, *e.g.*, considering scalability to very large datasets and models. However, they cannot be covered by this single paper and we leave them out of the current scope. We appreciate many intriguing and insightful comments and have accommodated the discussions in our FAQ (Appx. B).

## Revision

We have revised the paper following Reviewers' suggestions. The major modifications are summarized as follows.

* Sec. 3.2: Rigorous descriptions on entropy (Reviewer yeNU)
* Figure 2: An overview and illustration of SimATTA (Reviewer cDV9)
* Sec. 4.2: Incremental clustering explanation modifications (Reviewer BXz7)
* Sec. 5: Separate experimental setting descriptions. (Reviewer BXz7)
* Sec. 5.2: Clearer time/efficiency comparison explanations. (Reviewer yeNU, Reviewer 5EyE, Reviewer cDV9)
* Appx. B Q7. (Reviewer cDV9, Reviewer 5EyE)
* Appx. F: New **incremental clustering visualization experiments** and explanations. (Reviewer cDV9)
* Appx. I.2: The **extended ablation study experiments** with more budget selections and clearer explanations. (Reviewer cDV9, Reviewer yeNU)


We thank all reviewers again for the dedicated reviews. Hope our clarifications and revisions have addressed your concerns and shown the significance of our work. We look forward to further comments and discussions.

Sincerely,

Authors

---

### Meta-Review · Area_Chair_NeWr · 2023-12-08

**Metareview:**

The paper introduces Active Test-Time Adaptation (ATTA), an innovative framework that integrates active learning into test-time adaptation to address domain shifts and catastrophic forgetting in machine learning models. The authors develops SimATTA, which employs incremental clustering and entropy selection for online sample selection, aiming to enhance the efficiency and accuracy of models under domain shifts. The paper substantiates its claims with theoretical analysis and empirical studies across several benchmark datasets, demonstrating SimATTA's superior performance and generalizability compared to existing methods.

Key strengths (+) and weakness (-) are summarized as below:

Strength:

+ Novelty: The concept of ATTA is original. It represents a novel approach to combining active learning with test-time adaptation.
+ Theoretical rigor: The paper presents a sound theoretical foundation for ATTA, including learning bounds and mitigation strategies for catastrophic forgetting.
+ Empirical results: Extensive empirical studies validate the efficacy of the SimATTA algorithm, showing promising results in domain generalization and outperforming existing methods.
+ Presentation: The paper is well-structured, providing clear explanations and thorough discussions of the methodology, results, and related work.

Weakness:
- Lack of clarity in visualization: The paper could benefit from more intuitive visualizations or diagrams to illustrate the SimATTA algorithm and its framework.
- Theoretical concerns: Some aspects, like the efficiency of ATTA and the use of entropy as a metric, lack sufficient justification in the original submission.
- Scalability of ATTA to larger, real-world datasets is not adequately addressed.

**Justification For Why Not Higher Score:**

The lack of clear visualizations makes it challenging for readers to quickly grasp the SimATTA framework. Additionally, there're some concerns about the scalability and practical applicability in real-world scenarios, along with inconsistencies in the results in the current form.

**Justification For Why Not Lower Score:**

The ATTA framework provides a new perspective on addressing domain shifts and catastrophic forgetting. The empirical evidence demonstrating the efficacy of SimATTA is robust and convincing.

---

### Decision · Program_Chairs · 2024-01-16

Accept (poster)